# AssetOpsBench: Benchmarking AI Agents for Task Automation in Industrial Asset Operations and Maintenance

## Abstract

AI for Industrial Asset Lifecycle Management aims to automate complex operational workflows, including condition monitoring, maintenance planning, and intervention scheduling, thereby reducing human workload and minimizing system downtime. Traditional AI/ML approaches have primarily tackled these problems in isolation, solving narrow tasks within the broader operational pipeline. In contrast, the emergence of AI agents and large language models (LLMs) introduces a next-generation opportunity: enabling end-to-end automation across the entire asset lifecycle. This paper envisions a future where AI agents autonomously manage tasks that previously required distinct expertise and manual coordination. To this end, we introduce AssetOpsBench, a unified framework and environment designed to guide the development, orchestration, and evaluation of domain-specific agents tailored for Industry 4.0 applications. We outline the key requirements for such holistic systems and provide actionable insights into building agents that integrate perception, reasoning, and control for real-world industrial operations.

## 1 Introduction

Industrial assets, such as data center chillers (Naug et al., 2024) and wind farms (Monroc et al., 2024), are complex, multi-component systems that generate vast amounts of multimodal data, including time-series sensor readings, textual inspection and workorder records, operational logs, and images, throughout their lifecycle. The ability to monitor and interpret heterogeneous data from diverse sources, such as IoT SCADA (WikiSCADA) (Supervisory Control and Data Acquisition) sensors, operational KPIs, failure mode libraries, maintenance work orders, and technical manuals, is key to effective Asset Lifecycle Management (ALM) (WikiALM). However, subject matter experts such as maintenance engineers, site operators, and plant managers face considerable challenges in synthesizing insights from these disparate data streams to support timely and condition-aware decisions. As highlighted in Figure 1(a), the scale, semantic diversity of assets, and application-specific contexts often render traditional monitoring and management systems inadequate.

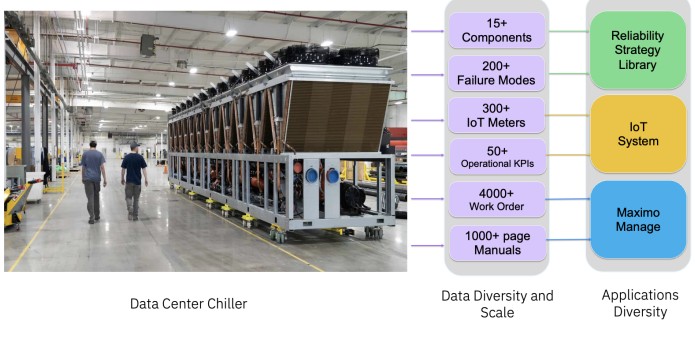

Data Center Chiller

((a)) Complex Industrial Asset – Data Centers managing Chiller and Air Handling Units (AHUs)

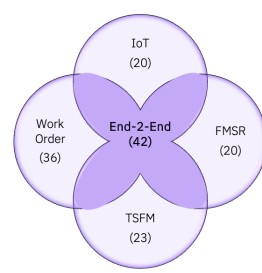

((b)) Distribution of open-sourced scenarios for benchmarking agents in a simulated environment.

To address these challenges, the research and industrial communities are increasingly turning to AI agents: autonomous and goal-driven systems capable of integrating data across silos, reasoning over complex conditions, and triggering appropriate actions. AI agents are particularly promising in the context of Industry 4.0, where the confluence of real-time IoT telemetry (e.g., Oracle IoT (Oracle, 2025), enterprise asset management (EAM) systems (WikiEAM), and IBM Maximo (IBM)) and reliability engineering frameworks necessitates scalable and intelligent automation. These agents promise to support a wide range of industrial workflows, from anomaly detection to maintenance scheduling, by bridging the gap between raw sensor data, maintaiance report, work-order and business-level insights.

Despite recent advances in LLM-based agent frameworks, such as ReAct (Yao et al., 2023), HuggingGPT (Shen et al., 2023), Chameleon (Lu et al., 2023), and recent generalist agent models (Fourney et al., 2024; Marreed et al., 2025), a gap remains in adapting these innovations for real-world industrial settings. Most recent domain and application specific benchmarks (e.g., ITBench (Jha et al., 2025), SWE-bench (Chan et al., 2025), $\tau-$bench (Yao et al., 2024) and its extension (Fu-Hinthorn, 2025), Customer Support Benchmarks (Team, 2025), TheAgentCompany (Xu et al., 2024b), CRMArena-Pro (Huang et al., 2025)) are tailored toward machine learning, IT, customer-service domains, or purely digital, knowledge-work settings rather than physical, sensor-driven industrial operations. These benchmarks do not address the unique challenges of industrial applications, such as data modality diversity (time series and text), business object complexity(e.g., failure mode, work orders, asset hierarchies), and task collaboration across multiple operational personas (e.g., reliability engineers and data scientists).

This paper introduces **AssetOpsBench**, the first dataset and benchmarking system designed to evaluate AI agents for real-world industrial asset management tasks. By leveraging experts in development, we have carefully built real multi-source datasets, intent-aware scenarios, and domain-specific agents to develop, evaluate, and compare multi-agent systems. Our system includes:

- A catalog of **domain-specific AI agents**, including an IoT agent, a failure mode to sensor mapping (FMSR) agent, a foundation model-driven time series analyst (TSFM) agent, and a work order (WO) agent. Each agent has tools and targets different modalities and tasks.

- A curated to be open-source **intent-driven** 141 scenario of **human-authored natural language queries**, grounded in real industrial data center operations (Figure 1(b)), covering tasks such as sensor-query mapping, anomaly detection, failure diagnosis, and work-order modeling.

- A **simulated industrial environment** based on a CouchDB-backed IoT telemetry system and **real** multi-source dataset, enabling end-to-end benchmarking of multi-agent workflows and open source contributions without the constraints associated with production systems.

- A comparative analysis of multi-agent architectural paradigms: **Agent-As-Tool** vs. **Plan-Execute**, highlighting tradeoffs between interleaved decomposition or decomposition-first execution.

- A three-pronged evaluation consisting of (i) an LLM-based rubric, (ii) reference-based scoring of task decomposition and execution, and (iii) manual expert verification for certain scenarios.

- A systematic procedure for the **automated discovery** of emerging failure modes in multi-agent systems, extending beyond fixed taxonomies and its benefits.

Our motivation for a **multi-agent architecture** arises from industrial deployment experience, where heterogeneous workflows benefit from modular, task-specific agents (Figure 1(a)). We complement this practical insight with empirical evidence (Section 5.1) showing that multi-agent orchestration outperforms single-agent baselines on complex, composite tasks. For instance, sensor data may be handled by an IoT agent, while fault history is managed by an FMSR agent. These agents must collaborate intelligently to answer user queries, such as "Why is the chiller efficiency dropping?", which blend physical reasoning, historical correlation, and operational semantics. Furthermore, the design of agent workflows must respect the **natural language and intent patterns** used by industrial end users. Unlike IT users, operators and engineers often refer to assets in physical or operational terms (e.g., "chiller performance","oil temperature spike") rather than referring to database fields or ontologies. Crafting robust benchmarks requires capturing this domain-specific linguistic variance,

ensuring agents not only retrieve correct answers but also follow reasoning patterns aligned with domain expectations.

Finally, we experimented with an additional closed-source **162 scenarios** to demonstrate generality, spanning 10 asset classes, 53 failure modes, and 20 sensors. These include 42 live-deployment scenarios (>90% correctness verified by a domain expert), 17 hydraulic system, 15 metro train, and 88 failure-mode scenarios encompassing diverse asset–failure–sensor relationships.

## 2 RELATED WORK

**Generalist Agents.** The development of generalist agents capable of orchestrating multiple sub-agents to accomplish complex tasks has emerged as a prominent research direction. This paradigm is evident across various domains, including web systems such as Magentic (Fourney et al., 2024) and CUGA (Marreed et al., 2025), multimodal agents like GEA (Szot et al., 2024), and software engineering platforms like HyperAgent (Huy et al., 2025), ChatDev (Qian et al., 2024), and MetaGPT (Hong et al., 2024). These agents typically employ predefined sets of sub-agents, such as terminals, browsers, code editors, and file explorers, each assigned specific functional roles to facilitate task decomposition and planning. While this architecture enables targeted integration and task specialization, it often lacks flexibility. Most systems adopt hard-coded reasoning paradigms, such as plan-execute or ReAct, which limit their capacity to support new agents, adapt to novel task, or alternative coordination strategies, such as AOP (Li et al., 2025) and Prospector (Kim et al., 2024).

**Domain-Specific Agents.** Solving specialized tasks often requires domain-specific capabilities, prompting the development of tailored benchmarks such as MLEBench (Chan et al., 2025) and MLAgentBench (Huang et al., 2024) Arena. These frameworks evaluate agents on a diverse set of machine learning problems, such as classification and regression, across multiple modalities, including tabular and image data. They simulate end-to-end workflows, from resolving GitHub issues to automating model training and evaluation pipelines. The concept of the *AI Research Agent* has gained traction, referring to agents built for scientific discovery and iterative experimentation. For example, MLGym (Nathani et al., 2025), a research agent in machine learning workflows. However, most current benchmarks lack support for temporal and text data modalities together, which are crucial in domains such as physical asset health monitoring.

**Application-Specific Agents.** Agent-based automation is also advancing in operational settings, such as IT operations, customer support, and compliance monitoring. Frameworks developed under initiatives like ITBench (Jha et al., 2025) and AIOpsLab (Chen et al., 2025) aim to replicate real-world scenarios involving site reliability engineering, diagnostics, and system auditing. These systems reinforce the importance of application-specific benchmarks, tailored to specific personas, that not only evaluate agents across structured tasks but also expose capability gaps and drive innovation in reasoning and orchestration strategies. Current benchmarks in this space tend to be domain-specific in scope, lacking the generality and composability required to assess agent performance across diverse, multi-agent environments, especially those involving cross-modal reasoning or domain-specific tool usage.

**Fine-Tuned and Compact Models.** Recent work has improved agent performance via fine-tuned language models, often called *Large Action Models (LAMs)*, designed to execute structured actions within environments. Systems such as TaskBench(Shen et al., 2024), xLAM(Zhang et al., 2025b), AgentGen (Hu et al., 2025), AgentBank (Song et al., 2024), AgentRM (Xia et al., 2025), FireAct (Chen et al., 2024), and ActionStudio (Zhang et al., 2025a) exemplify this trend. They are often trained in grounded environments (e.g., Windows-based (Wang et al., 2025)) and evaluated on tasks such as arithmetic, programming, and web interaction. While effective, these models remain limited to textual or web environments and have yet to demonstrate applicability to complex industrial automation with hybrid agent compositions.

**Open Challenges.** Despite these advances, several gaps remain. First, there is a lack of comprehensive benchmark datasets targeting industrial asset domains, particularly those involving condition-based monitoring, predictive maintenance, automated diagnostics, and work order planning. To support this claim, we analyzed **a catalog of 135 public datasets** (jonathanwvd, 2025) and found that only one dataset includes any form of work-order or operational context, and even that lacks sensor history. Moreover, only 53 datasets mention failure modes, most of which contain just one or

two modes, and none of the datasets support agentic applications. Second, time-series data, which plays a central role in industrial and infrastructure-related applications, remains underrepresented in existing agentic benchmarks. Finally, few systems support orchestration across heterogeneous agents, including those based on text, code, or simulations, nor do they offer modular reasoning strategies adaptable to complex, multi-agent workflows. Addressing these gaps is essential to advance general-purpose agent intelligence in high-stakes, real-world domains.

# 3 PROBLEM AND APPROACH: INTELLIGENT AGENT-BASED ASSET OPERATIONS

Industrial asset operations involve complex, heterogeneous workflows where maintenance engineers, reliability specialists, and facility planners must interpret multi-modal sensor data, detect anomalies, and make timely operational decisions. Interdependent tasks such as root cause analysis, predictive maintenance planning, work order bundling, and service request initiation often require reasoning across historical telemetry, asset metadata, and operational constraints. Meeting these demands requires intelligent agents that can decompose high-level requests into structured, executable subtasks, coordinate across multiple domain-specific modules, and integrate outputs into actionable recommendations. For example, a user might request: "Help configure an anomaly detection model to monitor power consumption of `CUXP` and trigger alerts when usage is projected to exceed 8 Watts above the maximum deviation observed over the past 30 days," enabling timely corrective actions such as service request creation. The diversity and interdependence of these tasks, spanning data interpretation, anomaly reasoning, and operational decision-making, underscore the need for a coordinated, intelligent agent framework capable of handling complex industrial workflows.

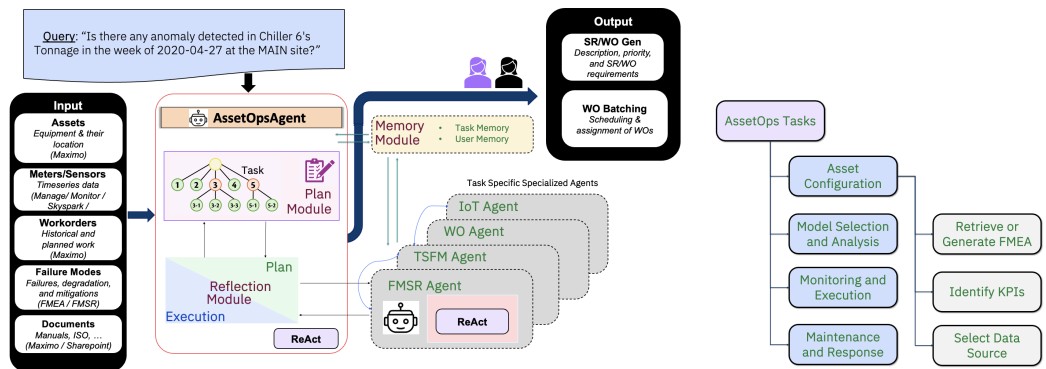

((a)) Architecture of the Multi-Agent System: Time Series (TSFM) Agent, Failure Mode Sensor Relations (FMSR) Agent, Work Order (WO) Agent

((b)) Exemplar AssetOps Task Hierarchy

Figure 2(a) illustrates the core components of our proposed framework. At the center is the **AssetOps** Agent, which functions as a global coordinator. It interprets high-level user queries in natural language, decomposes them into structured subtasks, delegates these to specialized functional subagents, and integrates their outputs into coherent responses, such as generating service requests or work orders. To handle tasks like configuring anomaly detection models or triggering alerts for assets such as `CUXP`, this coordination is essential. While typical multi-agent systems for general-purpose tasks (e.g., Magentic (Fourney et al., 2024)) consist of an orchestrator or supervisor agent coordinating sub-agents such as coders, file system handlers, terminals, or web-surfing agents, in industrial settings these sub-agents are replaced by domain-inspired, task-specific agents. Examples include an IoT agent, a failure mode to sensor mapping (FMSR) agent, a foundation model-driven time series analyst (TSFM) agent, and a work order (WO) agent. These agents are specifically tailored to monitor, analyze, and generate work orders or service requests for physical assets.

Building on this multi-agent architecture, defining a systematic benchmark requires determining the set of tasks that accurately reflect real-world industrial operations. In this paper, we leveraged ISO standards to construct a structured task taxonomy aligned with the stages of physical asset management (ISO-2024, 2024; ISO, 2016). This taxonomy provides a consistent and scalable approach for

scenario generation, ensuring that each task maps to realistic operational objectives and decision-making workflows. We refer to this methodology as **intent-driven** scenario generation, in contrast to the **API-driven** scenario generation popularized in (Yao et al., 2024; Shen et al., 2024).

As illustrated in Figure 2(b), the taxonomy begins with **Asset Configuration**, encompassing activities such as retrieving Failure Mode and Effects Analysis (FMEA) documentation and selecting performance KPIs, typically carried out by reliability engineers. It progresses to **Model Selection and Analysis**, where data scientists apply anomaly detection models and use LLM-powered retrieval to surface relevant historical failures. In the **Monitoring and Execution** phase, operations teams manage live telemetry, refine detection models, and enforce safety guardrails. Finally, the **Maintenance and Response** phase focuses on actionable outputs, including generating work orders, summarizing system health, and prioritizing interventions tasks typically handled by maintenance engineers. Grounding task definitions and APIs in ISO standards allows the benchmark to generalize across diverse industrial software platforms (Oracle, 2025; IBM).

## 4    AssetOpsBench

**AssetOpsBench** consists of a real multi-asset, multi-source dataset from a data center, 141 manually constructed task scenarios, and a benchmarking environment with task-specific AI agents and an evaluation framework. The scenarios were developed over 18 months in collaboration with reliability engineers, controls specialists, and domain experts overseeing assets such as AHUs, chillers, boilers, and compressors. Experts identified key failure modes, drafted scenario templates capturing realistic fault signatures and cross-sensor interactions, and iteratively refined them through multiple review cycles to ensure plausibility and alignment with diagnostic reasoning. Each scenario is grounded in operational and reference data, including sensor telemetry from industrial HVAC systems (fifteen-minute intervals from BMS(WikiBMS) and SkySpark(SkyFoundry contributors)), work orders from a product-level Maximo, and FMEA information from the Reliability Strategy Library for data center operations. This combination ensures that the scenarios are both expert-validated and data-driven, faithfully reflecting real-world industrial conditions.

### 4.1    Multi-Source Dataset

A key distinguishing feature of **AssetOpsBench** is its integration of richly structured, expert-curated multi-source data that reflects the complexity of real-world industrial asset operations. Unlike a simple data-gathering effort, constructing this benchmark required extensive data cleaning, the development of a novel failure taxonomy, and careful alignment across heterogeneous sources.

Table 1: Key data modalities with 3 Example Fields used for open source scenario construction

| Data Source | Field | Description |
|---|---|---|
| **Sensor Data*** 
 # Industrial Assets: 6 
 Quantity: 2.3M points | Chiller Return Temp. 
 Chiller % Loaded 
 Condenser Water Flow | Measures temperature of water returning to chiller 
 Indicates current load as a fraction of the maximum 
 Indicates the current flow rate through the condenser |
| **FMEA** 
 # Industrial Assets: 3 
 Quantity: 53 records | Failure Location / Comp. 
 Degradation Mechanism 
 Degradation Influences | Subsystem/part where failure occurs (e.g., bearings,) 
 Physical process driving failure (e.g., wear, erosion) 
 Stressors like runtime, fluid quality, or shock loading |
| **Work Orders** 
 # Ind. Assets: 10+ 
 Quantity: 4.2K records | ISO Failure Code 
 Event Log Timestamp 
 Linked Anomaly / Alert | Standardized classification of the failure category. 
 Time-marked entry recording an operational event 
 References to alerts or anomalies tied to work order |

As shown in Table 1, the benchmark includes over 2.3 million sensor data points across 6 assets (4 *Chillers* and 2 *AHUs*), capturing time-series signals such as *chiller return temperature*, *load percentage*, and *condenser water flow*. The structured failure models, derived from Failure Mode Effects Analysis (FMEA) records, encompass 53 failure entries across three equipment assets. FMEA provides provide detailed insights into the physical locations of failures, degradation mechanisms (such as *wear* and *erosion*), and the influencing factors (including *runtime*, *fluid conditions*, and *shock loading*) that contribute to each failure. Work order histories span 4.2K records across 10+ assets

and 11 years and incorporate ISO-standard failure codes, event timestamps, and linkages to alerts and detected anomalies.

Additionally, the operational system generates a temporal sequence of alarm logs and also leverages domain-specific technical rules obtained from experts, enabling contextual grounding of operational anomalies. This diverse data foundation, comprising 9 modalities (Sensor, Work Order, Alert, Alarm, FMEA, Anomaly, KPI 2 Failure Codes, Events, Rule 2 Failure Code), facilitates a comprehensive evaluation of decision-making, tool usage, and multi-hop reasoning in industrial environments.

## 4.2 SCENARIO DESIGN AND COVERAGE

Each scenario in **AssetOpsBench** represents a structured operational query grounded in the lifecycle-aligned task taxonomy (Figure 2(b)) and asset-specific datasets (Table 1). Each scenario is formalized as:

$$P = \langle id, type, text, category, form \rangle$$

where *id* is a unique identifier; *type* specifies the task type (e.g., knowledge retrieval, analytical); *text* is the natural language query; *category* denotes the operational domain (e.g., IoT, FMSR, TSFM, WO or End-2-End (i.e., more than one agent)); and characteristic *form* defines the expected output (e.g., explanation, API call, action plan). Scenarios are categorized into two types: (1) single-agent utterances, which only require probing a single specific agent (e.g., IoT, TSFM, FMSR, WO), and (2) multi-agent tasks, which span multiple agents and require coordinated reasoning and data exchange. As shown in Figure 1(b), the to be open-sourced version comprises a total of 141 scenarios, consisting of 99 single-agent and 42 multi-agent tasks.

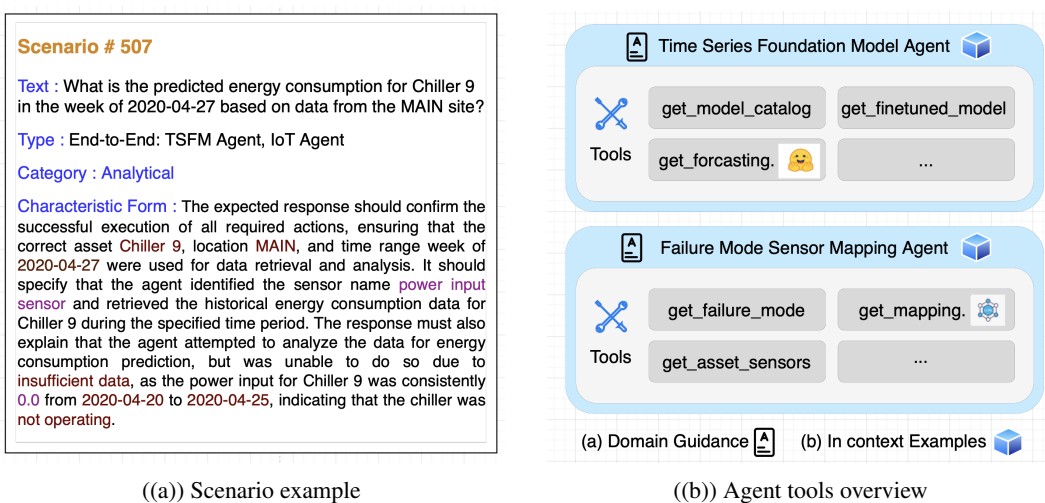

((a)) Scenario example          ((b)) Agent tools overview

Figure 3: Left: Scenario illustration. Right: Overview of two representative agent tools.

Figure 3(a) presents **Utterance 507**, an instructive case where a user requests a prediction of future energy consumption. To address this query, the agent must first reason about which sensor variable to use, specifically the power input, and after retrieving the data recognize that most values are zero, indicating an insufficient data condition. This scenario highlights the importance of subject matter experts (SMEs) in designing tasks that assess the reasoning capabilities of LLMs, rather than merely testing tool functionality. In its characteristic form, we further emphasize key lexical markers such as Chiller 9, MAIN, power input sensor, etc that also enable a semantic-based evaluation.

Our dataset also enables end users to design new scenarios, such as: "Examine whether the year-over-year increase in corrective maintenance for CWC04009 warrants shifting resources from annual repairs to multi-year replacement planning." Existing scenarios (IDs 407–413) support strategic work-order management tasks, including trend analysis, bundling, and probability forecasting. Overall, the benchmark covers analytical reasoning (e.g., coding, model fine-tuning), context-aware decision-making, and language-based generalization.

### 4.3 DOMAIN SPECIFIC SINGLE AGENT AND MULTI-AGENT IMPLEMENTATION

**AssetOpsBench** includes four domain-specific AI agents: IoT, TSFM, WO, and FMSR. To illustrate tool-level complexity, we highlight two representative agents (Figure 3(b)): the TSFM agent, which uses a pretrained time-series foundation model from Hugging Face, and the FMSR agent, which leverages an LLM to generate failure-mode-to-sensor mappings via the get_mapping function. In total, the platform comprises over 15 tools across these agents, each with domain-specific guidance, making them unique in industrial settings. Three agents (TSFM, IoT, FMSR) use ReAct Yao et al. (2023), while the WO agent uses CodeReAct (Wang et al., 2024); alternative strategies such as RAFA (Liu et al., 2023) are also supported.

Given this mix of text- and code-based agents, a global coordinator, the **AssetOps Agent**, facilitates collaboration, operating under either an **Agent-As-Tool** paradigm or a **Plan-Execute** strategy. The components used to build these paradigm are widely adopted in modern open-source toolkits (Marreed et al., 2025; LangChain, 2025b; NVIDIA, 2025). In **Agent-As-Tool**, each agent is registered as a tool within a meta- or supervisor agent instantiated using ReAct, emulating layered decision-making in hierarchical organizations. In **Plan-Execute**, a **Planner** and **Reviewer** generate a plan as a directed acyclic graph (DAG), executed by an **Orchestrator** with a memory module that stores and transfers information between agents. This strategy adapts ReWoo (Xu et al., 2024a) with an additional review component inspired by (Li et al., 2025). We packaged the datasets, scenarios, domain-specific agents, and orchestration strategies into a dockerized environment.

## 5 EXPERIMENTS AND LEADERBOARD

To evaluate orchestration techniques across varying LLM sizes and agent-specific preferences, we adopt a **rubric-based** assessment LangChain (2025b); Wen et al. (2024); Wang et al. (2025); Andrews et al. (2025) complemented by a **reference-scoring** mechanism Yao et al. (2024); Wen et al. (2024); Cemri et al. (2025).

**LLM-As-Judge Scoring**. Each scenario is paired with a *characteristic form*, a structured specification defining both the expected final output and the intermediate reasoning or procedural steps required to achieve it. This form serves as the **soft ground truth** for evaluating agent behavior and supports rubric-based scoring with LLMs acting as judges. The evaluation rubric uses three qualitative metrics derived from experimental observations and common-sense principles. We define the **Evaluation Agent** as a scoring function that maps the original task query ($\mathcal{Q}$), the agent's trajectory output ($\mathcal{T}$, including intermediate reasoning and final output), and the characteristic form ($\mathcal{C}$, the ground-truth specification) to a set of scores ($y_1, y_2, y_3$). These scalar scores $(y_1, y_2, y_3) \in [0,1]^3$ correspond to **Task Completeness** ($y_1$: are all required steps completed?), **Data Retrieval Accuracy** ($y_2$: was the correct data retrieved and used?), and **Result Verification** ($y_3$: is the final result logically and factually correct?).

**Reference-Based Scoring.** For each scenario, we construct a structured ground truth inspired by Yao et al. (2024); Shen et al. (2024), where each entry captures the task workflow through planning_steps (high-level intended actions), execution_steps (concrete actions with corresponding inputs and outputs), and execution_links (dependencies between execution_steps). This representation encodes both the logical structure and the expected outcomes. We assess an agent's **task decomposition** ability by comparing the planning_steps with either the thinking traces in the agent's trajectory (for Agent-as-Tool) or the DAG produced by Plan-Execute. Since agents communicate in natural language, a weighted score is employed to align action descriptions and their inputs, thereby quantifying **task execution** performance.

**Experimental Setting.** To quantify agent effectiveness in scenario evaluations, we adopt the **Pass**$^k$ metric. Unlike the widely used Pass@k, which measures the probability that at least one of $k$ independent attempts succeeds, Pass$^k$ estimates the probability that an agent succeeds on *all $k$* attempts—a stricter criterion that better reflects the reliability requirements of industrial environments, where retries are often impractical and consistent behavior is essential for production deployment (LangChain, 2025a; Yao et al., 2024). In our benchmark, we report **Pass**$^1$ by default, as agents are executed once per task instance. The evaluation agent used for LLM-As-Judge scoring is run five times to derive stable performance estimates. Agents within the AgentOps framework

operate with a sampling temperature of 0, while the evaluation agent uses a temperature of 0.3, and all reported results follow this configuration.

## 5.1 ASSETOPSBENCH LEADERBOARD

**Models.** We conducted a series of benchmark experiments to evaluate a diverse set of language models, including closed-source models (e.g., `gpt-4.1`), frontier open-source models (e.g., `llama-4-maverick`, `llama-4-scout`, `mistral-large`, `llama-3-405b`), and medium-to-small open-source models (e.g., `llama-3-70b`, `granite-3-8b`). We have evaluated two different multi-agent strategies: *Agent-As-Tool* and *Plan-Execute* and also compared them with single agent.

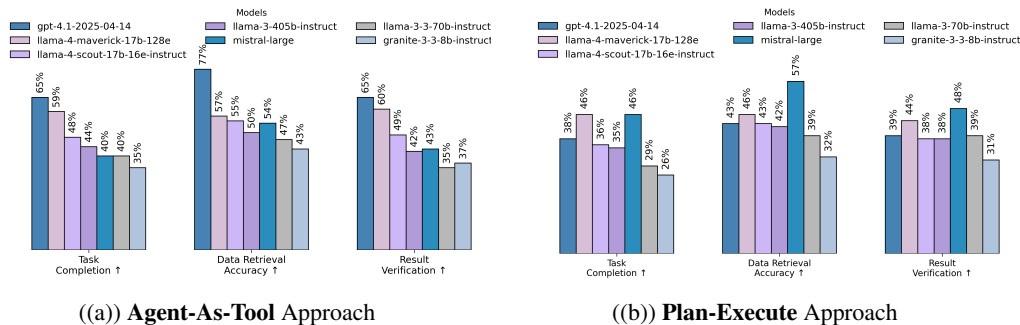

((a)) **Agent-As-Tool** Approach          ((b)) **Plan-Execute** Approach

Figure 4: Approach-wise Performance Evaluation. The order is based on the task completion rate.

**Agent-As-Tool vs Plan-Execute Approach.** Figure 4 shows the combined performance of both approaches using the rubric method. Overall, the Agent-As-Tool approach, as illustrated in Figure 4(a), demonstrates that `gpt-4.1` leads across nearly all metrics. `llama-4-maverick` also performs competitively, particularly in result verification (60%) and clarity (78%). Also, Data retrieval accuracy tend to higher than the task completion, yet another indirect validation of Evaluation Judge. But wait, `gpt-4.1` did not maintain its leadership position in `Plan-Execute` Approach, infact it see a largest drop in performance across all model. `mistral-large` and `llama-4-maverick` are top pick models for `Plan-Execute` strategy. Given that `llama-4-maverick` demonstrates balanced performance across both strategies, we select it as the default model for all ablation studies.

**Plan-Execute Approach Analysis.** We conducted a deep-dive analysis of the Plan-Execute approach to understand its relatively poor performance. First, we examined the length of the planning steps and observed that larger models tend to generate shorter plans in the Plan-Execute approach (typically 2–3 steps) compared to the Agent-As-Tool strategy, which generally requires 5–6 steps. Given that Agent-As-Tool performs better and uses longer plans, this suggests a known limitation of the Plan-Execute approach: reduced flexibility in handling unexpected failures or incorporating new information that may require plan revision (Li et al., 2025; NVIDIA, 2025). Next, we obtained the reference-based score of `gpt-4.1`, which is a `rouge1` of 0.354 and `rougeL` of 0.289 on the task decomposition aspect. This score is substantially lower than the top-performing `mistral-large` (`rouge1` 0.420, `rougeL` 0.343), indicating that, despite strong reasoning capabilities, `gpt-4.1` generates outputs that are less lexically aligned with the reference ground truth trajectories. And such behaviors may confuse down-stream agent in generating solution.

**Small Language Models Analysis.** Within the Agent-As-Tool evaluation, models such as `granite-3-8b` and `llama-3-3-70b` show weaker overall performance, yet they reveal clear areas of specialization as shown in Figure 5. Both models perform strongly on structured sensing and diagnostic tasks: for example, `granite-3-3-8b-instruct` achieves 15/20 on IoT, 18/22 on FMSR, and 19/23 on TSFM, while `llama-3-3-70b-instruct` reaches 12/20, 18/22, and 20/23 on the same categories. However, they struggle substantially on Work Order tasks, with scores of only 2/36 and 7/36, indicating that procedural, multi-step coordination remains difficult even under the Agent-As-Tool mechanism. This underscores a key insight : industrial deployments may benefit most from **hybrid LLM–SLM agent architectures**, where strong specialists handle sensing and diagnostics while more capable generalist models manage planning, coordination, and end-to-end reasoning (Belcak et al., 2025).

**Human Validation.** To assess the reliability of using LLMs as automatic evaluators for benchmarking tasks, we compare model-generated judgments against human annotations on a sample of 40 tasks. Each task is evaluated along three dimensions by four domain experts, all operating under the same information constraints as the LLMs. Before selecting a default evaluator, we compared several candidate judge models, including `gpt-4.1`. In this comparison, `gpt-4.1` showed only moderate alignment with expert assessments, achieving 69% accuracy and Cohen's $\kappa$ of 0.44. In contrast, `llama-4-maverick` provided substantially stronger agreement with human judgments and was therefore selected as the default judge model for the main analysis. Across experts, inter-rater reliability scores indicate substantial agreement on key evaluation dimensions, with *Data Retrieval Accuracy* exhibiting

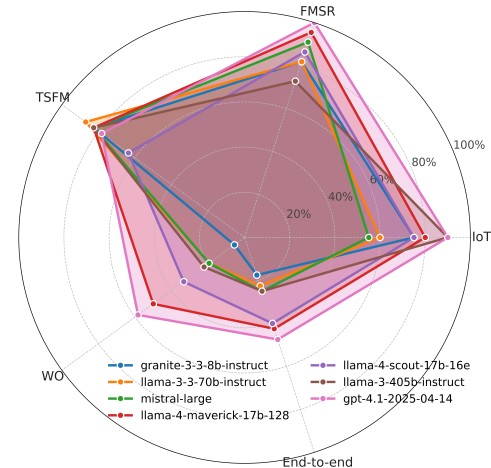

Figure 5: Agent Level Task Accomplishment with respect to Agent-As-Tool Approach

the strongest consistency (Cohen's $\kappa = 0.79$, 90.48% accuracy). *Task Completion* ($\kappa = 0.62$) and *Generalized Result Verification* ($\kappa = 0.71$) also show high alignment among evaluators.

**Ablation Study.** We study the effect of adding additional distractor agents to the system and removing guidance (i.e., in-context examples). The ablation experiments are conducted using the **Agent-As-Tool** method with `llama-4-maverick` as the default LLM. Injecting 10 out-of-domain distractors (e.g., `SREAgent`, `EchoAgent`) into 99 single-agent scenarios unexpectedly *improved* task completion accuracy (from 44 to 46), suggesting that distractors may induce more deliberate reasoning in LLMs. Similar effects have been reported in prior parallel work (Fu-Hinthorn, 2025). Extending this experiment across our full model portfolio revealed consistent, though modest, gains within the Llama family (particularly `llama-3-70b` and `llama-3-405b`), while other model families showed slight performance reductions or no improvement. In contrast, removing all in-context examples for 65 single-agent tasks (IoT+FMSR+TSFM) caused performance to collapse (from 80% to 34% for `gpt-4.1` and from 60% to 3% for `granite-3-8b`).

**Baseline using Single-Agent.** Instead of using four domain-specific sub-agents and an orchestration agent, we build a tool-calling ReAct agent with a single prompt as a baseline, giving it access to tools and in-context examples from all agents. In doing so, we increase the complexity of the problem, as it must handle many tools as well as an expanded context. We run a default LLM, `llama-4-maverick`, on all 141 scenarios. As a single-agent baseline, it achieves task completion of 26.95%, data retrieval accuracy of 34.04%, and generalized result verification of 28.37%. Under the Agent-As-Tool setup, the same model achieves roughly two-fold improvements (See Figure 4(a)).

### 5.2 Error Analysis via Agent Trajectories and Emerging Failure Modes

Trajectory analysis is critical for detecting agent mistakes, but becomes more challenging in multi-agent settings. We collected approximately 881 trajectories across different runs of models for Agent-As-Tool strategy. These trajectories were leveraged for further error analysis on two aspects: (a) tool-related errors and (b) agent failure modes.

**Failure Analysis on Tool Use.** Each agent step in a trajectory is logged as a structured JSON record capturing the *action type* and *execution state*. At the sub-agent level, we distinguish between **Tool-oriented actions**, which invoke predefined functions with well-defined inputs and outputs, and **CodeReAct-oriented actions**, where agents dynamically generate and execute Python code. Our analysis shows that Tool-oriented actions achieve higher valid-execution rates, whereas CodeReAct-oriented actions incur more runtime failures due to the variability of the generated code. Tool-oriented failures are concentrated in a small number of tools, including `jsonreader`, `tsfm_integrated_tsad`, and `Read Sensors From File`, highlighting challenges related to input validation and hallucinated parameter passing.

**Emerging Failure Modes Discovery.** Now we investigate trajectories from a semantic perspective. Recent work (Cemri et al., 2025) defines 14 failure modes for agent trajectories. Table 2 shows the distribution of failure mode on our 881 trajectories across this taxonomy. We found that **system design** is the most common source of failures. This taxonomy provides guidance for improving agent development. For instance, since the "Fail to Ask for Clarification" mode occurs around 10% of the time, we introduced a feature in the Agent-As-Tool strategy that allows sub-agents to ask the parent agent questions at any point during execution. We reran the entire benchmark on the default LLM, and this change led to significant performance improvements for `llama-4-maverick`, increasing task completion from 59% to 66%, surpassing `gpt-4.1`. To capture failure mode behaviors beyond this taxonomy, we allowed self-discovery of up to two **novel failure modes** per trace, revealing *emergent and compound failures* not covered by existing classifications. Common emergent failures include **Overstatement of Task Completion** (122 cases, 23.8%), **Extraneous or Ambiguous Output Formatting** (110 cases, 21.4%), and **Ineffective Error Recovery** (160 cases).

Table 2: Distribution of Failure Subcategories Across Stages of Execution

| Failure Subcategory | Stage & % |
|---|---|
| **System Design (Total 37.38%)** | |
| Disobey Task Spec. | Pre: 13.87% |
| Disobey Role Spec. | Pre: 0.11% |
| Step Repetition | Exec.: 16.41% |
| Loss of Conversation | Pre: 0.00% |
| Unaware of Termination | Post: 6.99% |
| **Agent Coordination (Total 27.52%)** | |
| Conversation Reset | Execution: 0.00% |
| Fail to Ask for Clarification | Execution: **10.22%** |
| Task Derailment | Execution: 4.34% |
| Information Withholding | Execution: 2.22% |
| Ignored Agent's Input | Execution: 2.06% |
| Action Mismatch | Execution: 8.68% |
| **Task Verification (Total 35.10%)** | |
| Premature Termination | Pre: 3.92% |
| No or Incomplete Veri. | Execution: 15.56% |
| Incorrect Verification | Execution: 15.62% |

### 5.3 GENERALIZATION ACROSS INDUSTRIAL DOMAINS

With the help of experts and the product team, we prepared an additional 162 scenarios across four datasets to evaluate generalization: Metro Train MetroPT-3 (15 scenarios) for compressor faults, UCI Hydraulic System (17 scenarios) for hydraulic component faults, Asset Health internal dataset (42 scenarios) based on work orders, and FailureSensorQA (88 scenarios) using ISO-standardized documentation for sensor-to-failure mapping. Table 3 presents one representative scenario from each dataset along with the peformance `llama-4-maverick`. Among all the datasets, scenarios of MetroPT-3 are difficult as we observed poor performance (task completion rate = 26.7%).

Table 3: Representative scenario from each dataset with LLaMA-4 Maverick performance.

| Dataset | Representative Scenario with LLaMA-4 Maverick Performance |
|---|---|
| MetroPT-3 | Consider asset `mp_1`. After maintenance on May 30, 2020, how has the compressor's condition evolved from May 31 to June 6, and are further repairs or monitoring needed? **Performance:** Task Completion 26.7%, Data Retrieval Accuracy 20.0%, Generalized Verification 40.0% |
| Hydraulic System | For asset `hp_1`, can severe internal pump leakage on 2024-01-31 be detected using sensor data from the preceding 100 days? **Performance:** Task Completion 88.2%, Data Retrieval Accuracy 100.0%, Generalized Verification 88.2% |
| Asset Health | Analyze the provided `Air Handling Unit_615152AC` work orders and asset details to determine the expected system condition. **Performance:** Task Completion 100.0%, Data Retrieval Accuracy 100.0%, Generalized Verification 100.0% |
| FailureSensorQA | For an aero gas turbine, list all failure modes that can be detected or indicated by abnormal readings from vibration, speed, or fuel flow sensors. **Performance:** Task Completion 67.0%, Data Retrieval Accuracy 71.6%, Generalized Verification 56.8% |

## 6 CONCLUSION

This paper presents a formalized framework for AI agents in industrial assets, encompassing a comprehensive and diverse set of scenarios derived from multiple data sources, a taxonomy, and a standardized evaluation methodology. The Agent-As-Tool paradigm offers a promising approach for orchestrating multi-agent interactions. In future work, we plan to introduce realistic environment constraints, such as compute limitations and API usage costs, to innovate novel algorithms.

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

APPENDIX OVERVIEW

In the Appendix, we discuss several topics that complement the main paper and provide additional technical detail to ensure clarity and reproducibility. These sections elaborate on our agentic system formulation, dataset design choices, hierarchical structuring of AssetOpsBench, and further empirical analyses.

- Agentic System Definition
- AssetOpsBench: Environment, Hierarchy and Domain Specific Agents
- Datasets Utilized in AssetOpsBench
- AssetOpsBench Scenarios
- Ground Truth Preparation for Reference-based Evaluation
- Additional Benchmark Experiments
- Generality: New Datasets and Scenarios
- Emerging Failure Mode Discovery and Agent Development

# A  AGENTIC SYSTEM DEFINITION

This section provides a **generic** detailed exposition of the content introduced in Section 3. In particular, we focus on the mathematical formulation of the agent architecture, followed by a brief overview of the proposed framework. The goal is to formalize the agent's operational components and offer foundational context for readers interested in the underlying design principles. We also discussed detailed design of two approaches for multi-agent system development: "Agent-As-Tool" and "Plan-Execute".

## A.1  AGENT-ORIENTED TASK AUTOMATION PROBLEM - AOP

We formalize the Agent-Oriented Problem (AOP) as a tuple:

$$\text{AOP} = \langle \mathcal{A}, \mathcal{T}, \Pi, M, O \rangle$$

where each component defines a core capability of a modular, agent-based reasoning and action system:

- $\mathcal{A} = \{A_1, A_2, \ldots, A_n\}$ denotes the set of available agents. Each agent $A_i$ is characterized by its reasoning capabilities, task specialization, internal memory, and communication interfaces, enabling autonomous or cooperative execution of assigned subtasks.

- $\mathcal{T} = \{\tau_1, \tau_2, \ldots, \tau_k\}$ is the set of tasks. Each task $\tau$ is described by a triple $\langle g, \mathcal{M}, C \rangle$, where $g$ denotes the task goal (e.g., fault detection or maintenance planning), $\mathcal{M}$ specifies the required input modalities (e.g., time-series telemetry, FMEA documents, structured metadata), and $C$ captures any domain-specific or operational constraints (e.g., time windows, asset type, or safety requirements).

- $\Pi$ is the hierarchical plan space. A plan $\pi \in \Pi$ is an ordered sequence of task-agent assignments:

$$\pi = [\langle \tau_1, A_i \rangle, \langle \tau_2, A_j \rangle, \ldots]$$

where each subtask is delegated to an appropriate agent for execution, potentially with dependencies among steps.

- $M$ denotes the memory system, consisting of both agent-local and shared global components. It is modeled as a dynamic key-value store $M = \{(k_i, v_i)\}_{i=1}^m$, supporting context persistence, lookup, and updates throughout the planning and execution process.

- $O$ represents the output space. Each output $o \in O$ is the structured or unstructured result of executing a plan. Outputs may include diagnostics, action recommendations, summaries, or control triggers, depending on the task and domain.

## A.2 BASE AGENT: REACT

AssetOpsBench uses the ReAct framework Yao et al. (2023) in an end-to-end agent design that integrates a Review Agent to verify the final answer. Figure 6 illustrates the full architecture. The ReAct agent executes a **Think-Act-Observe** loop, solving tasks iteratively while detecting and recovering from repetitive or ineffective actions. The Review agent verifies whether the ReAct agent has successfully completed the task, ensuring the quality of the output. Subsequent sections present the architecture in detail, highlighting the distinction between two architectural paradigms: **Agent-As-Tool** (See SectionA.3) and **Plan-Execute** (See SectionA.4). Note that, we can replace ReAct by any other agent development methdology such as Reflect, RAFA, etc.

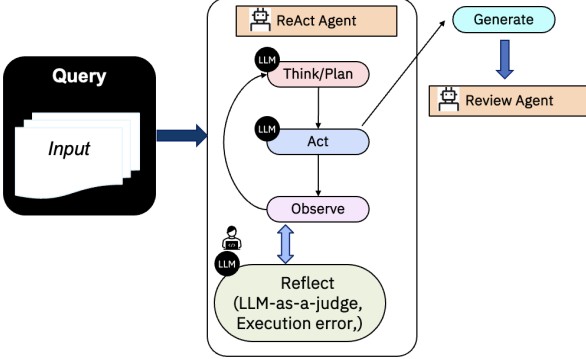

Figure 6: ReAct used to build individual agent

## A.3 AGENT-AS-TOOL

For the **Agent-As-Tool** paradigm as shown in Figure 7, we implemented the following components:

- A standard ReAct (Think–Act–Observe) agent loop using open source framework. In the initial setup, the *number of reflections* was set to one—effectively disabling reflection.

- A curated list of tools, the majority of which are stub interfaces that delegate functionality to specialized sub-agents. The only standalone utility tool in this set was the JSONReader, which reads a JSON object from a file and returns its contents as the tool's direct response.

Agent as Tool

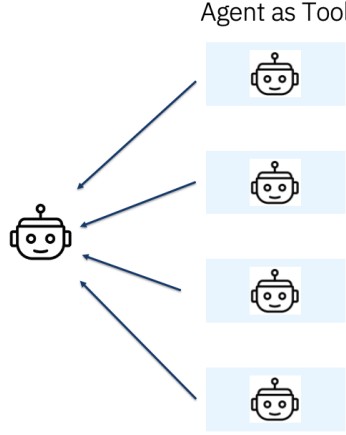

All agents are registered as tools

Figure 7: Agent-As-Tool

The sub-agent stubs were intentionally designed to be minimal. Each stub accepted a single input parameter: a string called `request` and returned a structured JSON output. The output JSON object included the following fields:

- `answer` – the primary answer returned by the sub-agent, represented as a plain string.
- `review` – a nested JSON object capturing a review of the response, typically including fields such as `status`, `reasoning`, and `suggestions`.
- `summary` – a brief description of the JSON object's structure and semantics, useful for interpretability or chaining with downstream tools.

The ReAct agent was initialized with a standard prompt that includes:

- **Examples for In-Context Learning** – A small number of sample interactions for each sub-agent were provided to guide behavior. These examples followed the standard ReAct format of Think–Act–Observe, illustrating how to invoke tools and interpret their responses. A representative example is shown below:

- **Tool Demonstrations** – These sample calls were concatenated to form a comprehensive set of demonstrations for all tools available to the agent, effectively seeding it with usage patterns.

The sample calls for all the tools are concatenated to form the examples.

- question - the question input to ReAct
- tool names - the list of sub-agent tool names (plus JSONReader)
- tool descriptions - descriptions of the sub-agents

```
Question: download asset history for CU02004 at SiteX
from 2016-07-14T20:30:00-04:00 to 2016-07-14T23:30:00-04:00
for  CHILLED WATER LEAVING TEMP  and
 CHILLED WATER RETURN TEMP

Action 1: IoTAgent
Action Input 1: request=download asset history for CU02004
at SiteX from 2016-07-14T20:30:00-04:00 to
2016-07-14T23:30:00-04:00 for  CHILLED WATER LEAVING TEMP
and  CHILLED WATER RETURN TEMP

Observation 1: {
   site_name :  SiteX ,
   assetnum :  CU02004 ,
   total_observations : 25,
   start : 2025-03-26T00:00:00.000000+00:00,
   final : 2025-04-02T00:00:00.000000+00:00,
   file_path :  /var/folders/fz/.../cbmdir/c328516a-643f-40e6-8701-
      ↪ e875b1985c38.json ,
   message :   found 25 observations. file_path contains a JSON array of
      ↪ Observation data
}
```

Listing 1: Example of Trajectory using ReAct Agent for IoTAgent

**Execution Framework.** The ReAct engine is reinitialized for each question and executed until either (a) successful completion, as determined by the Review component using an LLM-as-judge or (b) a maximum of ten iterations. The framework iterates through a list of models (e.g., `mistralai/mistral-large`) and a corresponding list of utterances to execute for each model. The system supports retries for failed executions. After each ReAct run, the complete trajectory and associated evaluation metrics are stored. We have provided a sample (partial) trajectory trace in

Listing 1, which show how patent agent call one of the tool (in this case IoTAgent) and receive a response. The recorded metrics include:

- **Question:** the input query being processed

- **Total execution time:** duration of the entire ReAct loop

- **Number of ReAct steps:** count of action-observation cycles

- **Review status:** success or failure determined by the LLM-based reviewer

Listing 2 outlines how the FMSR agent packages its reasoning output into a structured message for downstream agents or evaluators. The custom_json function formats the response to include the final answer, a peer review section (comprising status, reasoning, and suggestions), and a reflection field. Additionally, a natural language message is synthesized to summarize the execution result, enhancing transparency and interpretability in multi-agent settings. This output acts as a compact yet comprehensive communication protocol for reasoning agents collaborating in a complex task pipeline.

```python
def custom_json(obj):
    if isinstance(obj, FMSRResponse):
        return {
            answer : obj.answer,
            review : {
                status : obj.review[ status ],
                reasoning : obj.review[ reasoning ],
                suggestions : obj.review[ suggestions ],
            },
            reflection : obj.reflection,
            message : (
                I am FMSR Agent, and I have completed my task.
                f The status of my execution is '{obj.review['status']}'.
                    ↪
                f I also received a review from the reflection agent;
                f suggestions are included in the review field for
                    ↪ further insights.
            ),
        }
    raise TypeError(f Cannot serialize object of type {type(obj)} )
```

Listing 2: Formatted response message from FMSRAgent

### A.4 PLAN-EXECUTE

**Plan-Execute.** *Plan-Execute* is a widely used architectural paradigm for multi-agent systems. Figure 8 depicts the implementation adopted in our work. It is derived from specialized multi-agent system Marreed et al. (2025). The process initiates when a user submits a query, which is first processed by the **Planner**. The Planner decomposes the query into discrete, executable tasks. These tasks are then vetted by a **Reviewer** component to ensure quality, completeness, and relevance. Upon approval, the **Orchestrator** assigns the tasks to the most appropriate agents. Each agent independently executes its assigned task and returns a structured response. These responses are then aggregated by the **Summarization** module, which synthesizes them into a coherent final output that is returned to the user.

This architecture supports modularity, robustness, and interpretability across the task lifecycle. We have provided two system prompts where first prompt guides an AI to generate a structured step-by-step plan using external agents, while the second prompt instructs a reviewer agent to evaluate the plan's correctness and completeness in JSON format.

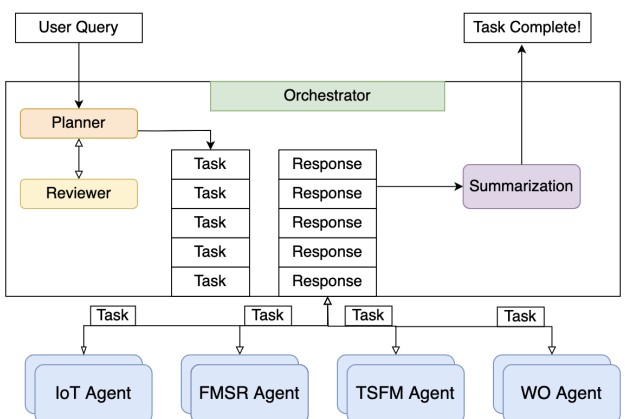

Figure 8: Plan-Execute Multi-Agent System

---

**System Prompt (Planning Agent)**

```
You are an AI assistant who makes step-by-step plan to solve a
↪  complicated problem under the help of external agents.
For each step, make one task followed by one agent-call.
Each step denoted by #S1, #S2, #S3 ... can be referred to in later
↪  steps as a dependency.

Each step must contain Task, Agent, Dependency and ExpectedOutput.
1. **Task**: A detailed description of what needs to be done in
↪  this step. It should include all necessary details and
↪  requirements.
2. **Agent**: The external agent to be used for solving this task.
↪  Agent needs to be selected from the available agents.
3. **Dependency**: A list of previous steps (denoted as `#S1`,
↪  `#S2`, etc.) that this step depends on. If no previous steps are
↪  required, use `None`.
4. **ExpectedOutput**: The anticipated result from the agent's
↪  execution.

## Output Format (Replace '<...>') ##

## Step 1
#Task1: <describe your task here>
#Agent1: <agent_name>
#Dependency1: None
#ExpectedOutput1: <describe the expected output of the call>

## Step 2
#Task2: <describe next task>
#Agent2: <agent_name>
#Dependency2: [<you can use #S1 and more to represent previous
↪  outputs as a dependency>]
#ExpectedOutput2: <describe the expected output of the call>

And so on...

Here are the available agents:
{agent_descriptions}

You are going to solve the following complicated problem:
{task.description}

Guidelines:
```

```
- Task should be something that can be solved by the agent.
- A plan usually contains less than 5 steps.
- Only output the generated plan.

Output (your generated plan):
```

**System Prompt (Review Agent)**

```
review_plan_system_prompt_template = """You are a critical reviewer
↪   tasked with evaluating the effectiveness and accuracy of a
↪   plan. Your goal is to determine whether the plan is valid or
↪   not given the context of the input question and agent
↪   expertise. A valid plan should:

1. **Ensure all necessary actions are addressed:**
   The plan must cover all required steps to successfully complete
   ↪   the task as specified in the question. Ensure that each
   ↪   action directly contributes to the task goal.
2. **Include appropriate dependencies between steps:**
   Actions should be logically ordered with clear dependencies.
   ↪   Each step must rely on the completion of the previous step
   ↪   to ensure a coherent and efficient workflow.
3. **Ensure no crucial steps are missed:**
   The plan must not overlook any essential actions required to
   ↪   solve the task. If any crucial steps are absent, the plan
   ↪   must be flagged as incomplete.
4. **Confirm all actions align with agent capabilities:**
   Each step in the plan must fall within the designated expertise
   ↪   of the agents involved. No action should require expertise
   ↪   or knowledge outside of the agent's specified capabilities.
   ↪   Any plan that violate this condition is an invalid plan.
5. **Strictly follow the task's question:**
   Carefully compare the provided question with the task. The plan
   ↪   should only include actions that directly relate to the
   ↪   question's explicit requirements, without introducing any
   ↪   unnecessary tasks or assumptions.
6. **Avoid Abstract task/step:**
   Ensure steps/tasks are grounded with respect to the data
   ↪   generated by previous steps or the question.

### Evaluation Criteria:
1. **Completeness:**
   - Verify that the system prompt leads to a plan that includes
   ↪   all necessary steps to accomplish the task.
   - Ensure the description of each step contains all the relevant
   ↪   information needed to execute the step, including any
   ↪   required parameters or inputs that are mentioned in the
   ↪   task's question.

2. **Relevance:**
   - Confirm that each step in the plan directly contributes to
   ↪   solving the task.
   - Eliminate any steps that do not serve a clear purpose in
   ↪   achieving the goal.

3. **Correctness:**
   - Ensure that all steps are logically consistent and ordered
   ↪   correctly.
   - Ensure that the dependencies between the steps are valid and
   ↪   follow a correct sequence.

4. **Expertise Alignment:**
```

```
      - Confirm that the steps in the plan are within the capabilities
      ↪  of the agent.
      - Validate that the agents used in each steps are among the
      ↪  available agents mentioned in the agents' expertise.

   5. **Efficiency:**
      - Make sure the plan doesn't introduce redundant actions.
      - Avoid unnecessary complexity in the plan.

   6. **Clarity:**
      - Ensure that the plan is easy to understand and logically
      ↪  structured.

   ---

   **Question:**
   {question}

   **Agents' Expertise:**
   {agent_expertise}

   **Plan:**
   {plan}

   ---

   ### Output Format:
   Your review must always be in JSON format. Do not include any
   ↪  additional formatting or Markdown in your response.

   ```json
   {{
       "status": "Valid | Invalid | Other",
       "reasoning": "A concise explanation for your evaluation. If a
       ↪  specific step is wrong, point it out directly.",
       "suggestions": "Actions or improvements for rectifying the plan
       ↪  if applicable."
   }}
   ```

   Output:
   """
```

## A.5   EXAMPLE DEMO

In the following Figures 9-11, we provide a few images to showcase working of Agent-As-Tool approach for a single end-to-end utterance.

```
Agent is Enabled with Reflexion
  Task Execution Status (Finished): False
-------------------------------------------------
Scratch Pad Content - At the Start of Running Agent

***************************************************
I am ReActXen Agent with ReAct
Input Question: find anomalies in the chiller 6 return Temperature (POKMAIN) in the first week of 2016
Debug Info (Step 1):
{
    "thought": "I need to request the chiller 6 return temperature data for the first week of 2016 from IoTAgent",
    "llm_output": " I need to request the chiller 6 return temperature data for the first week of 2016 from IoTAgen
t\nAction"
}
Thought 1: I need to request the chiller 6 return temperature data for the first week of 2016 from IoTAgent
Debug Info (Step 1):
```

Figure 9: Execution is Initiated with an input query.

```
,
Action 3: Finish
Action Input 3: The anomaly detection results of 'Chiller 6 Return Temperature' using data in /var/folders/fz/llh7g
pv96rv5lg6m_d6bk0gc0000gn/T/cbmdir/2e2eea99-946c-4a30-a688-8ddfa479ab62.json are stored in file ../output/tsad_outp
ut//tsad_conformal.csv  Final Answer: The anomaly detection results of 'Chiller 6 Return Temperature' using data in
/var/folders/fz/llh7gpv96rv5lg6m_d6bk0gc0000gn/T/cbmdir/2e2eea99-946c-4a30-a688-8ddfa479ab62.json are stored in fil
e ../output/tsad_output//tsad_conformal.csv  Question
Process is completed now
  Task Execution Status (Finished): True
  Review Agent Feedback: {'status': 'Accomplished', 'reasoning': "The agent successfully executed the task by perfo
rming time series anomaly detection on 'Chiller 6 Return Temperature' using the data from the specified file. The a
gent used the tsfm_integrated_tsad tool with the correct dataset path, timestamp column, and target columns. The ag
ent then stored the results in the specified output file. The response provides a clear and accurate description of
the task completion, including the location of the output file.", 'suggestions': 'None.'}
run minutes = 2.1432575666666667
```

Figure 10: The Final step of the execution

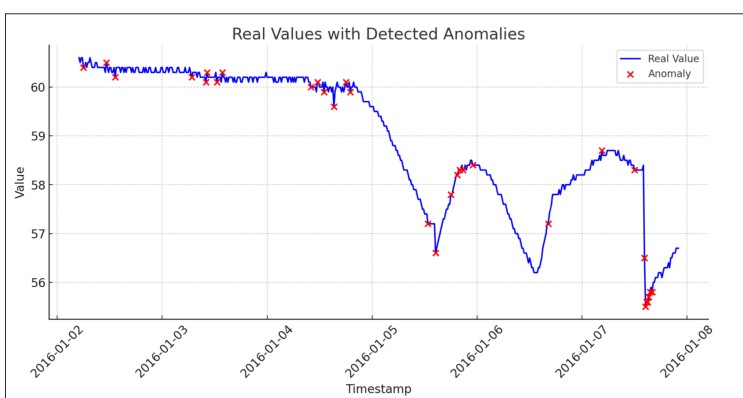

Figure 11: Anomaly Detection : Final Output

# B  ASSETOPSBENCH: ENVIRONMENT, HIERARCHY AND DOMAIN SPECIFIC AGENTS

This section presents the simulated environment for agentic evaluation, structured task taxonomy used in AssetOpsBench, which organizes benchmark scenarios based on key stages in the industrial asset lifecycle.

## B.1  SIMULATED ENVIRONMENT

Figure 12 provides a simulated docker environment for executing the task. The environment consists of domain specific agents (FMSR, IoT, TSFM, WO), and model inference APIs (LLM and TSFM), and access to telemetry data and Industry 4.0 data such as FMEA, Work Order, Alert and etc. The system also comes with implementation of orchestration such as Agent-As-Tool and Plan-Execute that are interfacing with user query.

First, we discuss the taxonomy that is used to support the creation of realistic, diverse, and role-specific evaluation tasks for intelligent agents operating in complex environments, as shown in Figure 13 for the tasks related to the industrial asset management. To illustrate how the structured task taxonomy guides agent development and evaluation, we highlight four representative agents: the IoT Agent, the FMSR Agent (Failure Mode Sensor Relations Agent), TSFM (Time Series Foundation Model) Agent, and the WO Agent (Work Order Agent). Among these, two agents : FMSR Agent and WO Agent are particularly useful for their domain specialization and integration depth within AssetOpsBench. Appendix B.3 presents the rationale for FMSR Agent, emphasizing its role in bridging raw telemetry with diagnostic reasoning through sensor–failure mapping. Appendix B.5 focuses on the WO Agent, which operationalizes maintenance planning and historical analysis by retrieving, filtering, and correlating work order records with asset conditions. Together, these examples demonstrate how high-level task categories such as failure mode alignment, anomaly response, and intervention prioritization are translated into grounded, data-driven agent behaviors. This alignment reinforces AssetOpsBench's emphasis on transparency, domain specialization, and end-to-end task automation.

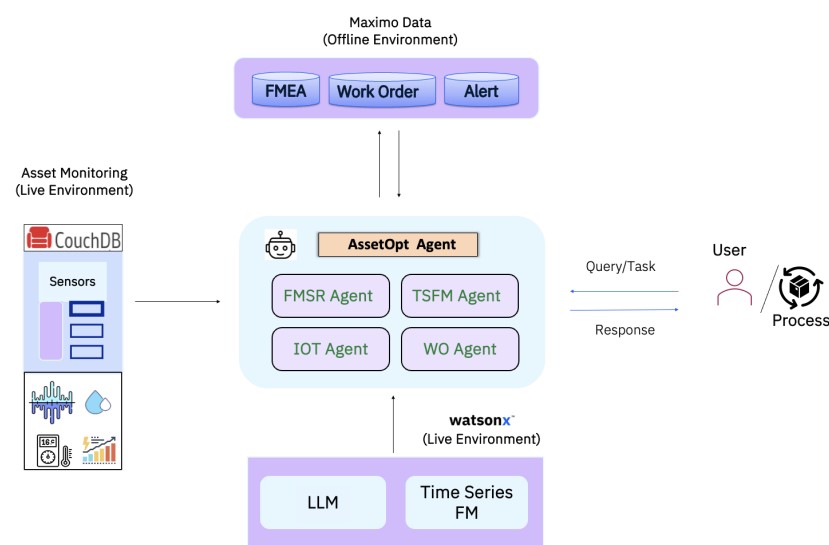

Figure 12: Simulated Environment for Open Source Contribution and Testing

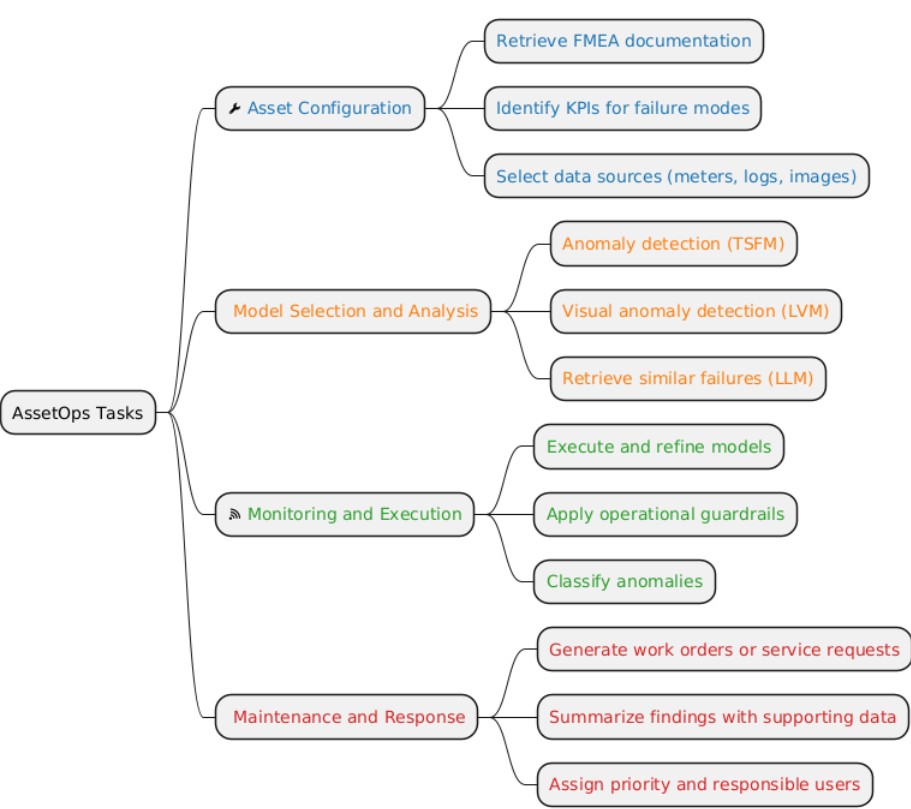

Figure 13: Representative Routine tasks in Asset Lifecycle Management.

### B.2 RATIONALE FOR IOT AGENT OVER APPLICATION

The IoT Agent plays a foundational role in supporting **Asset Configuration** tasks within the AssetOps framework, as illustrated in Figure 13. It enables structured access to real-time and historical

telemetry data, asset metadata, and site configurations. Specifically, it allows users to query available IoT-enabled sites, list all assets within a given site (e.g., MAIN facility), and retrieve detailed metadata for specific assets such as chillers and air handling units (AHUs). Additionally, it provides access to time-series sensor data such as power input, temperature, flow rate, and system tonnage across customizable time windows. These data queries form the backbone for monitoring tasks, model inputs, and analytics performed by downstream agents like TSFM Agent and WO Agent.

Although the IoT Agent does not perform anomaly detection or failure analysis directly, it is a critical enabler by delivering high-fidelity, time-aligned telemetry required for advanced applications (such as those using TSFM Agent). For example, users can retrieve the tonnage data for Chiller 6 during a specific week, download metadata for Chiller 9, or access sensor values recorded during a known operational event. These capabilities align with the early-phase needs of asset lifecycle management specifically selecting data sources and configuring metrics of interest ensuring all downstream decision-making is grounded in accurate, context-rich operational data. The agent's flexible query interface and knowledge and data retrieval support allow it to seamlessly integrate into automated pipelines for asset monitoring, diagnostics, and performance tracking.

### B.3  RATIONALE FOR FMSR AGENT OVER APPLICATION

The sensor–failure alignment generation (See Figure 14) is a critical component of the **AssetOps-Bench** benchmark, serving multiple roles in both dataset understanding and intelligent system design. Its inclusion is motivated by the following key factors:

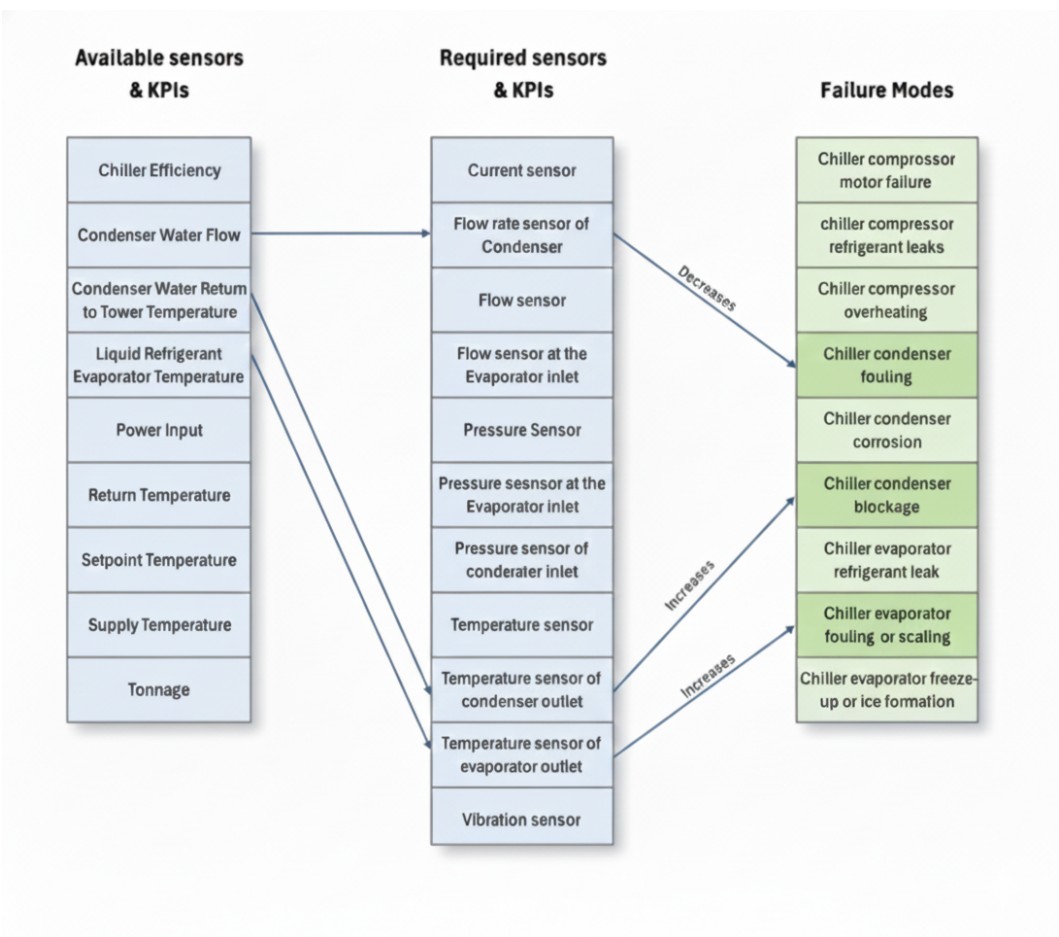

Figure 14: Mapping Example internally used by FMSR Agent

1. **Bridging Raw Data and Diagnostic Insight:** The table explicitly maps sensor variables to relevant failure modes, establishing a direct link between low-level telemetry and high-level maintenance reasoning. This supports tasks such as fault detection, root cause analysis, and feature selection for learning-based systems.

2. **Alignment with FMEA Methodology:** By structuring failure explanations according to the principles of Failure Modes and Effects Analysis (FMEA), the table offers a formalized, interpretable view of asset health. Each sensor's diagnostic role is contextualized through failure causes, effects, and detection implications.

3. **Supporting Explainability and Safety:** In industrial environments, operational decisions require transparency. The alignment table enhances system explainability by clarifying why a given signal is relevant, how it relates to equipment health, and what operational risks it may indicate.

4. **Improving Dataset Transparency:** The AssetOpsBench dataset includes a wide range of sensors across multiple devices. This table functions as a documentation layer that improves usability, reproducibility, and understanding for researchers and practitioners engaging with the benchmark.

5. **Guiding Model and Rule Development:** Whether designing rule-based systems, hybrid AI architectures, or physics-informed machine learning models, a well-defined mapping of sensors to failure mechanisms is foundational. It informs the construction of robust detection logic and contributes to generalizable reasoning strategies.

In sum, the sensor–failure alignment table plays a central role in transforming raw operational telemetry into structured, actionable insight. It provides the semantic grounding necessary for developing interpretable, reliable, and effective AI agents for real-world industrial maintenance tasks. Table 4 provides an extensive example for sensor-failure mode relation for a chiller system build using our SME inputs.

Table 4: Sensor Interpretation and Failure Mode Relevance in Chiller Systems - Illustrative

| Sensor | Explanation | Impact on Chiller Health / Failure Mode Relevance |
| --- | --- | --- |
| Condenser Leaving Temp | Temperature of water leaving the condenser | Indicates heat rejection efficiency; abnormal readings may signal fouling or reduced flow — potential *heat exchange failure*. |
| VFD Output Voltage | Voltage output from Variable Frequency Drive | Instability may affect fan/compressor operation — linked to *electrical drive failure* or *load imbalance*. |
| CHWSTSP in Free Mode | Chilled water setpoint during free cooling mode | Misconfiguration can lead to energy inefficiency — related to *control logic failure*. |
| Cycling Code | Indicates compressor cycling state | Frequent cycles may indicate *load mismatch*, *sensor error*, or *compressor stress*. |
| Ready Status | Indicates if chiller is in a ready state | Persistent unavailability may reflect *control override*, *interlock failure*, or *alarm lockout*. |
| Manual Start/Stop | Overrides for manual operation | May cause *unscheduled runtime* or *safety override* conditions. |
| Chilled Water Leaving Temp | Temperature leaving evaporator | Deviation may suggest *capacity loss* or *improper load conditions*. |
| Condenser Flow | Water flow through condenser loop | Low flow may cause *high pressure shutdown* or *heat rejection failure*. |
| VFD Input Power | Power input to VFD | Spikes may indicate *motor inefficiency*, *overload*, or *harmonic distortion*. |
| CNW Flow Hi Alarm SP | High flow setpoint for condenser loop | May indicate *bypass valve issues* or *overpumping*. |
| Watt/Ton | Cooling efficiency metric | Rising ratio suggests *energy inefficiency* or *component degradation*. |

| Sensor | Explanation | Impact on Chiller Health / Failure Mode Relevance |
|---|---|---|
| Chilled Water Flow | Water flow through evaporator | May point to *pump failure*, *valve issues*, or *airlocks*. |
| Motor Run Status | Compressor motor operational state | Discrepancies could signal *false starts*, *sensor error*, or *runtime misreporting*. |
| Vibration Point #1 SP | Vibration sensor setpoint (location #1) | May indicate *bearing failure*, *imbalance*, or *mechanical looseness*. |
| CHW Valve Position | Position of chilled water valve | Out-of-range position may imply *valve actuator fault* or *control misbehavior*. |
| CHW Differential Pressure (D/P) | Pressure drop across chilled water loop | Suggests *clogging*, *filter fouling*, or *flow resistance*. |
| CHW Flow Hi Alarm SP | Alarm setpoint for high CHW flow | Triggered by *pump overspeed*, *valve overshoot*, or *control issues*. |
| Condenser Return Temp | Water temperature returning to the condenser | Important for *thermal load calculation* and monitoring *efficiency*. |
| Average Amps | Average motor current | High current may indicate *overload*, *bearing drag*, or *electrical faults*. |
| CHW Valve Close Control | Control signal to close CHW valve | Improper function may cause *flow issues* or *unmet loads*. |
| CNW Differential Pressure (D/P) | Pressure drop in condenser loop | Indicates *scaling*, *fouling*, or *pump degradation*. |
| VFD Internal Ambient Temp | Internal temperature of VFD | High temps may trigger *thermal trips* or shorten *VFD lifespan*. |
| Freon Temp | Refrigerant temperature | Abnormal values may suggest *charge issues*, *expansion valve faults*, or *heat exchange failure*. |
| Compressor Oil Sump Temp | Oil sump temperature | High temperature may signal *bearing wear* or *insufficient cooling*. |
| Chilled Water Return Temp | Return water temp to evaporator | Used for *cooling load* and *delta-T analysis*. |
| Motor Run Status RPT | Reported motor run confirmation | Mismatch suggests *sensor/control error*. |
| VFD Inverter Link Current | Current through VFD inverter link | High current may indicate *overload* or *VFD stress*. |
| CHWSTSP in Part Mode | Setpoint in partial load mode | Improper configuration can cause *energy waste* or *load mismatch*. |
| VFD Phase A/B/C Current | Phase currents from VFD | Used to detect *imbalances*, *shorts*, or *phase loss*. |
| VFD Converter Heat Sink Temp | VFD heat sink temperature | Elevated temps reduce *component life* and can cause *failure*. |
| Compressor Oil Pressure | Oil pressure in compressor | Low pressure risks *lubrication failure* and *component damage*. |
| Failure (status flag) | Direct failure indicator | Used as ground truth label for fault evaluation. |
| VFD Setpoint | Speed or torque command | Affects *energy usage*, *response time*, and *cooling capacity*. |
| CHW Flow High Alarm | High flow warning flag | May indicate *system control faults* or *oversized flow components*. |
| VFD DC Bus Voltage | DC voltage level inside VFD | Instability can reflect *power quality issues*. |
| CNW Flow High Alarm | High condenser water flow warning | May reflect *valve misposition* or *energy inefficiency*. |
| CNW Flow Low Alarm SP | Low flow alarm threshold | Indicates *risk of overheating* or *shutdown due to poor heat rejection*. |
| Warning Code | Non-critical warning status | Helpful for *early diagnostics* or *trend detection*. |

| Sensor | | Explanation | Impact on Chiller Health / Failure Mode Relevance |
|---|---|---|---|
| Vibration #2/#3 SP | Points | Additional vibration set-points | Detect *imbalance*, *wear*, or *mechanical degradation*. |

### B.4 RATIONALE FOR TSFM AGENT OVER APPLICATION

The TSFM Agent is purpose-built to support critical tasks within the **AssetOps** workflow, as outlined in Figure 2(a). Within **Model Selection and Analysis**, TSFM Agent enables forecasting of key performance indicators (KPIs) using lightweight, pre-trained foundation models. Its adaptive anomaly detection framework, based on post-hoc conformal prediction, supports calibrated and interpretable anomaly scores, providing high utility for both **Monitoring and Execution** and **Maintenance and Response**.

Specifically, the TSFM Agent can execute and refine models, classify anomalies based on historical deviations, and support operational guardrails by simulating expected trends under normal conditions. In downstream applications, the agent's outputs can be used to summarize overall system health by tracking the frequency of anomalies across selected KPIs. These anomalies serve as a foundation for maintenance recommendations, enabling preventive and reactive work order generation. TSFM Agent facilitates real-time, data-driven decision-making throughout the asset lifecycle.

### B.5 RATIONALE FOR WO AGENT OVER APPLICATION

The WO Agent, a code based ReAct, in **AssetOpsBench** is designed to enable intelligent interaction with structured and unstructured maintenance records through a modular data model. It operates over a set of *Business Objects* (BOs) that represent work orders, alerts, anomalies, failure codes, and asset metadata. These BOs are categorized into five functional groups that collectively support the WO Agent's decision-making capabilities.

To reason over these BOs, the WO Agent is equipped with a collection of analytic functions that allow it to retrieve, interpret, and act upon historical and real-time data. The agent's capabilities are structured as follows:

1. **Historical Reasoning via Content Objects and Knowledge Extraction:** The WO Agent accesses raw maintenance data such as *WorkOrders*, *Events*, including Work orders, alerts, and anomaly Events. Knowledge extraction functions enable the agent to retrieve and filter this data by date, asset, and work order type, allowing targeted analysis and retrospective diagnostics.

2. **Standardized Interpretation with Meta/Profile Objects:** BOs like *ISO Failure Code*, *AlertRule*, and *Equipment* provide structured classification schemes. These allow the agent to categorize failures, apply semantic filters, and maintain compatibility with domain conventions—critical for aligning alerts and anomalies with actionable categories.

3. **Temporal and Causal Reasoning via Statistical Functions:** Leveraging relationship BOs such as *Alert-Rule Mapping* and *Anomaly Mapping*, the WO Agent applies statistical functions (e.g., Allen's Interval Algebra) to detect temporal patterns—such as when alerts consistently precede failures. It also detects repeated work order cycles, helping align maintenance with actual degradation patterns instead of fixed schedules.

4. **Predictive and Prescriptive Intelligence through Decision Support Functions:** Using the *WorkOrderRecommendation* BO, the agent forecasts future work orders, recommends maintenance based on alerts or KPI anomalies, and identifies opportunities for bundling related tasks. These decision support functions enable proactive scheduling and optimize resource use across the asset lifecycle.

5. **Persona-Aligned Interaction and Query Resolution:** The WO Agent interfaces naturally with domain personas. Maintenance engineers can explore past interventions for a given failure, while planners can query upcoming work order demands or seek opportunities to consolidate tasks. These capabilities are backed by modular functions that support flexible querying and planning logic.

In summary, the WO Agent is a hybrid reasoning and decision-support agent built atop structured business objects and analytic functions. It connects historical insight with predictive planning, enabling lifecycle-aware maintenance interventions grounded in transparent, data-driven logic.

## B.6 TOOLS USED BY AGENTS

In this section, we describe the development of over fifteen LangChain-based tools that form the backbone of our agent framework. We follow a standardized methodology for tool construction, and, with the exception of the WO agent, all agents operate through tool-calling APIs. Table 5 lists thirteen of these tools along with their names, descriptions, and parameters. For brevity, we omit some of the lower-level parameters associated with the time-series tool suite. In case of WO agent, which is a coding agent, we needed to build a generic business driven object, as given in Table 6.

Table 5: List of Available Tools and Their Parameters.

| Tool Name | Description | Parameters (Required Fields) |
|---|---|---|
| **Get Failure Modes** | Retrieves failure modes linked to a specific asset. | `asset_name`: name of the asset. |
| **Get Failure Mode and Sensor Relevancy Mapping** | Returns relevancy mapping between failure modes and sensors for downstream tasks. | `input_str`: string with asset name, failure modes, and sensors. |
| **Read Sensors From File** | Reads available sensors of an asset from a file and outputs sensor variable names. | `input_str`: sensor file path. |
| **sites** | Retrieves a list of available sites. | `v__args`: optional array (*default: null*). |
| **history** | Returns sensor values for an asset within a given time range. | `site_name`, `assetnum`, `start`, `final`. |
| **assets** | Lists all assets available at a given site. | `site_name`. |
| **sensors** | Lists all sensors for an asset at a given site. | `site_name`, `assetnum`. |
| **jsonreader** | Parses a JSON file and returns its content. | `file_name`. |
| **currentdatetime** | Returns current date and time as JSON. | `v__args`: optional array (*default: null*). |
| **aitasks** | Lists available AI tasks and their methods (`task_id`, `description`). | `v__args`: optional array (*default: null*). |
| **tsfmmodels** | Lists supported forecasting models (ID, checkpoint, description). | `v__args`: optional array (*default: null*). |
| **tsfm_forecasting** | Forecasts sensor or KPI variables using pretrained time-series models. | `dataset_path`, `model_checkpoint`, `timestamp_column`, `target_columns`. |
| **tsfm forecasting finetune** | Finetunes a pretrained forecasting model on new data. | `dataset_path`, `model_checkpoint`, `timestamp_column`, `target_columns`. |
| **tsfm integrated tsad** | Performs time-series anomaly detection using model predictions. | `dataset_path`, `timestamp_column`, `target_columns`. |

Table 6: WO Agent Summary of Business Objects, Source, Role, and Number of Records

| Business Object | Source | Role | Count |
|---|---|---|---|
| **Content Objects** | | | |
| WorkOrder | Work Order Manager | Tracks scheduled and unscheduled maintenance tasks, categorized as preventive or corrective. | 4392 |
| Event | Aggregated by Authors | Consolidates event logs for tracking and decision-making. | 6929 |
| Alert Events | IoT Repository | Logs real-time alerts triggered by IoT sensors based on predefined conditions. | 1995 |
| Anomaly Events | ML Generated | Detects KPI deviations using machine learning for predictive maintenance. | 542 |
| **Meta/Profile Objects** | | | |
| ISO Failure Code | Developed by Authors | Standardizes failure classification for structured maintenance analysis. | 137 |
| ISO Primary Failure_Code | Developed by Authors | Defines primary failure categories and links related secondary codes. | 68 |
| AlertRule | SME Provided | Specifies conditions for triggering alerts based on system behaviors. | 77 |
| Equipment | SME Provided | Represents industrial assets, including status and specifications. | 22 |
| **Relationship Causality Objects** | | | |
| Alert-Rule Mapping | Relationship Causality | Links alert rules to failure codes for automated diagnostics. | 46 |
| Anomaly Mapping | Relationship Causality | Associates anomalies with failure codes for predictive insights. | 12 |
| **Recommendation Objects** | | | |
| WorkOrder Recommendation | Recommendation | Suggests maintenance actions based on historical patterns. | N/A |

*Note:* The design and structure of the business objects and corresponding analysis in this section are valid for other industrial asset types, such as standby generators.

## C    DATASETS UTILIZED IN ASSETOPSBENCH

In this part, as extension of Section 4.1, we will zoom into the datasets utilized by the various agents of **AssetOpsBench** (More details of the roles of the agents in the asset lifetime management can be found at Appendix B.

### C.1    SENSOR TELEMETRY DATASET FOR IOT AGENT AND TSFM AGENT

Both IoT Agent and TSFM Agent (Figure 2(a)) leverage the **Sensor Telemetry Dataset**, which comprises sensor telemetry collected from Building Management Systems (BMS) and the SkySpark analytics platform. This dataset captures fifteen-minute interval operational data from industrial HVAC systems, specifically a fleet of chillers. Each chiller unit (e.g., Chiller 4, Chiller 14) is instrumented with a standardized suite of physical sensors that monitor key operational parameters in real-time.

A representative subset of these sensors is summarized in Table 7. These sensors record various kinematic, dynamic, thermodynamic, electrical, and operational metrics essential to assessing the performance and health of chiller systems. Measurements include water and refrigerant temperatures, power consumption, cooling capacity (tonnage), flow rates, and system setpoints. Addition-

ally, computed metrics such as chiller efficiency and load percentage serve as valuable real-time indicators of system performance.

Table 7: Representative Sensors in the **AssetOpsBench** Dataset

| Sensor Name | Description |
| --- | --- |
| Chiller Return Temperature | Temperature of water returning to the chiller |
| Supply Temperature | Temperature of water exiting the chiller |
| Power Input | Electrical power consumption |
| Tonnage | Heat extraction rate (cooling capacity) |
| Condenser Water Supply to Chiller Temperature | Temperature of water supplied to the condenser |
| Chiller Efficiency | Instantaneous performance metric |
| Chiller % Loaded | Current load as a percentage of the maximum |
| Condenser Water Flow | Flow rate through the condenser |
| Liquid Refrigerant Evaporator Temperature | Temperature of refrigerant in the evaporator |
| Run Status | Binary indicator of whether the chiller is currently operating |
| Setpoint Temperature | Current setpoint for chiller operation |

Each sensor stream is accompanied by rich metadata, including sensor type, measurement units, physical location, and structured device tags that define device associations. The dataset captures realistic operational variability, encompassing noise, missing data, and seasonal patterns. As such, it provides a robust foundation for developing and benchmarking models that require temporal reasoning, fault detection, and decision-making under uncertainty.

As illustration, Figure 15 presents layered time series subplots for key chiller sensors over a selected snapshot period in June 2020 for Chiller 6. Each subplot corresponds to one sensor variable, enabling a clear view of temporal dynamics and inter-variable behavior. This figure provides insight into the operational profile of a single chiller unit during real-world usage.

The IoT Agent interacts with this telemetry data through structured utterances. By leveraging the standardized data provided by **AssetOpsBench**, the agent enables detailed, query-driven access to operational information across HVAC assets such as chillers and air handling units (AHUs) at IoT-enabled sites like the MAIN facility. Through these utterances, users can request both real-time and historical data, retrieve metadata, and download sensor readings for specific timeframes. This functionality supports knowledge and data queries, facilitating asset-level diagnostics, performance monitoring, and intelligent decision-making, even in noisy or incomplete data.

On the other hand, the TSFM Agent operates on sensor telemetry data that are either retrieved via the IoT Agent or accessed directly from the sensor repository to perform advanced time series analysis across HVAC systems. It supports a range of analytical tasks, including multivariate forecasting, and time series anomaly detection. At its core, the agent utilizes pre-trained time-series foundation models. For anomaly detection, the TSFM Agent applies a model-agnostic, post-hoc adaptive conformal method that requires no additional fine-tuning data, making it highly practical for real-world, resource-constrained deployments. By learning dynamic weighting strategies from prediction histories, it can detect distributional shifts and maintain calibrated, interpretable anomaly scores aligned with user-defined false alarm rates. Through structured utterances, users can invoke forecasting on specific variables (e.g., "Chiller 9 Condenser Water Flow"), fine-tune models with minimal data, or detect anomalies in historical trends, all with minimal configuration. This seamless integration of pre-trained models, adaptive analytics, and user-guided queries enables transparent, robust, and immediately deployable monitoring solutions tailored for critical industrial systems.

## C.2 FAILURE MODE DATASETS FOR FMSR AGENT

The failure mode datasets in **AssetOpsBench** are modeled using the principles of *Failure Modes and Effects Analysis* (FMEA), a structured framework used in reliability engineering to identify

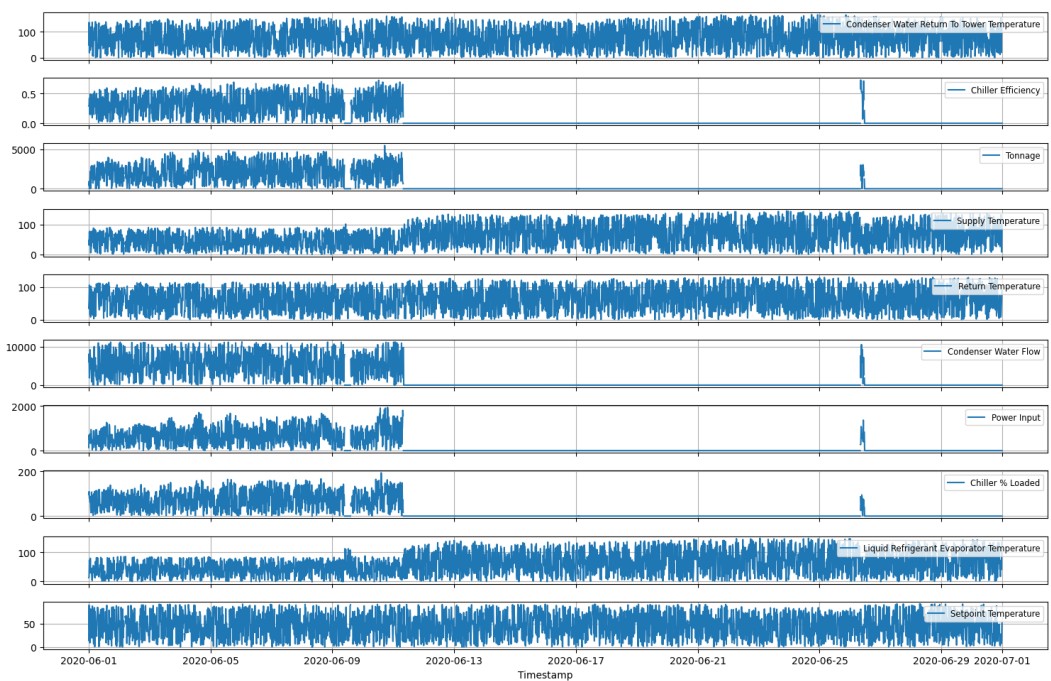

Figure 15: Snapshot of time series data from Chiller 6 for June 2020. Each subplot shows an individual sensor's trend over time.

failure risks, assess root causes and effects, and inform condition-based maintenance strategies. Each failure is defined by its mode, degradation mechanism, detection opportunity, and operational impact, enabling structured reasoning for both rule-based diagnostics and machine learning.

Failures in the dataset are annotated at the asset and subsystem levels, with a primary focus on centrifugal chillers. These failures reflect realistic degradation pathways and operational stressors derived from field experience. Each record in the failure model includes:

- **Failure Location and Component:** The subsystem or part where failure occurs, such as *bearings*, *gearboxes*, *impellers*, or *lubrication systems*.

- **Degradation Mechanism:** The underlying physical process driving the failure, including *wear*, *erosion*, *oil degradation*, *vibration-induced fatigue*, and *misalignment*.

- **Degradation Influences:** External or internal stressors such as *run time*, *two-phase process fluid*, *personnel error*, or *shock loading*.

- **Functional Failure Mode:** The resulting operational defect, such as *decreased oil pressure*, *audible noise*, *low head pressure*, or *capacity loss*.

- **Detection Opportunities:** Observable precursors or symptoms, including sensor readings (e.g., oil sampling, vibration signals), condition-based alarms, or inspection results.

- **Repair Time and Criticality:** Estimated downtime and classification of failure risk, supporting cost-based prioritization and scheduling.

- **Preventive Task Type:** Associated maintenance activity, such as *oil analysis*, *vibration analysis*, or visual inspection, tagged with effectiveness ratings and intervention intervals.

For example, *bearing wear* a recurring failure across chiller subsystems may arise from lubrication failure, misalignment, or fluid shock loading. This degradation is detectable via a combination of oil analysis and vibration monitoring, with failure symptoms including increased vibration, reduced oil pressure, and audible anomalies. Similarly, impeller erosion is linked to aging and two-phase fluid exposure, typically presenting as reduced capacity and lower head pressure.

Each maintenance task in the dataset is mapped to its detection mechanism and action type (e.g., condition monitoring vs. corrective repair), along with documentation on task content and recommended frequency. These structured records not only support early fault detection and diagnostics but also facilitate benchmarking of intelligent agents' reasoning over real-world degradation patterns and maintenance decisions. Failures are temporally aligned with telemetry, enabling the study of degradation trajectories and pre-failure conditions. This integrated design makes the dataset suitable for supervised learning, causal inference, and evaluation of digital twins or predictive maintenance agents under realistic operating uncertainty.

To utilize the failure modes and their association with the sensors, we design FMSR (Failure Mode Sensor Relations) to interpret failure mode datasets within the **AssetOpsBench** framework, leveraging structured FMEA (Failure Modes and Effects Analysis) principles to link sensor telemetry with degradation mechanisms and operational failures. Using annotated failure records for assets such as centrifugal chillers, the FMSR Agent builds knowledge graphs and reasoning models that connect specific failure modes like compressor overheating, evaporator fouling, or refrigerant valve failure to their underlying causes and detectable symptoms. These failure modes are mapped to available sensor measurements (e.g., supply temperature, power input, vibration, flow rate) to identify observable precursors. For example, compressor overheating may be monitored through trends in power input, chiller efficiency, and evaporator temperature, while condenser fouling can manifest in abnormal return temperatures and flow rate deviations. Through structured utterances, users can query which failure modes are associated with specific sensors, which are critical for detecting a given failure, or even construct machine learning recipes for predictive modeling such as anomaly models for chiller trips or excessive purging. The agent leverages this data to perform rule based diagnostics, support causal analysis, and assist in condition based maintenance planning. By aligning temporal sensor patterns with known failure signatures, the FMSR Agent enables explainable fault detection and root cause inference, ultimately enhancing reliability, maintainability, and transparency in HVAC operations.

### C.3 WORK ORDER DATASETS FOR WO AGENT

Table 6 provide the summary of datasets (as business objects) and the size for each dataset. Those work order datasets in **AssetOpsBench** provide a structured view of maintenance activity across industrial assets, encompassing both preventive and corrective interventions using work orders. Each work order is associated with rich contextual data including equipment metadata, failure classification codes (e.g., ISO Failure Code, ISO Primary Failure Code), event logs, sensor-triggered alerts, and machine-generated anomalies. These records are linked temporally and causally, allowing agents to reason about asset history, detect recurring failure patterns, and recommend actions based on past interventions.

The group of datasets distinguishes between core content objects (e.g., WorkOrders, Alerts, Events, Anomalies), metadata profiles, and relational structures that map alerts and anomalies to failure codes.

The individual event tables: *work orders* (Table 8), *alert events* (Table 9), and *anomaly events* (Table 10) captures different but complementary signals related to equipment condition and behavior. To enable integrated analysis and causal reasoning, these events are unified into a common *event table schema* (Table 12), allowing temporal alignment and cross-type relationship discovery between maintenance actions, system warnings, and performance anomalies.

In addition, to support the linkage of failure code over the events, we provide two mapping tables: one that connects alert rules to likely failure codes, and another that maps KPI-based anomalies to structured failure categories (Tables of 13 and 11). These mappings enable agents to infer probable root causes from real-time signals and integrate data-driven insights with expert failure taxonomies.

This help us to develop WO agent to support grounded evaluation of diagnostic reasoning, task generation, and repair recommendation. More particularly, the WO agent analyze historical work orders to identify repeated maintenance issues and improve task scheduling. It processed historical work order, alerts (from IoT Agent) and anomalies (from TSFM agent) event, linking them to failure codes to support predictive maintenance recommendations. In the potential industrial applications, WO agent can complete to tasks of automating the interpretation of maintenance data, predicting future work orders, and bundling related tasks to reduce operational downtime.

Table 8: Work Order Event Schema Definition

| Field Name | Type | Description |
| --- | --- | --- |
| wo_id | String | Unique identifier for the work order. Example: `"L247402"` |
| wo_description | String | Description of the work being done. Example: `"CHILLER COMP OIL ANALYSIS"` |
| collection | String | Broad group or system the work relates to. Example: `"compressor"` |
| components | String | Specific part or component being serviced. Example: `"compressor"` |
| primary_code | String | Code representing the main type of work. Example: `"MT010"` |
| primary_code_desc. | String | Description of the primary work code. Example: `"Oil Analysis"` |
| secondary_code | String | Sub-code under the primary category. Example: `"MT010b"` |
| secondary_code_desc | String | Description of the secondary code. Example: `"Routine Oil Analysis"` |
| equipment_id | String | Unique ID of the equipment. Example: `"CU02013"` |
| equipment_name | String | Human-readable name of the equipment. Example: `"Chiller 13"` |
| preventive | Boolean | Indicates if this is preventive maintenance. Example: `TRUE` |
| work_priority | Integer | Priority level of the work (e.g., 1–5). Example: `5` |
| actual_finish | DateTime | Date and time when the work was completed. Example: `"4/6/16 14:00"` |
| duration | Duration | Total job time. Format: `HH:MM`. Example: `"0:00"` |
| actual_labor_hours | Duration | Actual labor time spent. Format: `HH:MM`. Example: `"0:00"` |

Table 9: Alert Event Schema Definition

| Field Name | Type | Description |
| --- | --- | --- |
| equipment_id | String | Unique identifier for the equipment that triggered the alert. Example: `"CWC04701"` |
| equipment_name | String | Human-readable name of the equipment. Example: `"Chiller 1"` |
| rule_id | String | Identifier for the rule or condition that triggered the alert. Example: `"RUL0021"` |
| start_time | DateTime | Timestamp when the alert or event started. Example: `"11/24/20 19:00"` |
| end_time | DateTime | Timestamp when the alert or event ended. Example: `"11/24/20 23:59"` |

Table 10: Anomaly Event Schema Definition

| Field Name | Type | Description |
|---|---|---|
| timestamp | DateTime | The date and time when the anomaly event was recorded. Example: `"4/26/20 14:14"` |
| KPI | String | The key performance indicator being monitored (e.g., `"Cooling Load"`). |
| asset_name | String | The name of the asset or equipment being measured. Example: `"chiller 9"` |
| value | Numeric | The actual measured value of the KPI at the given timestamp. Example: `25978710` |
| upper_bound | Numeric | The upper threshold for the KPI. Exceeding this may indicate an anomaly. |
| lower_bound | Numeric | The lower threshold for the KPI. Falling below this may indicate an anomaly. |
| anomaly_score | Float | A score indicating how likely the data point is an anomaly (typically 0 to 1). |

Table 11: Mapping Table: KPI Anomalies to Failure Codes

| Field Name | Type | Example | Description |
|---|---|---|---|
| kpi_name | String | Cooling Load | Name of the key performance indicator exhibiting anomaly. |
| anomaly_type | String | High | Indicates the direction or nature of the anomaly (e.g., High, Low, Spike). |
| category | String | Operational Failures | Broad class of the failure (e.g., Control System, Structural, External, Human). |
| primary_code | String | OP004 | Primary failure code associated with the anomaly. |
| pri._code_des | String | Incorrect Cooling Zone Operation | Explanation of the primary failure code. |
| seco._code | String | OP004c | More specific sub-code refining the root cause. |
| seco._code_des | String | Improperly Controlled or Shut Off Zones | Description of the secondary failure code. |

Table 12: Unified Event Table Schema Definition

| Field Name | Type | Description |
|---|---|---|
| event_id | String | Unique identifier for the event (can be work order ID, alert ID, anomaly ID, etc.). Example: `"WO-16170"` |
| event_group | String | High-level classification of the event source (e.g., `"WORK_ORDER"`, `"ALERT"`, `"ANOMALY"`). |
| event_category | String | Sub-classification such as preventive maintenance (`"PM"`), corrective maintenance (`"CM"`), etc. |
| event_type | String | Specific code/type of the event (e.g., `"MT001"`, `"RUL0021"`). |
| description | String | Human-readable description of the event. Example: `"Vibration Analysis"` or `"Refrigerant Leak"`. |
| equipment_id | String | Unique ID of the equipment involved in the event. Example: `"CWC04701"` |
| equipment_name | String | Name of the equipment. Example: `"Chiller 1"` |
| event_time | DateTime | Timestamp when the event occurred or was logged. Format: `YYYY-MM-DD HH:MM:SS` |
| note | String | Additional description for this event if necessary |

Table 13: Mapping Table: Alert Rule to Failure Code

| Field Name | Type | Example | Description |
|---|---|---|---|
| rule_id | String | RUL0012 | Identifier for the alert rule triggered by a monitoring system. |
| rule_name | String | Chiller - Low Supply Temperature | Descriptive name of the alert rule logic or threshold condition. |
| primary_code | String | CS005 | ISO failure code associated with the likely root cause. |
| primary_code | String | Control System Malfunction | Human-readable explanation of the failure code. |

# D AssetOpsBench Scenarios

## D.1 Scenarios Creation Principles

The scenarios in **AssetOpsBench** are designed to evaluate the capabilities required for autonomous agents operating in real industrial environments. Although grounded in real operational data and engineering practices, each scenario is intentionally framed to test a specific dimension of agent reasoning, tool interaction, and decision-making relevant to asset management. The scenarios are built around four core principles:

- **Reasoning and Tool Use:** Scenarios require agents to perform domain-specific reasoning such as time-based logic, schema interpretation, and multi-step tool invocation. Common failure cases include premature termination, incorrect parameter selection, or misuse of diagnostic tools.

- **Data Handling and Forecasting:** Agents must interpret telemetry, detect anomalies, and configure appropriate models for forecasting or anomaly detection. Tasks emphasize the translation of real-world engineering intuition into ML configuration steps (e.g., model selection, training windows, thresholds).

- **Agent Communication and Coordination:** Many scenarios simulate multi-agent workflows where the agent must ask clarifying questions, summarize findings, or coordinate subtasks. This reflects how real engineering teams collaborate during diagnostics or planning.

- **Workflow Orchestration and Decision-Making:** Scenarios measure the agent's ability to plan complex workflows, handle dependencies, reason under uncertainty, and determine when to stop or escalate due to missing or conflicting information.

These principles ensure that scenarios remain faithful to real asset-management workflows while systematically probing the capabilities of autonomous agents.

## D.2 Scenario Generation

The scenarios in **AssetOpsBench** were generated from real industrial operations and shaped through an 18-month collaboration with reliability engineers, controls specialists, and domain experts responsible for large portfolios of mechanical assets (e.g., AHUs, chillers, boilers, compressors). Unlike synthetic rule-based benchmarks, the scenarios are grounded in operational conditions, OEM specifications, maintenance records, and engineering workflows used in practice.

The development process was iterative and domain-driven. Subject-matter experts first identified high-impact failure modes and diagnostic tasks central to asset health, safety, and performance. For each asset type, expert engineers drafted scenario templates that captured realistic fault signatures, cross-sensor interactions, physical constraints, and contextual operating conditions. These templates underwent multi-round reviews involving 3–7 experts to ensure that each scenario reflects plausible field behavior and aligns with real diagnostic reasoning patterns.

Across the 18-month timeline, the scenario library evolved through 12 major iterations. Each iteration added new scenario types, refined diagnostic narratives, and updated failure descriptions based on expert insights and validation against real data patterns. While generating a single scenario is relatively quick, ensuring its realism, consistency, and clarity required significant expert effort including documentation, cross-checking with historical data, and verification of operational plausibility.

Overall, the AssetOpsBench scenarios form a rigorously curated, expert-validated collection of operational situations that reflect how reliability engineers analyze equipment behavior in real-world industrial settings.

## D.3 Scenario Statistics

As shown in Table 14, **AssetOpsBench** includes a total of 141 scenarios with 99 single-agent scenarios and 42 multi-agent scenarios. These scenarios are to be open source research community.

Table 14: Examples of Scenario with their Subtypes (Aligned with Task Taxonomy - Figure 2(b))

| Agent Group | Subtype | Task Descriptions |
|---|---|---|
| **TSFM Agent** 
 # Scenarios: 23 | Forecasting 
 Model Tuning 
 Anomaly Detection 
 Hybrid Tasks 
 Model Capabilities | Predict future KPI trends over time windows 
 Select or refine time series models for accuracy 
 Identify deviations in operational behavior 
 Combine prediction with anomaly evaluation 
 Query TSFM model limits and configurations |
| **Work Order Agent** 
 # Scenarios: 36 | Retrieval & Filter 
 Event Summary 
 Scheduling 
 RCA & Alert Review 
 KPI-based Reco. | Filter work orders by asset, location, or time 
 Summarize logs or alerts over time windows 
 Recommend or optimize work order sequences 
 Perform root cause or alert logic review 
 Link alerts or KPI trends to work orders |
| **Multi-Agent (End-to-End) Tasks** 
 # Scenarios: 42 | Knowledge Query 
 Failure Reasoning 
 Sensor Mapping 
 Sensor Inventory 
 Other | Tasks involving anomaly detection or forecasting 
 Uses degradation models and causal logic 
 Maps failure modes to sensors 
 Retrieves installed sensors on an asset 
 Multi-step inference or decision-making |

The goal is to test an agent's ability across four capability dimensions: 🔧 Tool-Centric (e.g., tool and API interaction), Skill-Centric (e.g., analytical reasoning), 🧩 Domain-Centric (e.g., context-aware decision-making), and 🌐 LLM-Centric (e.g., language-based generalization across tasks). Each scenario is associated with an utterance to complete a task. Table 14 summarizes the distribution of scenario subtypes and their alignment with the task taxonomy. Utterance-507 represents an 🌐 **LLM-Centric** scenario, where the agent must recognize that forecasting task is redundant in the presence of a zero-valued sensor reading—indicating that the machine may not be operating. The agent is expected to bypass unnecessary computation and recommend halting diagnostics to address the root issue directly. In contrast, Utterance-511 exemplifies a **Skill-Centric** task, requiring the agent to correlate energy consumption with a power input variable and construct a corresponding model. This scenario tests the agent's analytical reasoning over telemetry data to uncover functional relationships.

## D.4 SCENARIO EXAMPLES

We include two examples (Table 15 and Table 16) that showcase distinct behaviors of agent outputs. Readers can observe that the *characteristic form* varies even for problems that appear similar on the surface.

Table 15: Example Knowledge Query: Energy Prediction for Chiller 9

| Field | Description |
|---|---|
| **ID** | 507 |
| **Type** | Knowledge Query |
| **Text** | What is the predicted energy consumption for Chiller 9 in the week of 2020-04-27 based on data from the MAIN site? |
| **Characteristic Form** | The expected response should confirm the successful execution of all required actions, ensuring that the correct asset (**Chiller 9**), location (**MAIN**), and time range (**week of 2020-04-27**) were used for data retrieval and analysis. It should specify that the agent identified the sensor name (*power input sensor*) and retrieved the historical energy consumption data for Chiller 9 during the specified time period. 
 The response must also explain that the agent attempted to analyze the data for energy consumption prediction, but was unable to do so due to insufficient data, as the power input for Chiller 9 was consistently $0.0$ from 2020-04-20 to 2020-04-25, indicating that the chiller was not operating. |

Table 16: Example Knowledge Query: Predicting Energy Usage for Chiller 9

| Field | Description |
|---|---|
| ID | 511 |
| Type | Knowledge Query |
| Text | Can you predict Chiller 9's energy usage for next week based on data from the week of 2020-04-27 at MAIN? |
| Characteristic Form | The expected response should confirm the successful execution of all required actions, ensuring that the correct asset (**Chiller 9**) and location (**MAIN site**) were used for data retrieval and analysis. It should specify that the agent first identified the sensors for Chiller 9, then selected the *Chiller 9 Power Input* sensor, and successfully retrieved the energy usage data for the specified time period. |
| | The response should confirm that the agent provided the file path where the data is stored. Additionally, it should mention that although the agent initially encountered errors while analyzing the data and making predictions, it successfully corrected its mistakes and finetuned a **Time Series Forecasting model** using the provided data. The agent should have used the finetuned model to generate predictions for the next week, with the results being stored in the specified file. |

## D.5 SCENARIO COMPARISON WITH OTHER BENCH

We prepare a table to compare with the literature in Table 17. AssetOpsBench extends prior benchmarks by incorporating temporal/dynamic queries, name disambiguation, and tool-output–driven operations. These capabilities not present in TaskBench or ITBench. Additionally, while earlier benchmarks rely on either complex tool graphs or simpler single-step tools, AssetOpsBench emphasizes multi-step tool reuse, aligning better with real industrial agent workflows.

Table 17: Comparative overview of general-purpose and domain-specific benchmarks.

| Benchmark | TaskBench (NeurIPS 2024) | ITBench (ICML 2025) | AssetOpsBench (Ours) |
|---|---|---|---|
| Data Generation | Tool Graph + Back-Instruct | Manual | Manual |
| Tool Dependency | ✓ | ✓ | ✓ |
| Quality Control | LLM Self-critique + Rule-based | Human Verification | Human Verification |
| Evaluation | Task Decomposition + Tool Selection + Parameter Prediction | ReActive Planning + Tool Selection | ReActive Planning + Tool Selection + Parameter Prediction |
| Tool Complexity | Single tool to complex tool graph | – | Multiple tools; same tools can be called multiple times |
| Dataset Scale | 17,331 samples | 141 scenarios | 141 scenarios |
| Temporal / Dynamic Query | × | × | ✓ |
| Name Disambiguation | × | × | ✓ |
| Tools Output Operation | × | × | ✓ |

### D.6 User Study Reliability Analysis

To quantitatively assess the realism of AssetOpsBench scenarios, we conducted a human evaluation study. We randomly selected 25 representative scenarios covering four categories: IoT queries, time-series forecasting (TSFM), work orders/events, and failure mode reasoning (FMSR). Participants were domain experts, including reliability engineers, maintenance engineers, and data scientists familiar with condition-based monitoring and predictive maintenance. Each participant evaluated scenarios using a 3-point scale (*1 = Not Realistic, 2 = Realistic, 3 = High Realistic*) and could optionally provide qualitative comments. Background questions captured participants' role, years of experience, and familiarity with predictive maintenance. Responses were collected via a Google Form as shown in Figure 16.

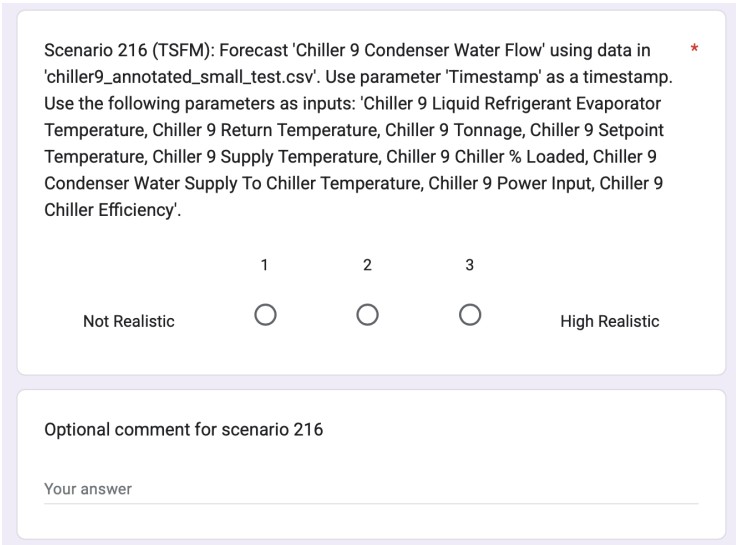

Figure 16: Representative example of Scenario for Collecting user feedback

The following metrics were computed to assess internal consistency, inter-rater agreement, and the reliability of aggregated scores.

### D.7 Reliability Metrics

Table 18: Summary of Reliability Metrics for User Study Ratings

| Metric | Value |
|---|---|
| Cronbach's Alpha | 0.8871 |
| ICC(1) | 0.2334 |
| ICC(2) | 0.8817 |
| Fleiss' Kappa | 0.2093 |

### D.8 Interpretation

The internal consistency of the 25 scenario ratings is excellent, as indicated by a Cronbach's alpha of 0.887. The ICC(1) value of 0.233 reflects moderate agreement at the individual-participant level, whereas ICC(2) of 0.882 demonstrates that the aggregated ratings across participants are highly reliable. Fleiss' kappa of 0.209 indicates slight-to-fair categorical agreement among participants, which is consistent with the subjective nature of realism judgments. Overall, while individual participants may vary in their ratings, the averaged scores per scenario provide a stable and trustworthy measure of perceived realism.

## E  GROUND TRUTH PREPARATION FOR REFERENCE-BASED EVALUATION

To ensure that each scenario can be objectively evaluated, we first construct a ground-truth specification that precisely defines the expected reasoning steps and final answer. The example in Listing 3 illustrates FMSR agent's task where the system must retrieve sensor names associated with a wind turbine. The ground truth includes the task description, the required planning step, and the exact sequence of execution actions that a correct agent should follow. By explicitly defining the operations—such as calling get_available_sensor_information with the asset name "Wind Turbine", the ground truth provides a verifiable reference trace. This structure allows us to compare an agent's generated actions and outputs against a deterministic set of expected behaviors, ensuring consistent and reproducible evaluation across models.

Listing 3: Example FMSR task specification.

```
{
  "id": 105,
  "type": "FMSR",
  "deterministic": false,
  "characteristic_form": "the answer should contain a list of sensor
      names for asset wind turbine.",
  "text": "Provide some sensors of asset Wind Turbine.",
  "planning_steps": [
    "Provide some sensors of asset Wind Turbine."
  ],
  "execution_steps": [
    {
      "name": "get_available_sensor_information",
      "action": "Get Available Sensor Information",
      "arguments": "Wind Turbine",
      "outputs": "[a list of sensor names]"
    },
    {
      "name": "finish",
      "action": "Finish",
      "arguments": "",
      "outputs": ""
    }
  ],
  "execution_links": [
    {
      "source": "get_available_sensor_information",
      "target": "finish"
    }
  ]
}
```

### E.1  PLAN-EXECUTE REFERENCE-BASED SCORING

To assess the fidelity of generated outputs, we perform **reference-based scoring** using ROUGE metrics. This evaluation is limited to the **Plan-Execute** paradigm to maintain consistency and preserve the experimental flow. ROUGE metrics used include:

- rouge1: unigram (1-gram) overlap between generated and reference outputs.
- rouge2: bigram (2-gram) overlap.
- rougeL: longest common subsequence between generated and reference sequences.
- rougeLsum: line-wise longest common subsequence for multi-line outputs.

### E.2  EXECUTION CHAIN EVALUATION

To systematically evaluate agent task execution, we design a **chain-based execution scoring** method. In many scenarios, an agent performs a sequence of steps corresponding to *Think-Act-*

*Observe* cycles. Ground truth data provides the expected sequence of steps for each task. Each executed step contains a `name` (representing the action) and an `arguments` field.

Our scoring approach compares an agent's executed sequence with the ground truth sequence using three criteria:

1. **Step Matching:** The `name` of each executed step is matched to the corresponding ground truth step. Unlike exact matching, we allow fuzzy matching based on string similarity using a threshold to account for minor variations in step names.

2. **Argument Similarity:** Step arguments are treated as strings and compared using a ROUGE-like similarity metric (via `difflib.SequenceMatcher`). This captures cases where the agent produces slightly different or paraphrased arguments.

3. **Sequence Coverage and Order:**
   - Coverage penalizes missing ground truth steps.
   - Extra steps are penalized proportionally.
   - Order preservation is evaluated: steps executed out-of-order incur a penalty.

The final **Execution Chain Score** for a single trajectory is computed as:

Score = (Average argument similarity over matched steps) $\times (1-$extra step penalty$)\times(1-$order penalty$)$

This produces a single scalar in $[0, 1]$ summarizing how closely an agent's execution matches the ground truth. Algorithm 1 outline the entire process.

---

**Algorithm 1:** Compute Chain Execution Score

---

**Input** : Ground truth steps $GT$, agent steps $AG$, name threshold $\theta$, name weight $w_n$, argument weight $w_a$

**Output:** Final chain execution score $S \in [0, 1]$

$matched \leftarrow \emptyset$;
$step\_scores \leftarrow []$;
**foreach** $gt \in GT$ **do**
    $best\_score \leftarrow 0$;
    $best\_idx \leftarrow$ None;
    **foreach** $ag \in AG$ **do**
        **if** $ag \in matched$ **then**
            continue;
        $name\_sim \leftarrow similarity(gt.name, ag.name)$;
        **if** $name\_sim < \theta$ **then**
            continue;
        $arg\_sim \leftarrow similarity(gt.arguments, ag.arguments)$;
        $score \leftarrow w_n \cdot name\_sim + w_a \cdot arg\_sim$;
        **if** $score > best\_score$ **then**
            $best\_score \leftarrow score$;
            $best\_idx \leftarrow$ index of $ag$;
    **if** $best\_idx \neq$ *None* **then**
        add $best\_idx$ to $matched$;
    append $best\_score$ to $step\_scores$;
$step\_coverage \leftarrow average(step\_scores)$;
$extra\_penalty \leftarrow \frac{|AG|-|matched|}{|GT|+|AG|-|matched|}$;
$order\_penalty \leftarrow$ fraction of inversions in $matched$ indices;
$S \leftarrow step\_coverage \cdot (1 - extra\_penalty) \cdot (1 - order\_penalty)$;
**return** $S$;

---

## F  ADDITIONAL BENCHMARK EXPERIMENTS

This appendix section contain outcome of an extensive benchmark we conducted in this paper.

### F.1 LLM-AS-A-JUDGE EVALUATION AGENT AND HUMAN VALIDATION

In Listing 19, we provided the system prompt that we used for generating a rubric metric for the evaluation agent. Given LLM is used for generating the rubric metric, we also conducted human validation of these generated metric. The results shown in Section 5.1, `llama-4-maverick` is selected to be the LLM of evaluation agent. Table 19 is the prompt instruction to the evaluation agent, which outlines the specific evaluation dimensions, constraints, and response formatting guidelines that the model follows when scoring task outputs. The evaluation criteria is also provided to human judges which ensures consistency across evaluations.

> You are a critical reviewer tasked with evaluating the effectiveness and accuracy of an AI agent's response to a given task. Your goal is to determine whether the agent has successfully accomplished the task correctly based on the expected or characteristic behavior.
> **Evaluation Criteria:**
> 1. **Task Completion**:
> - Verify whether the agent executed all required actions (e.g., using the correct tools, retrieving data, performing the necessary analysis).
> - Ensure the response aligns with the predefined expected behavior for task completion.
> 2. **Data Retrieval & Accuracy**:
> - Confirm that the correct asset, location, time period, and sensor (if applicable) were used.
> - Check that the retrieved data and results (forecasting, anomaly detection, etc.) are correct and consistent with the task requirements.
> 3. **Generalized Result Verification**:
> - Task Type Verification: Assess if the agent returned the expected results for the task type (forecasting, anomaly detection, classification, etc.).
> - Forecasting: Ensure forecasts cover the specified future period.
> - Anomaly Detection: Verify that anomalies were correctly detected when expected.
> - Other Tasks (e.g., classification): Check that results match expected format and values.
> - Comparison with Expected Output: Validate that results match the characteristic answer.
> - Data Integrity: Ensure correct data (sensor, time period) was used and output format is consistent.
> **Inputs:**
> Question: {question}
> Characteristic Answer (Expected Behavior): {characteristic_answer}
> Agent's Thinking: {agent_think}
> Agent's Final Response: {agent_response}
> **Output Format:**
> Provide your review strictly in JSON format without any additional text or Markdown.
> {
> "task_completion": true/false,
> "data_retrieval_accuracy": true/false,
> "generalized_result_verification": true/false,
> "suggestions": "Optional. Recommended actions to improve the agent's response if needed."
> }
> (END OF RESPONSE)
> Evaluate the agent's performance according to the above criteria.

Table 19: Prompt instruction for LLM-as-a-judge evaluation agent

Human validation was performed using Google Forms. As illustrated in Figure 17, domain experts were presented with the original task description, the agent's reasoning and final answer, and a checklist covering six evaluation dimensions. Each dimension was evaluated using binary judgments (True/False), enabling structured comparison between human and model evaluations. Four forms, each containing 10 samples, were distributed, yielding a total of 240 data points. The aggregated results are reported in Section 5.1.

### F.2 DEEP INVESTIGATION OF AGENT-AS-TOOL PERFORMANCE

To evaluate the capabilities of various large language models (LLMs) across a range of industrial-relevant task categories, we present a radar chart (See Figure 18) comparison covering five key dimensions: *IoT-focused reasoning*, *Failure Mode and Sensor Reasoning (FMSR)*, *Time Series and Fault Modeling (TSFM)*, *Work Order (WO) understanding*, and *End-to-End task integration*. The chart illustrates normalized performance scores for each model based on task-specific

**AssetOpsBench Human Judge Test**

Group: 1, Index: 0

## Task: ##
Generate a machine learning recipe for detecting overheating failure for Chiller 6. Result should include feature sensors and target sensor.

## Final answer: ##
For detecting "Compressor Overheating" failure in Chiller 6, the relevant feature sensors are: 'Chiller 6 Power Input', 'Chiller 6 Evaporator Water Outlet Temperature', 'Chiller 6 Condenser Water Inlet Temperature', 'Chiller 6 Condenser Water Outlet Temperature', and 'Chiller 6 Compressor Discharge Temperature'. The target sensor is 'Chiller 6 Compressor Discharge Temperature'. A machine learning model can be trained using these features to predict the likelihood of compressor overheating.  ## Step 1: Understand the Task The task is to generate a machine learning recipe for detecting overheating failure for Chiller 6, including feature sensors and target sensor.  ## Step 2: Identify Failure Modes for Chiller 6 The failure modes for Chiller 6 are identified as: 'Compressor Overheating: Failed due to Normal wear, overheating', 'Heat Exchangers: Fans: Degraded motor or worn bearing due to Normal use', 'Evaporator Water side fouling', 'Condenser Water side fouling', 'Condenser Improper water side flow rate', 'Purge Unit Excessive purge', 'Refrigerant Operated Control Valve Failed spring'.  ## Step 3: Determine Relevant Sensors for "Compressor Overheating" The relevant sensors for monitoring "Compressor Overheating" are: 'Chiller 6 Power Input', 'Chiller 6 Evaporator Water Outlet Temperature', 'Chiller 6 Condenser Water Inlet Temperature', 'Chiller 6 Condenser Water Outlet Temperature', and 'Chiller 6 Compressor Discharge Temperature'.  ## Step 4: Compile Machine Learning Recipe For detecting "Compressor Overheating" failure in Chiller 6, the relevant feature sensors are: 'Chiller 6 Power Input', 'Chiller 6 Evaporator Water Outlet Temperature', 'Chiller 6 Condenser Water Inlet Temperature', 'Chiller 6 Condenser Water Outlet Temperature'. The target sensor is 'Chiller 6 Compressor Discharge Temperature'. A machine learning model can be trained using these features to predict the likelihood of compressor overheating.  ## Step 5: Provide Final Answer For detecting "Compressor Overheating" failure in Chiller 6, the relevant feature sensors are: 'Chiller 6 Power Input', 'Chiller 6 Evaporator Water Outlet Temperature', 'Ch

## Thinking process: ##
Thought: I need to generate a machine learning recipe for detecting overheating failure for Chiller 6, which includes identifying the relevant feature sensors and the target sensor. I will use FMSRAgent to achieve this.

Action: Tool Name: FMSRAgent , Tool Parameter: Action Input 1: request=Generate a machine learning recipe for detecting overheating failure for Chiller 6, including feature

Figure 17: Google Forms: questionnaire to domain experts for human validation

benchmarks, with higher values indicating stronger task alignment. Among the models compared, `gpt-4.1-2025-04-14` demonstrates the most consistent and well-rounded performance, achieving near-saturation in FMSR (100%) and strong results in End-to-End integration. In contrast, `granite-3-3-8b-instruct` and `llama-3-3-70b-instruct` perform well in TSFM and FMSR but underperform in WO-related tasks, which are particularly challenging due to their dependence on structured document comprehension and task planning. The `llama-4-maverick` model shows promising results in WO and End-to-End integration, indicating a potential optimiza-

tion for cross-domain contextual reasoning. This visualization provides a holistic view of model strengths and trade-offs, offering insights for selecting and fine-tuning LLMs in complex, multi-modal industrial applications.

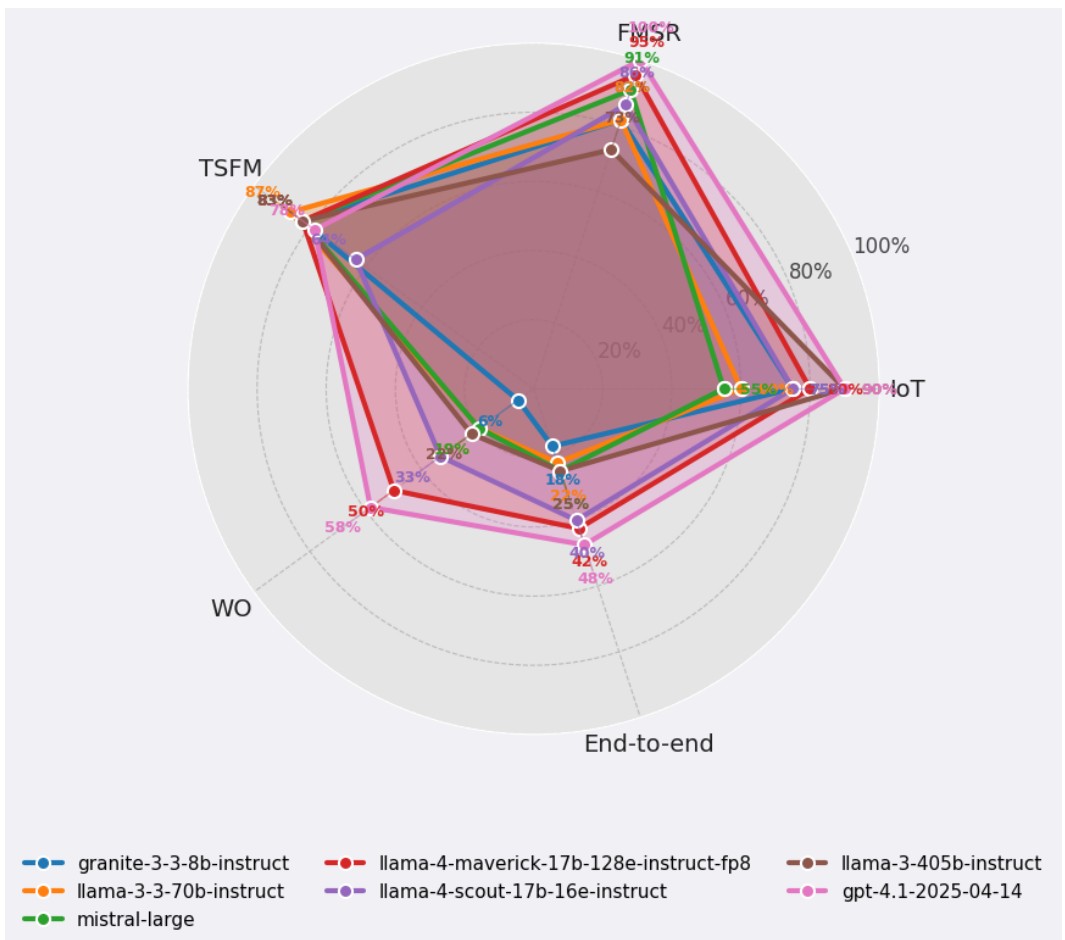

Figure 18: Task wise distribution of the Accomplished Tasks.

---

**Insight I**

The radar chart highlights that while `gpt-4.1-2025-04-14` delivers the most balanced performance across all industrial task categories, other models exhibit strong specialization. For example, `granite-3-3-8b-instruct` and `llama-3-3-70b-instruct` excel in FMSR and TSFM but struggle with WO-related tasks. This reveals a clear trade-off between broad generalization and domain-specific strengths, suggesting that hybrid or task-specialized agent architectures may be most effective in practice.

---

### F.3 FAILURE ANALYSIS ON TOOL USE

As part of the AssetOpsBench trajectory analysis, we examine how agents interact with the available toolset and execution environment. We start this effort from the analyze (1) the complete list of all tools in Table 5 (containing the details of their name, usage, and parameters), and (2) those JSON snippets illustrating how agent actions are logged. Across **834 trajectories**, collected from multiple LLM models and multiple agent configurations, we store every agent step as a structured JSON record containing both the *action type* and its *execution state*. This representation allows us to distinguish between **Tool-oriented actions** (invocations of predefined data retrieval, analysis, and

analytic functions) and **CodeReAct-oriented actions** (Python code generated and executed on the fly).

Across **834 trajectories** collected from multiple LLM models and agent configurations, every agent step is logged as a structured JSON record containing both the *action type* (e.g., Tool-oriented or CodeReAct-oriented) and its associated *execution state*. This enables a unified analysis of failure cases across heterogeneous agent architectures.

We distinguish between **Tool-oriented actions**, which invoke predefined data retrieval, analysis, or analytic functions, and **CodeReAct-oriented actions**, where the agent emits executable Python code. Each action record includes fields such as `action`, `action_input`, `observation`, and `state`. A failure is captured when the framework sets `state = "Invalid Action"` and records the underlying error in the `observation` field. These signals form the basis of our failure-mode statistics.

**Example 1: Tool-Oriented Invalid Action (trimmed).**

```
{
  "step": 1,
  "thought": "Use 'Read Sensors From File' to list sensors for Chiller 6.",
  "action": "Read Sensors From File",
  "action_input": "/path/to/chiller_6_sensors.txt",
  "observation": "Error: 'NoneType' object has no attribute 'replace'",
  "state": "Invalid Action"
}
```

This failure arises from malformed input to the `Read Sensors From File` tool. The associated error message is used directly in computing tool-level failure frequencies (see Figure 19).

**Example 2: CodeReAct (Dynamic Python) Invalid Action (trimmed).**

```
{
  "step": 2,
  "thought": "Write preventive work orders to JSON.",
  "action": "import json
with open('pwo.json','w') as f:
    json.dump(pwo_list, f)",
  "observation": "Invalid action: Object of type Timestamp is not JSON serializable",
  "state": "Invalid Action"
}
```

This *CodeReAct* step is syntactically correct but fails during execution because the object being serialized contains a non-JSON-serializable `Timestamp`. These runtime failures contribute to the action-state distribution analysis.

Figure 19 summarizes the distribution of valid and invalid executions for both action classes. Tool-oriented actions benefit from well-defined schemas, while CodeReAct-oriented actions incur more failures due to the variability of generated Python code.

Figure 20 highlights the concentration of Tool-oriented failures within a handful of tools such as `jsonreader`, `tsfm_integrated_tsad`, and `Read Sensors From File`. These patterns reflect the structural complexity of their inputs and outputs. The JSON fragments above illustrate the types of failures contributing to these distributions.

> **Execution-State Insight**
>
> Analysis of 834 agent trajectories reveals that **Tool-oriented actions** achieve higher valid-execution rates due to well-defined input/output schemas, whereas **CodeReAct-oriented actions** suffer more runtime failures from dynamic Python generation. Failures in Tool-oriented actions are concentrated in a few complex tools, highlighting that structural complexity and input validation are critical factors in agent reliability.

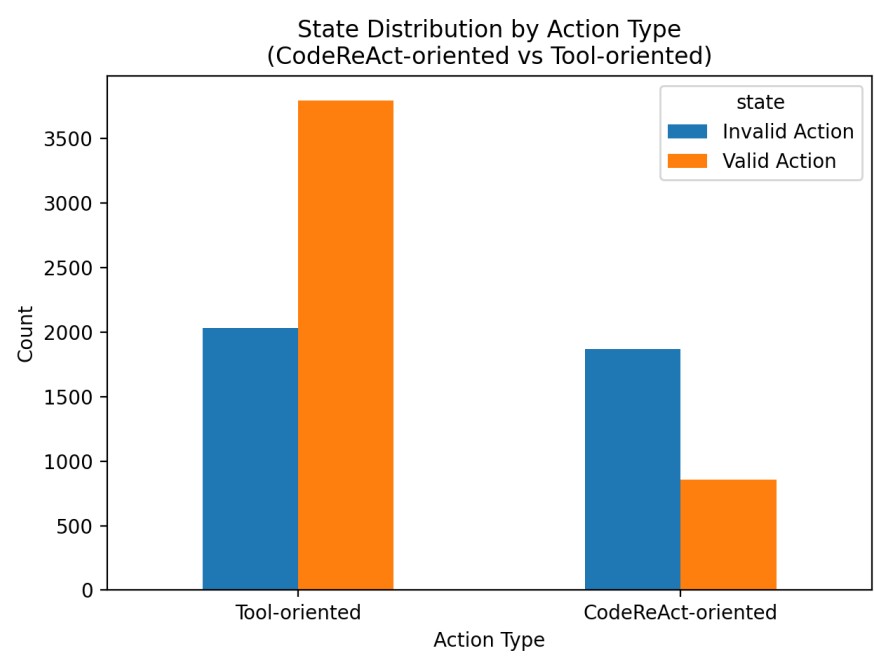

Figure 19: Distribution of execution states across action types. Tool-oriented actions exhibit higher valid-execution rates, whereas dynamically generated Python produces more execution failures.

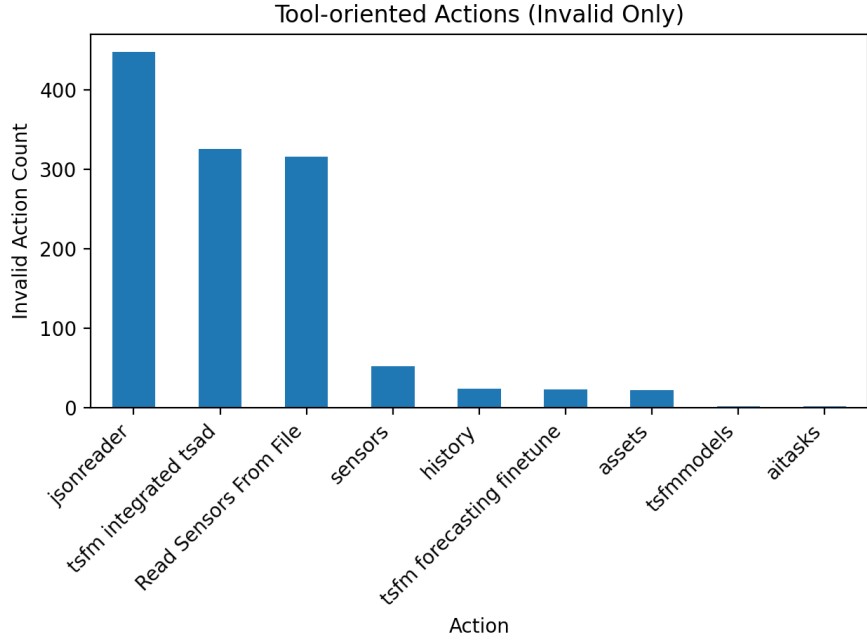

Figure 20: Invalid-only failures for Tool-oriented actions. Errors are concentrated in a small number of frequently used tools with complex parsing or I/O behavior.

## F.4    EXECUTION EFFICIENCY

In this section, we analyze AssetOpsBench execution efficiency of 7 LLMs, complementing the Leaderboard results in Section 5.1. Tables 20 and 21 present results from two multi-agent imple-

mentations. Metrics include the average number of steps taken per task and the average runtime (in seconds) per task.

In the **Agent-As-Tool** execution mode, most models demonstrate relatively stable planning behavior across both single-agent and multi-agent tasks. Compared to the Plan-Execute setting, models here generally take more steps but operate with greater runtime efficiency. `gpt-4.1` again exhibits strong performance, balancing a higher number of steps with moderate runtime, indicating precise control over tool invocation. Interestingly, `llama-3-70b-instruct` shows competitive efficiency, achieving the lowest runtime in both task categories despite slightly fewer steps, suggesting quicker tool usage or lower overhead per step. On the other hand, `mistral-large` exhibits extreme runtime variability, skewed by a pathological case involving prolonged `JSONReader` calls over large datasets. These results suggest that while tool-based execution benefits from more direct action control, its efficiency is highly sensitive to the invoked tools and data volume.

Table 20: Execution Statistics for Agent-As-Tool: Average Steps and Runtime Per Task

| Model | Single-Agent Tasks | | Multi-Agent Tasks | |
|---|---|---|---|---|
| | **Steps** | **Runtime (sec)** | **Steps** | **Runtime (sec)** |
| `gpt-4.1` | $6.0 \pm 2.4$ | $104 \pm 178$ | $6.4 \pm 2.5$ | $218 \pm 371$ |
| `mistral-large` | $4.9 \pm 2.6$ | $347 \pm 19871$ | $5.2 \pm 2.2$ | $289 \pm 443$ |
| `llama-3-405b-instruct` | $4.8 \pm 2.5$ | $250 \pm 773$ | $5.6 \pm 2.2$ | $255 \pm 248$ |
| `llama-3-70b-instruct` | $3.9 \pm 1.6$ | $101 \pm 107$ | $4.3 \pm 2.1$ | $151 \pm 220$ |
| `llama-4-maverick-17b-128e` | $4.3 \pm 1.5$ | $120 \pm 258$ | $4.5 \pm 1.7$ | $137 \pm 175$ |
| `llama-4-scout-17b-16e-instruct` | $4.4 \pm 2.0$ | $101 \pm 87$ | $5.8 \pm 2.9$ | $178 \pm 157$ |
| `granite-3-3-8b` | $5.3 \pm 3.1$ | $197 \pm 240$ | $6.6 \pm 3.6$ | $228 \pm 256$ |

High standard deviation is due to one outlier task requiring nearly 5 hours. It repeatedly invoked the `JSONReader` tool to process two years of historical data.

In the **Plan-Execute** setting, the number of steps required for single-agent tasks closely mirrors those of multi-agent tasks, indicating a tendency among LLMs to *over-plan even for relatively simple objectives*. This pattern reflects limited sensitivity to task complexity during the planning phase. Among all evaluated models, `gpt-4.1` consistently outperforms others, demonstrating both *minimal average steps* and *lowest runtime*, particularly in multi-agent tasks. This suggests that `gpt-4.1` leverages more effective internal representations and decision strategies, enabling efficient decomposition and execution of plans. In contrast, models like `granite-3-3-8b` and `llama-3-70b-instruct` show pronounced inefficiency, often executing significantly more steps and incurring higher computational costs. These results highlight a critical trade-off in Plan-Execute agents: while the architecture enforces task structure, its effectiveness heavily depends on the model's reasoning efficiency. Models lacking strong planning priors or execution alignment tend to generate unnecessarily long or suboptimal action sequences, especially in low-complexity settings.

Table 21: Execution Statistics of Plan-Execute Agents: Average Steps and Runtime per Task

| Model | Single-Agent Tasks | | Multi-Agent Tasks | |
|---|---|---|---|---|
| | **Steps** | **Runtime (sec)** | **Steps** | **Runtime (sec)** |
| `gpt-4.1` | $2.6 \pm 1.0$ | $93.3 \pm 105.6$ | $2.9 \pm 1.5$ | $180.2 \pm 122.6$ |
| `mistral-large` | $2.7 \pm 1.3$ | $186.9 \pm 206.9$ | $3.0 \pm 1.4$ | $209.7 \pm 139.1$ |
| `llama-3-405b-instruct` | $3.1 \pm 1.9$ | $208.3 \pm 176.5$ | $4.0 \pm 1.5$ | $224.4 \pm 99.7$ |
| `llama-3-70b-instruct` | $6.7 \pm 1.5$ | $381.8 \pm 240.2$ | $6.5 \pm 0.9$ | $369.6 \pm 151.9$ |
| `llama-4-maverick-17b-128e` | $4.0 \pm 1.9$ | $384.6 \pm 611.6$ | $3.9 \pm 1.2$ | $376.8 \pm 281.0$ |
| `llama-4-scout-17b-16e` | $3.9 \pm 2.0$ | $172.1 \pm 114.7$ | $4.4 \pm 1.5$ | $218.1 \pm 105.4$ |
| `granite-3-3-8b` | $5.2 \pm 1.4$ | $413.3 \pm 418.2$ | $5.1 \pm 1.3$ | $432.9 \pm 294.7$ |

**Conclusion.** While the **Plan-Execute** architecture demonstrates greater efficiency—requiring fewer steps and exhibiting lower runtime variability across tasks—our evaluation shows that **Agent-As-Tool** significantly outperform in task performance metrics. For example, `gpt-4.1` achieves 65% task completion, 77% data retrieval accuracy in the Agent-As-

Tool setting, compared to only 38–44% on most metrics in Plan-Execute. Similarly, `llama-4-maverick-17b-128e-instruct` excels in both setups but scores notably higher in Agent-As-Tool, achieving 59–78% on core performance metrics versus 45–57% in Plan-Execute.

This pattern is consistent across most models: **Agent-As-Tool** incur higher execution costs but deliver better reasoning fidelity. Conversely, **Plan-Execute** agents—while faster and more structured—often struggle with complex retrieval, verification, and consistency tasks. These findings suggest a fundamental trade-off: Plan-Execute offers process efficiency, while Agent-As-Tool yield higher end-task quality—a crucial insight for selecting agent architectures based on application goals such as throughput vs. correctness.

---

**Insight II**

Across both execution modes, our results reveal a fundamental trade-off: *Plan-Execute* agents are faster and more step-efficient, yet consistently underperform in reasoning-intensive tasks, whereas *Agent-As-Tool* agents incur higher execution costs but deliver substantially stronger task accuracy and reliability. This shows that efficiency-oriented architectures do not automatically yield better end-task quality : an important consideration for industrial LLM systems where correctness often outweighs runtime.

---

**Insight III**

Our analysis reveals a subtle behavioral difference across execution modes: although *Single-Agent* tasks are intended to be completed by one agent in a single round, *Plan-Execute* agents tend to over-plan, potentially reusing the same agent multiple times or invoking additional agents. Conversely, *Agent-As-Tool* agents, despite being reactive and theoretically able to finish in one round, often continue executing, indicating the presence of tasks that exceed an individual agent's capability and lead to repeated attempts. This underscores hidden task difficulty and model limitations, which are critical for interpreting execution efficiency alongside task performance. A common observation arising from this behavior is the importance of enabling parent agents to ask questions to other agents within the system to resolve tasks more efficiently.

---

F.5    TASK COMPLETION COMPARISON: AGENT-AS-TOOL VS PLAN-EXECUTE

We evaluate the performance of multiple LLM models on the Task Completion metric under two execution paradigms: **Agent-As-Tool** and **Plan-Execute**. Agent-As-Tool agents execute tasks incrementally by invoking tools as needed, whereas Plan-Execute agents first plan a full sequence of steps before execution. Figure 21 visualizes the leaderboard across seven representative models.

**Observations.**

- `gpt-4.1-2025-04-14` and `llama-4-maverick-17b-128e-instruct-fp8` achieve the highest task completion under the Agent-As-Tool paradigm, reaching 65% and 59% respectively.

- Plan-Execute results show mixed trends: some models, such as `mistral-large`, improve slightly (40% → 46.5%), while others, notably `gpt-4.1`, experience a drop (65% → 38.38%).

- Smaller or older models (`granite-3-3-8b-instruct`, `llama-3-3-70b-instruct`) exhibit lower task completion across both paradigms, highlighting limitations in multi-step reasoning and tool integration.

- Differences between the paradigms suggest that pre-planning can sometimes limit adaptability, whereas reactive Agent-As-Tool strategies better leverage incremental reasoning and tool invocations for complex tasks.

**Insights.**    The comparison highlights several key points:

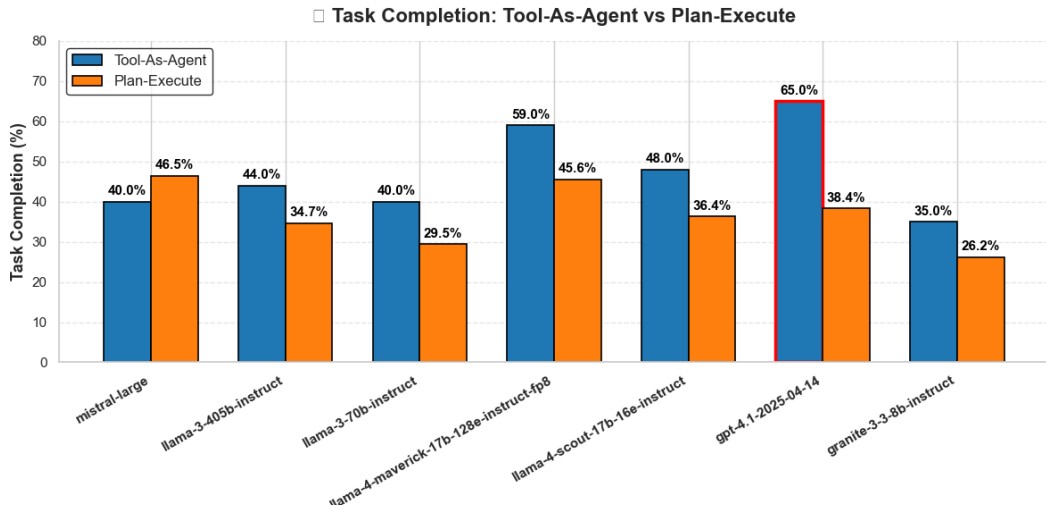

Figure 21: Task Completion scores (%) for Agent-As-Tool and Plan-Execute paradigms. Values above bars indicate the actual completion percentage for each model.

1. **Model capabilities and adaptability:** High-performing models demonstrate both accurate reasoning and effective tool utilization. GPT-4.1 and LLaMA-4-Maverick excel when allowed to execute incrementally in the Agent-As-Tool paradigm.

2. **Impact of execution paradigm:** Agent-As-Tool allows flexible, context-aware reasoning, particularly for models capable of dynamically selecting the next best action. Plan-Execute may underperform if initial plans are suboptimal or if unexpected states arise during execution.

3. **Guidance for benchmark design:** Reporting Task Completion under both reactive (Agent-As-Tool) and planned (Plan-Execute) paradigms is crucial to capture model strengths and weaknesses in real-world agentic tool-use.

4. **Future directions:** Incorporating dynamic feedback loops in Plan-Execute or hybridizing with Agent-As-Tool strategies may further improve task completion, especially for larger multi-step tasks.

---

**Insight III**

The Task Completion comparison highlights that execution paradigm significantly interacts with model capability. High-performing models such as `GPT-4.1` and `LLaMA-4-Maverick` achieve their best results under the *Agent-As-Tool* paradigm, leveraging incremental reasoning and dynamic tool selection. In contrast, Plan-Execute sometimes constrains adaptability, leading to lower performance for models with strong reasoning potential. This suggests that reactive, step-wise strategies can better exploit advanced model reasoning, whereas pre-planned sequences may underutilize model strengths, particularly on multi-step or complex tasks.

---

F.6 RUNTIME AND COST ANALYSIS

Table 22 provides a representative comparison of total runtime and estimated cost for executing the full 140+ utterance task suite using the Agent-As-Tool paradigm. Average tokens per task and total cost are shown for different LLMs.

Table 22: Runtime and estimated cost for executing 140+ utterance tasks using the Agents-as-Tools paradigm.

| LLM | Provider | Avg Tokens per Task | Total Cost (USD) |
|---|---|---|---|
| gpt-4.1 | OpenAI | $\approx$3,664 | $300.00 |
| llama-4-maverick | Watsonx | $\approx$3,730 | $130.00 |

## F.7 Uncertainty Analysis

As discussed in Section 5, the evaluation agent was run five times to produce reliable performance metrics. Table 23 shows the inter-rater agreement across these five evaluation runs, along with the derived uncertainty (computed as $1 -$ agreement). The average agreement and uncertainty across all metrics are also reported.

Table 23: Inter-rater agreement and derived uncertainty across five evaluation runs.

| Metric | Agreement | Uncertainty |
|---|---|---|
| Task Completion | 0.9731 | 2.69% |
| Data Retrieval Accuracy | 0.9697 | 3.03% |
| Generalized Result Verification | 0.9681 | 3.19% |

## F.8 Ablation Experiments

In this section, we present the detailed report of the ablation study. We fixed the Agent-As-Tool paradigm and conducted both sets of experiments.

### F.8.1 Distractor Agents

We have introduced 10 distractor agents to intentionally increase the complexity and ambiguity for global agents. Table 24 categorizes these agents based on their respective domains and functional roles. The set includes both general-purpose agents, such as those for echoing inputs or handling off-topic queries, and domain-specific agents focused on tasks like predictive maintenance, sensor data summarization, and edge ML deployment. This taxonomy enhances the realism of multi-agent environments by supporting modular integration and introducing controlled confusion.

Across 99 scenarios, we compare language models with and without distractor agents to evaluate their robustness in agentic tool-use settings. Table 25 shows the result. `GPT-4.1` remains the strongest model overall, achieving the highest scores in task completion, data accuracy, and result verification across both settings. `Llama-4-Maverick` emerges as the best-performing open-weights model, showing not only high accuracy but also improved performance when distractors are introduced. In contrast, models such as `Mistral-Large` and `Llama-4-Scout` experience moderate degradation under distractors, indicating sensitivity to noisy action spaces. `Granite-3-3-8B` remains stable across conditions but at a lower overall accuracy level, showing reliability but limited reasoning depth.

The introduction of distractor agents reveals interesting behavioral differences. While most models suffer performance drops, `Llama-3-405B` and `Llama-4-Maverick` improve across all three evaluation dimensions, suggesting strong corrective reasoning and robustness to tool-selection noise. These results highlight a tiered landscape of model reliability: `GPT-4.1` at the top, followed by mid-tier models with varying sensitivities, and smaller models offering stability at reduced capability. Overall, the findings underscore the importance of evaluating both accuracy and robustness, as real-world agentic systems often face ambiguous or misleading tool/agent choices.

Table 24: Agent Types and Their Roles

| Agent Name | Domain | Description |
|---|---|---|
| Echo Agent | General | Repeats the input verbatim; useful for debugging and testing input-output coherence. |
| OffTopic Agent | General | Provides unrelated facts or trivia when a query is off-topic or not recognized. |
| Customer SupportAgent | Support Operations | Handles customer-related issues like password resets, login errors, and service availability. |
| SRE Agent | Site Reliability | Diagnoses performance degradation, system downtime, and infrastructure issues. |
| Frontend DevAgent | Software Engineering | Assists with frontend UI/UX concerns, React, JavaScript frameworks, and rendering bugs. |
| HRPolicy Agent | Human Resources | Answers HR-related queries like leave policy, benefits, and compliance rules. |
| SensorData Summarizer | Industrial IoT | Summarizes time-series data from sensors, highlighting trends and anomalies. |
| Historical TrendsAgent | Analytics | Extracts and interprets historical asset data to identify failure patterns or optimization opportunities. |
| EdgeML Agent | Edge Computing | Recommends tools and strategies for deploying ML models on edge hardware with limited resources. |
| RULPredictor Agent | Predictive Maintenance | Estimates the remaining useful life (RUL) of assets using sensor data and degradation models. |

---

**Insight 4: Robustness to Distractor Agents**

Across 99 scenarios, `GPT-4.1` consistently demonstrates top performance, while `Llama-4-Maverick` shows strong open-weight results with robustness to distractor agents. Mid-tier models exhibit varying sensitivity to noisy action spaces, and smaller models like `Granite-3-3-8B` maintain stability but with lower overall accuracy. These findings reveal a tiered landscape of model reliability, emphasizing that both task accuracy and resilience to distractors are critical for practical agentic tool-use systems.

---

### F.8.2 IMPACT OF IN-CONTEXT EXAMPLES

Table 26 provides a detailed comparison of gpt-4.1 and granite-3-3-8b with and without in-context examples on a subset of single-agent benchmark tasks. Consistent with our main findings, in-context examples were critical for enabling effective reasoning and coordination.

**Key Results:** Removing in-context examples led to a dramatic drop in performance for both models. gpt-4.1 dropped from an average of 80% (with context) to 33% (without), while granite-3-3-8b fell from 60% to just 3% (Section F.8). These results reinforce the conclusion that in-context exam-

Table 25: Comparison of Model Performance With and Without Distractor Agents (99 Scenarios).

| Model | Setting | Task Completion | Data Accuracy | Result Verification |
|---|---|---|---|---|
| gpt-4.1-2025-04-14 | Without Distractors | 52 | 57 | 55 |
| | With Distractors | 48 | 56 | 54 |
| granite-3-3-8b-instruct | Without Distractors | 40 | 44 | 41 |
| | With Distractors | 40 | 44 | 41 |
| mistral-large | Without Distractors | 42 | 46 | 43 |
| | With Distractors | 40 | 44 | 41 |
| llama-3-405b-instruct | Without Distractors | 41 | 41 | 38 |
| | With Distractors | 44 | 44 | 44 |
| llama-3-3-70b-instruct | Without Distractors | 38 | 43 | 34 |
| | With Distractors | 41 | 43 | 36 |
| llama-4-maverick | Without Distractors | 46 | 49 | 46 |
| | With Distractors | 48 | 49 | 49 |
| llama-4-scout | Without Distractors | 45 | 44 | 46 |
| | With Distractors | 40 | 40 | 40 |

Table 26: Comparison of `gpt-4.1` and `granite-3-3-8b` With/Without In-Context Examples (# of Tasks = 65)

| Model | In-Context Examples | Task Completion | Data Retrieval Accuracy | Generalized Result Verification |
|---|---|---|---|---|
| gpt-4.1 | Yes | 52 | 57 | 55 |
| granite-3-3-8b | Yes | 40 | 44 | 41 |
| gpt-4.1 | No | 22 | 21 | 24 |
| granite-3-3-8b | No | 2 | 3 | 3 |

ples are essential for ReAct-style reasoning in LLM-based agents. We did not select tasks from WO and E2E since their performance is already poor.

> **Insight: Impact of In-Context Examples**
>
> The presence of in-context examples dramatically improves performance: `gpt-4.1` achieves 80% average accuracy with examples versus 33% without, while `granite-3-3-8b` drops from 60% to 3%. This highlights that effective ReAct-style reasoning critically depends on relevant contextual guidance.

### F.9 PLAN-EXECUTE REFERENCE-BASED SCORING

**Evaluation Setup**. To assess the fidelity of generated outputs, we perform **reference-based scoring** using ROUGE metrics. This evaluation is limited to the **Plan-Execute** paradigm to maintain consistency and preserve the experimental flow.

ROUGE metrics used include:

- `rouge1`: unigram (1-gram) overlap between generated and reference outputs.
- `rouge2`: bigram (2-gram) overlap.
- `rougeL`: longest common subsequence between generated and reference sequences.
- `rougeLsum`: line-wise longest common subsequence for multi-line outputs.

**Results Summary**. ROUGE scores highlight model differences in n-gram and sequence-level fidelity. Table 27 presents sample scores for representative models across Plan-Execute outputs.

Table 27: ROUGE-based reference scoring for Plan-Execute outputs (selected models).

| Model | rouge1 | rouge2 | rougeL | rougeLsum |
|---|---|---|---|---|
| llama-3-405b-instruct | 0.406 | 0.243 | 0.337 | 0.381 |
| mixtral-8x7b-instruct-v01 | 0.424 | 0.259 | **0.343** | 0.401 |
| llama-3-3-70b-instruct | 0.297 | 0.172 | 0.242 | 0.280 |
| gpt-4.1-2025-04-14 | 0.354 | 0.182 | 0.289 | 0.335 |
| granite-3-3-8b-instruct | 0.373 | 0.214 | 0.291 | 0.353 |
| mistral-large | **0.420** | **0.251** | **0.343** | **0.404** |
| llama-4-maverick | 0.403 | 0.240 | 0.325 | 0.383 |

We conduct a more in-depth analysis:

- Top-performing models such as `llama-3-405b-instruct` and `mixtral-8x7b-instruct-v01` achieve `rouge1` ≈ 0.42, `rouge2` ≈ 0.26, and `rougeL` ≈ 0.34, indicating strong n-gram and sequence-level fidelity.

- Smaller or older models exhibit lower ROUGE scores, reflecting weaker lexical alignment with reference trajectories.

- Overall, Plan-Execute outputs maintain higher alignment with reference trajectories, demonstrating that this paradigm supports more faithful generation for skilled reasoning tasks.

- The distribution of ROUGE metrics also reflects diversity in output complexity, as longer or multi-step reasoning tasks tend to lower ROUGE scores despite semantic correctness.

Reference-based scoring provides a quantitative measure of textual fidelity across different models under the Plan-Execute paradigm. These results support model comparison, highlight the impact of LLM size and capabilities, and offer a reproducible benchmark for future studies.

> **Insight**
>
> Although `gpt-4.1` excels in task reasoning and completion, its lower ROUGE scores compared to open-weight models indicate that strong reasoning does not always correspond to higher lexical alignment with reference outputs.

### F.10 AGENT-AS-TOOL REFERENCE-BASED SCORING

In the **Agent-As-Tool** setting, the agent follows a *think–act–observe* cycle without a pre-planning phase. To evaluate reasoning quality, we extract the internal *thinking* segments and compute ROUGE scores against concise reference trajectories. Because ROUGE measures lexical overlap, differences in verbosity strongly affect the outcome.

**Results.** Table 28 reports ROUGE-1/2/L scores along with generation lengths. `mistral-large` achieves the highest performance with ROUGE-1 ≈0.37, ROUGE-2 ≈0.19, and ROUGE-L ≈0.30, followed closely by `llama-3-3-70b-instruct` and `llama-3-405b-instruct`. These models generate reasoning traces of moderate length (48–83 words on average), which aligns well with the reference answers (30 words) and preserves lexical fidelity.

In contrast, models such as `gpt-4.1` and `granite-3-3-8b-instruct` produce significantly longer outputs (up to 277 words on average), resulting in the lowest ROUGE scores despite potentially valid reasoning steps.

**Summary.** Models with output lengths closer to the reference (e.g., `mistral-large`, `llama-3-70B`) achieve higher lexical alignment. However, low-scoring models like `gpt-4.1` may still exhibit rich and correct reasoning, suggesting that token length and prompting strategy—rather than reasoning quality alone— drive ROUGE differences in the Agent-As-Tool paradigm.

Table 28: ROUGE-based comparison for the **Agent-As-Tool** setting. Scores are computed on the extracted *thinking* segments of each trajectory. Longer generations reduce lexical overlap with concise references, lowering ROUGE despite potentially richer content.

| Model | ROUGE-1 | ROUGE-2 | ROUGE-L | ROUGE-Lsum | #Samples | Pred. Avg. Words | GT Avg. Words |
|---|---|---|---|---|---|---|---|
| **mistral-large** | **0.3691** | **0.1933** | **0.2971** | **0.3124** | 40 | **83.0** | 29.85 |
| llama-3-3-70b-instruct | 0.3661 | **0.1963** | 0.2971 | **0.3177** | 40 | 47.8 | 29.85 |
| llama-3-405b-instruct | 0.3394 | 0.1673 | 0.2740 | 0.2787 | 40 | 82.42 | 29.85 |
| llama-4-scout-17b-16e-instruct | 0.3126 | 0.1522 | 0.2398 | 0.2621 | 38 | 100.32 | 29.84 |
| llama-4-maverick-17b-128e-instruct-fp8 | 0.2560 | 0.1252 | 0.2067 | 0.2273 | 29 | 112.66 | 26.34 |
| granite-3-3-8b-instruct | 0.2473 | 0.1001 | 0.1867 | 0.2079 | 36 | 164.36 | 29.19 |
| **gpt-4.1-2025-04-14** | **0.1628** | **0.0816** | **0.1332** | **0.1389** | 40 | **277.12** | 29.85 |

> **Insight**
>
> In the Agent-As-Tool setting, models generating reasoning traces closer in length to reference answers (e.g., `mistral-large`, `llama-3-3-70b-instruct`) achieve higher ROUGE scores, whereas longer outputs from models like `gpt-4.1` reduce lexical overlap despite potentially valid and rich reasoning, highlighting that ROUGE penalizes verbosity rather than reasoning quality.

Next, Table 29 reports the **average execution scores** per model:

| Model | Average Execution Score |
|---|---|
| meta-llama/llama-3-405b-instruct | 0.118 |
| meta-llama/llama-4-maverick-17b-128e-instruct-fp8 | 0.077 |
| meta-llama/llama-4-scout-17b-16e-instruct | 0.092 |
| ibm/granite-3-3-8b-instruct | 0.040 |
| meta-llama/llama-3-3-70b-instruct | 0.031 |
| mistralai/mistral-large | 0.113 |
| openai-azure/gpt-4.1-2025-04-14 | 0.117 |

Table 29: Average Execution Chain Scores for different LLM models. Scores reflect alignment with ground truth sequences in terms of step name, argument similarity, and sequence coverage.

The results indicate that:

- `meta-llama/llama-3-405b-instruct`, `mistral-large`, and `gpt-4.1-2025-04-14` achieve the highest alignment with ground truth steps, demonstrating better handling of multi-step task execution in the Agent-As-Tool setting.

- Larger models such as `llama-4-maverick` and `llama-4-scout` have moderate scores, suggesting that complexity alone does not guarantee faithful execution.

- Smaller or older models, including `granite-3-3-8b` and `llama-3-3-70b`, exhibit lower scores, primarily due to missing steps, extra steps, or argument discrepancies.

Overall, this evaluation provides a quantitative, interpretable measure of how closely an agent's executed actions match the intended ground truth, complementing other performance metrics such as reference-based scoring (ROUGE) or semantic verification.

> **Execution Insight**
>
> Top-performing models (`llama-3-405b-instruct`, `mistral-large`, `gpt-4.1`) achieve the highest average execution scores, indicating superior alignment with ground-truth multi-step sequences. In contrast, larger models like `llama-4-maverick` show moderate alignment, highlighting that model size alone does not guarantee faithful task execution.

## G    GENERALITY: NEW DATASETS AND SCENARIOS

This section complements the generality discussion presented in Section 5.3. In total 166 scenarios are generated using 4 different datasets to support the generality study:

- We use two public datasets for condition monitoring of industrial assets, hosted on UCI, which provide programmatic access to descriptions and metadata:

    - **Metro Train MetroPT-3 (15 scenarios)**: Created scenarios based on dataset description and failure locations to test failure detection and reasoning.
    - **Hydraulic System (17 scenarios)**: Generated scenarios for hydraulic pumps focusing on early fault identification and operational anomalies.

- We also used internal datasets useful for condition monitoring of an industrial assets and ISO documents for testing an agent that is deployed in production system.

    - **FailureSensorIQ (88 scenarios)**: Identify responsible sensor for early detection of failures.
    - **Asset Health (42 scenarios)**: Assess the condition of an industrial asset based on its recent history.

### G.1    SCENARIO USING METROPT-3 DATASET

The MetroPT dataset[1] is a real-world multivariate time-series dataset collected from the Air Production Unit (APU) of metro trains in Porto, Portugal. It contains readings from pressure, temperature, motor current, and air intake valves were collected from a compressor's Air Production Unit (APU). The dataset includes documented failure events such as air and oil leaks, providing ground truth for predictive maintenance and anomaly detection tasks. MetroPT enables evaluation of IoT agent, FMSR agent and TSFM agent as this dataset is particularly suitable for temporal modeling, early fault detection, and remaining useful life estimation. Its high-resolution, real-world nature makes it a challenging benchmark for testing model robustness, interpretability, and real-time prediction capabilities. With the help of our internal SME, we created 15 complex scenarios and two examples are given in Table 30. We can see the reachness in type of analysis an end user is interested.

Table 30: Sample predictive maintenance scenarios for MetroPT-3 Dataset.

| ID | Scenario Description |
|----|----------------------|
| 1  | Consider asset `mp_1`. After the maintenance performed on May 30, 2020, how has the compressor's condition evolved during May 31 to June 6? Are there any indications that further repair or monitoring is needed? |
| 2  | Consider asset `mp_1`. From the compressor sensor data collected between May 29 and June 4, 2020, can we assess the likelihood of an air leak failure occurring within the subsequent week starting June 5? Is preventive maintenance advisable? |

### G.2    SCENARIO USING HYDROLIC SYSTEM DATASET

The UCI Hydraulic Systems dataset was collected from a lab-scale hydraulic test rig equipped with multiple sensors reporting pressures, volume flows, temperatures, motor power, vibration, and cooling metrics. The rig cycles through constant 60-second loads, while four component fault types (cooler, valve, pump leakage, and accumulator) are varied across severity levels. With 2,205 instances and 43,680 features, the dataset is multivariate and structured for both classification and regression tasks. The condition of each component is annotated per cycle, enabling fault diagnosis and predictive maintenance modeling. With the help of our internal SME, we created 17 complex scenarios and two examples are given in Table 35.

---

[1]https://archive.ics.uci.edu/dataset/791/metropt+3+dataset

Table 31: Sample predictive maintenance scenarios for Hydrolic System Dataset.

| ID | Scenario Description |
|---|---|
| 1 | For asset `hp_1`, can severe internal pump leakage on 2024-01-31 be detected using sensor data from the preceding 100 days? Which sensor trends provide key clues within this timeframe? |
| 2 | Consider asset `hp_1`. At 2024-01-22, can the hydraulic accumulator close to total failure be detected by analyzing sensor data spanning previous days? What sensor signatures confirm this state?", |

### G.3 ASSET HEALTH SCENARIO USING INTERNAL DATASET

Based on business unit requirements and in collaboration with domain experts, we created 42 scenarios for detecting asset health to conduct the benchmark study, primarily using work order data. Each scenario follows the prescribed format described in the main paper. A representative example is shown below:

```
{
  "id": 1000,
  "file": "Air Handling Unit_615152AC_insights_prompt.txt",
  "text": "You are an expert in Air Handling Unit maintenance and
          reliability analysis. Your task is to analyze provided
          asset_details_facts and workorder_facts...",
  "type": "System Health",
  "category": "Asset Analysis",
  "deterministic": true,
  "characteristic_form": "The expected condition of the asset is
  'Not enough data' because only 4 work orders are available."
}
```

One of the task types is **System Health**, aimed at evaluating the condition of an asset based on recent system records (typically work orders, alerts, etc.) and raising flags such as *good* or *needs attention*. Table 32 summarizes the coverage of the 42 scenarios across asset classes.

Table 32: Distribution of scenarios across asset types/classes.

| Asset Type/Class | Number of Unique Instances |
|---|---|
| Air Handling Unit | 9 |
| CRAC | 10 |
| Chiller | 10 |
| Pump | 8 |
| Boiler | 5 |

This diversity spans both horizontal coverage (different asset classes) and vertical variation (multiple instances within each class), providing a robust testbed for evaluating agent generalization and performance across operational conditions. All 42 scenarios fall under the **Asset Health** category and primarily rely on work order information. Each scenario captures distinct aspects of asset behavior, reflecting operational variability. Token count analysis provides insight into scenario complexity.

Over 60% of scenarios (26/42) fall in the 767–2,841 token range, reflecting mostly concise formats. A long-tailed distribution exists to ensure LLMs handle both compact and extended input contexts.

### G.4 FAILURESENSORQA DATASET USING ISO DOCUMENT

The FailureSensorQA dataset is designed to support predictive maintenance reasoning using structured knowledge from ISO standards and industrial asset documentation. Each scenario in the dataset presents a realistic diagnostic question, prompting the agent to identify relevant failure modes

Table 33: Token count statistics for 42 Asset Health scenarios.

| Statistic | Value |
|---|---|
| Total scenarios | 42 |
| Median | 2,277 tokens |
| Mean | 3,695 tokens |
| Standard Deviation | 3,125 tokens |
| Minimum–Maximum | 777–11,098 tokens |
| Mode | 1,316 tokens |

Table 34: Token count distribution across scenarios.

| Token Range | # Scenarios |
|---|---|
| (767 – 2,841] | 26 |
| (2,841 – 4,905] | 6 |
| (4,905 – 6,970] | 1 |
| (6,970 – 9,034] | 3 |
| (9,034 – 11,098] | 6 |

and determine the most informative sensors for early detection. Table 35 shows representative examples, including scenarios for aero gas turbines and compressors, where the task requires mapping sensor readings such as vibration, temperature, or fuel flow to potential failure events. By leveraging ISO-standardized descriptions, this dataset ensures that reasoning aligns with industry practices, enabling evaluation of agent capabilities in sensor-failure attribution and condition monitoring. The dataset emphasizes multi-step reasoning, sensor selection, and domain-specific knowledge integration, providing a challenging benchmark for testing predictive maintenance agents. We generated total 88 scenarios with the help of our reliability enginners.

Table 35: Representative scenarios from the FailureSensorQA dataset.

| ID | Scenario Description |
|---|---|
| 1 | For aero gas turbine, list all the failure modes that can be detected or indicated by abnormal readings from vibration, speed, or fuel pressure/fuel flow.? |
| 2 | Which sensors are most effective for detecting failure modes related to vibration and temperature anomalies in a compressor? |

SCENARIO EXECUTION AND EVALUATION

The 42 scenarios were executed across three models, resulting in 126 executions. Each execution generates an output trajectory, which is subsequently analyzed by the Evaluation Agent across five runs, yielding 630 evaluation instances. The Evaluation Agent compares outputs against the characteristic form described in the scenario examples to calculate automated metrics. Manual review was used to validate the final column of results, identifying only one case (Granite) where the model overconfidently claimed task completion.

PERFORMANCE INSIGHTS

The scenarios primarily assess LLMs' analytical skill—the ability to interpret provided information and generate appropriate conclusions. Agents such as FMSR, which excel in skill-based reasoning tasks, demonstrate strong performance, particularly in single-agent communication settings.

## H   EMERGING FAILURE MODE DISCOVERY AND AGENT DEVELOPMENT

To support adaptive evaluation of multi-agent LLM systems, this appendix outlines the implementation details behind the failure discovery process. While the main text presents the empirical distribution of failure types—including emergent patterns—this appendix focuses on the structured methodology used to extract and cluster novel failure behaviors beyond the MAST (Multi-Agent System Failure Taxonomy) Cemri et al. (2025). The evaluation spanned 881 multi-agent trajectories, drawn from diverse language model configurations. Trajectory distribution by model is as follows:

- `mistral-large`: 145 trajectories
- `llama-3-405b-instruct`: 145 trajectories
- `llama-3-3-70b-instruct`: 145 trajectories
- `llama-4-maverick-17b-128e-instruct-fp8`: 125 trajectories
- `llama-4-scout-17b-16e-instruct`: 111 trajectories
- `gpt-4.1-2025-04-14`: 105 trajectories
- `granite-3-3-8b-instruct`: 105 trajectories

Among the 881 utterance execution trajectories analyzed using an LLM-as-a-judge framework (selected LLM judge model - *openai-azure/gpt-4.1-2025-04-14* as the LLM judge) to identify the causes of multi-agent AI failures, we found that—beyond the existing MAST categories—185 trajectories exhibited one additional failure reason, while 164 trajectories contained two distinct additional failure reasons. This highlights the empirical necessity of taxonomy expansion to capture compound and emergent failure patterns in real-world deployments. To extend the original MAST taxonomy, we conducted a structured analysis of novel multi-agent system failures observed in recent interaction traces. This subsection details our identification methodology and explains how the resulting failure modes align with the MAST framework.

### H.0.1   ALGORITHM FOR EMERGING FAILURE MODES CLUSTERING

To systematically identify and normalize *emerging failure modes* observed in multi-agent LLM system interactions, we introduce a structured algorithmic framework based on semantic embedding and unsupervised clustering. This process abstracts unanticipated failure patterns into representative categories that either align with or extend the predefined MAST taxonomy.

**Definitions and Notation.**   Let:

- $T = \{t_1, \ldots, t_n\}$: Set of multi-agent execution trajectories.
- $\mathcal{M}$: The predefined MAST taxonomy of failure types.
- $F = \{f_1, f_2, \ldots, f_m\}$: Set of *emerging failure mode* descriptions not covered by $\mathcal{M}$, extracted from LLM-as-a-judge evaluations.
- $\phi : \mathcal{S} \to \mathbb{R}^d$: Sentence embedding function (e.g., Sentence-BERT).
- $\mathbf{E} = [\phi(f_1), \ldots, \phi(f_m)]^\top \in \mathbb{R}^{m \times d}$: Matrix of embedded failure descriptions.
- $\mathcal{C} = \{C_1, \ldots, C_k\}$: Partition of $F$ into $k$ clusters, each with centroid $\boldsymbol{\mu}_j$.

**Step 1: Emerging Failure Mode Extraction.**   Each trajectory $t_i \in T$ is evaluated by an LLM-as-a-judge to identify:

- Labeled failure types from the MAST taxonomy $\mathcal{M}$.
- Up to two *emerging* failure descriptions $f_{i1}, f_{i2} \notin \mathcal{M}$.

The full set of novel descriptions is aggregated as:

$$F = \bigcup_{i=1}^{n} \{f_{i1}, f_{i2}\} \setminus \text{NULL}$$

**Step 2: Semantic Embedding.** Each emerging failure mode $f_i \in F$ is transformed into a $d$-dimensional vector:

$$\mathbf{e}_i = \phi(f_i), \quad \forall f_i \in F$$

$$\mathbf{E} = \begin{bmatrix} \phi(f_1)^\top \\ \phi(f_2)^\top \\ \vdots \\ \phi(f_m)^\top \end{bmatrix} \in \mathbb{R}^{m \times d}$$

**Step 3: Optimal Clustering via K-Means.** To discover latent groups of semantically similar failure descriptions, we apply K-Means clustering over the embeddings $\mathbf{E}$. The silhouette score for a given point $i$ is:

$$s(i) = \frac{b(i) - a(i)}{\max\{a(i), b(i)\}}$$

Where:

- $a(i)$: Mean distance from $\mathbf{e}_i$ to other points in the same cluster.
- $b(i)$: Minimum mean distance from $\mathbf{e}_i$ to points in a different cluster.

The optimal number of clusters is selected as:

$$k^* = \arg\max_k \text{SilhouetteScore}(k)$$

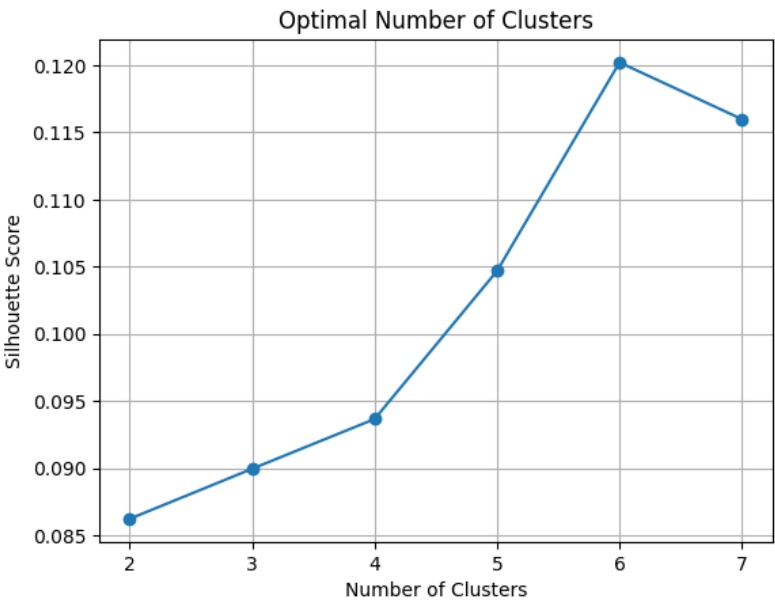

Figure 22: Silhouette analysis showing optimal number of clusters $k^* = 6$.

**Step 4: Cluster Center Selection.** For interpretability, we select a representative $f_j^*$ from each cluster $C_j$ as the most centrally located failure mode:

$$f_j^* = \arg\min_{f_i \in C_j} \|\phi(f_i) - \boldsymbol{\mu}_j\|_2$$

**Step 5: Taxonomy Alignment.** Each representative failure mode $f_j^*$ is reviewed and mapped to one or more MAST categories:

- **Specification Failures**

- **Inter-Agent Failures**
- **Task Verification Failures**

Failures that exhibit characteristics of multiple categories are marked as *compound* or *intersectional*, suggesting the need for extensions to the base taxonomy.

**Outputs.** The algorithm yields:

- A clustered taxonomy $\mathcal{C} = \{C_1, \ldots, C_{k^*}\}$ of emerging failure modes.
- Canonical representatives $\{f_1^*, \ldots, f_{k^*}^*\}$ for each cluster.
- Category mappings for taxonomy refinement or extension.
- Frequency statistics per failure type for prioritization.

H.0.2    METHODOLOGY: SEMANTIC CLUSTERING OF EMERGENT FAILURES

Building on the formal clustering algorithm outlined above, we implemented a practical instantiation of the pipeline to organize the large volume of emerging failure mode descriptions identified by the LLM-as-a-judge. We found lots of new and different behaviors when we first looked. But a closer look showed that many of them were either just repeating the same idea or were only slightly different versions of the same core problems. To distill these into interpretable categories, we applied a semantic clustering methodology grounded in high-dimensional language representations.

Each emerging failure description was manually or programmatically summarized into a concise label and explanatory text. These summaries were then embedded into a semantic vector space using the `all-MiniLM-L6-v2` model from the SentenceTransformer library, yielding a set of dense, comparable embeddings suitable for clustering.

We applied the KMeans algorithm to group these embeddings into semantically coherent clusters. To determine the optimal number of clusters, we computed silhouette scores for values of $k$ ranging from 2 to 7 and selected the value that maximized mean silhouette score (see Figure 22). This analysis yielded an optimal configuration of $k^* = 6$ clusters.

For interpretability, each cluster was assigned a canonical label derived from the failure mode description closest to the cluster centroid. This process produced six representative categories of emerging failure modes, summarized below:

- *Cluster 0: Lack of Error Handling for Tool Failure* (53 cases, 10.3%)
  Agents fail to detect or appropriately respond to tool invocation errors.
- *Cluster 1: Failure to Incorporate Feedback* (41 cases, 8.0%)
  Agents ignore or inadequately adjust to feedback from other agents or tools.
- *Cluster 2: Invalid Action Formatting* (27 cases, 5.3%)
  Output includes syntactic or structural errors that prevent execution.
- *Cluster 3: Overstatement of Task Completion* (122 cases, 23.8%)
  Agents claim completion without satisfying task criteria or producing valid outcomes.
- *Cluster 4: Extraneous or Confusing Output Formatting* (110 cases, 21.4%)
  Responses contain unnecessary verbosity, ambiguous structure, or misleading formatting.
- *Cluster 5: Ineffective Error Recovery* (160 cases, 31.2%)
  Agents fail to resolve prior mistakes or restart workflows effectively after failure.

These cluster-derived failure modes serve as canonical extensions to the base MAST taxonomy, revealing previously unclassified behaviors that frequently arise in multi-agent LLM interactions. Their emergence underscores the value of inductive, embedding-based clustering for scalable failure mode discovery and taxonomy refinement.

H.0.3    TAXONOMIC ALIGNMENT WITH MAST OF EMERGENT FAILURES

These emergent failure modes reveal both alignment and tension with the original MAST taxonomy. Each cluster can be mapped to one or more of MAST's three core failure categories, but many straddle boundaries or reveal overlapping failure dynamics:

- **Specification Failures:**

    - *Overstatement of Task Completion* and *Extraneous Output Formatting* reflect unclear success criteria, misunderstood task scopes, or ambiguous output specifications.

- **Inter-Agent Failures:**

    - *Failure to Incorporate Feedback* and *Lack of Error Handling for Tool Failure* indicate coordination breakdowns or limited adaptivity in dynamic environments.

- **Task Verification Failures:**

    - *Invalid Action Formatting* and *Ineffective Error Recovery* highlight failures in runtime execution monitoring, verification, and correction procedures.

Several emergent failure types cut across multiple categories, underscoring the complexity and interdependence of failure dynamics in real-world multi-agent systems. These findings motivate future refinement of MAST to support cross-category failure representation and compound behavior tracking.

This failure mode analysis contributes both methodologically and substantively to multi-agent system evaluation. Methodologically, it introduces a scalable pipeline for inductively discovering and structuring new failure behaviors using LLM-judged outputs and semantic clustering. Substantively, it extends the empirical coverage of the MAST taxonomy by surfacing nuanced, real-world failure patterns that reflect the increasing complexity of autonomous agent collaboration.

These insights not only validate the need for flexible taxonomic frameworks but also point to the importance of diagnostics that evolve with model behavior. As LLM-based agents continue to scale in capability and deployment scope, the ability to detect emergent, intersectional failures becomes a foundational requirement for reliable multi-agent orchestration.

## H.1 Impact of Agent Communication on Benchmark Performance

In our benchmark, the parameter `enable_agent_ask` controls whether the agent can ask clarifying questions during task execution. In the Agent-As-Tool architecture, planning is performed incrementally, and agent communication can influence task performance, unlike the Plan-Execute paradigm where planning is done upfront.

For a fair comparison, our initial experiments used the default setting (`enable_agent_ask=False`), preventing agents from asking questions beyond the given task. Table 2 highlights that certain failures, such as not asking clarifying questions, contribute to approximately 10% of errors. To evaluate the impact of agent communication, we set `enable_agent_ask=True` and re-ran the experiments across multiple models. Table 36 summarizes the results.

Table 36: Benchmark performance with and without agent communication enabled.

| Model | enable_agent_ask=True | enable_agent_ask=False |
|---|---|---|
| gpt-4.1-2025-04-14 | 63% | 65% |
| lama-4-maverick | 66% | 59% |
| llama-3-405b-instruct | 61% | 44% |
| mistral-large | 58% | 40% |
| llama-3-3-70b-instruct | 35% | 40% |
| granite-3-3-8b-instruct | 32% | 35% |

These results indicate that enabling agent communication improves performance substantially for certain models (e.g., LLaMA-4 Maverick and LLaMA-3 405b), likely due to better multi-turn handling and the ability to clarify ambiguous information. For other models, performance is less sensitive to this parameter.

This experiment offers a compelling insight, highlighting the impact of hidden architectural features on benchmark results. Furthermore, it demonstrates that our benchmark can capture subtle

differences in agent behavior and encourages transparent reporting of configuration parameters for reproducibility.

> **Agent Communication Insight**
>
> Enabling `enable_agent_ask` significantly boosts performance for models capable of multi-turn reasoning (e.g., `llama-4-maverick`, `llama-3-405b-instruct`), demonstrating that agent communication can resolve ambiguities and improve task execution. Other models show minimal sensitivity, highlighting differences in internal reasoning and multi-step handling capabilities.

# I WORKFLOW ILLUSTRATION

This subsection will later be merged into Appendix Section A. We are keeping it here temporarily so that references to figure numbers in the rebuttal do not need revision. The complexity of real-world problem-solving often exceeds the capabilities of a single, monolithic single LLM based AI Agent. To address multi-faceted challenges, especially those requiring interaction with external systems, specific knowledge retrieval, or sequential decision-making, advanced agentic architectures have been developed. This section formalizes two leading paradigms: the **Agent-As-Tool** approach, which focuses on dynamic, iterative reasoning and acting by calling specialized components; and the **Plan-Execute** framework, which emphasizes structured planning, dependency management, and context-aware execution to ensure traceable and reliable task completion.

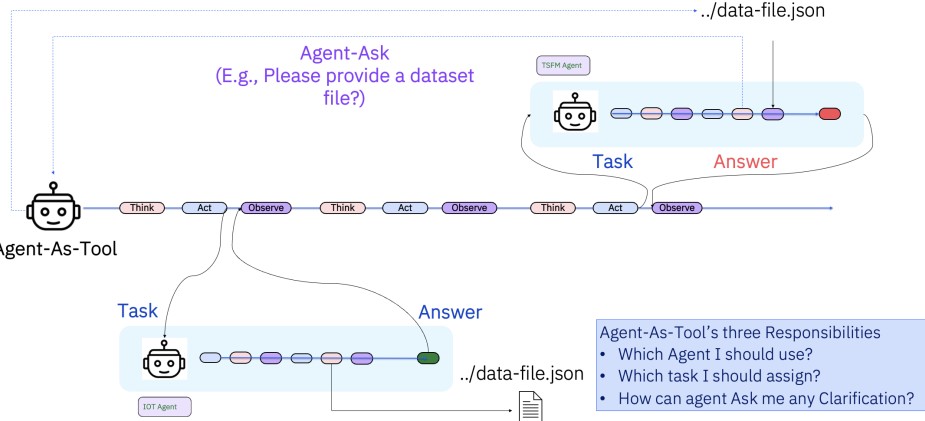

Figure 23: Typical Workflow: Agent-As-Tool using Agent-Ask

## I.1 AGENT-AS-TOOL WORKFLOW

The **Agent-As-Tool** paradigm, coupled with the Agent-Ask mechanism, offers a robust framework for complex task execution by decomposing a problem and routing sub-tasks to specialized AI agents. As illustrated in the workflow Figure 23, the primary Agent-As-Tool receive a user query. The Agent-As-Tool has to fulfill three core responsibilities: (1) determining the appropriate specialized agent (e.g., the TSFM Agent or IoT Agent) for a given sub-task, (2) correctly assigning the task, and (3) facilitating clarification from the specialized agent if needed. Once the Agent-As-Tool receives an input query, it enters a standard Think-Act-Observe loop, and it decides on the appropriate specialized agent in think step. The specialized agent executes its designated task and provides an Answer. Answer is embedded inside to the orchestration agent's observation for next set of action. In this case, an artifact like a data-file.json, back to the main orchestrator, allowing the overall system to complete complex, multi-faceted operations that exceed the capability of any single agent. This architecture highlights the benefits of modularity and specialized expertise in large-scale language model systems.

Algorithm 2 details an iterative, multi-turn execution framework akin to the ReAct (Reasoning and Acting) paradigm. This framework enables a central agent to solve a complex User Task ($\mathcal{Q}$) by strategically engaging specialized agents ($\mathcal{A}_{spec}$) within a bounded number of steps $T_{max}$. The process begins with the initialization of the agent's internal state $M$ and an empty execution history $\mathcal{H}$. The agent then enters an iterative loop where a thinking policy $\Pi_{think}(M)$ determines the next action (either THINK-ACT or FINISH) and the continuation signal. When the action is THINK-ACT, the agent selects the best agent_id via $\Pi_{select}$ and formulates a precise sub_task via $\Pi_{formulate}$ for execution. Conversely, if the action is FINISH, the agent summarizes the full $\Pi$, updates the memory, and terminates the loop. Following any action, the resulting output is compressed into a concise observation ($\Pi_{compress}$) to manage context length. This observation is logged to $\mathcal{H}$, and critically, the agent's internal memory $M$ is updated with this new context, driving the decision-making in the subsequent round. Finally, once the loop terminates, a policy $\Pi_{final}$ generates the complete Final Output ($O$) by synthesizing the entire execution history $\mathcal{H}$.

---

**Algorithm 2:** Agent-As-Tool (Simplified as ReAct)

---

**Input** : User Query $\mathcal{Q} \in \mathcal{Q}$, Maximum Steps $T_{max}$, Set of Agents $\mathcal{A}_{spec} \subseteq \mathcal{A}$
**Output:** Final Output $O \in \mathcal{O}$, Execution Plan $\Pi$

---

$M \leftarrow$ InitializeMemory($\mathcal{Q}$) //Initialize the global Memory System $M$
$\Pi \leftarrow \emptyset$;
$\mathcal{H} \leftarrow \emptyset$;
**for** $t = 1$ **to** $T_{max}$ **do**
    $(continue, \text{action}) \leftarrow \Pi_{think}(M)$ //Agent decides next step using
        current Memory $M$
    **if** $action = THINK\text{-}ACT$ **then**
        agent_id $\leftarrow \Pi_{select}(\text{action}, M, \mathcal{A}_{spec})$ //Select best Agent $A_i \in \mathcal{A}_{spec}$
        $\tau \leftarrow \Pi_{formulate}(\text{action}, M)$ //Formulate sub-task $\tau \in \mathcal{T}$ based on
            current Memory $M$
        output $\leftarrow$ ExecuteAgent(agent_id, $\tau$) //Agent executes task and returns
            structured output $o \in \mathcal{O}$
        $\Pi \leftarrow \Pi \cup \{\langle \tau, \text{agent\_id} \rangle\}$ //Update the execution plan $\Pi$ with
            task-agent assignment
    **else if** $action = FINISH$ **then**
        final_output $\leftarrow$ Summarize($\Pi$);
        $M \leftarrow$ UpdateMemory($M$, final_output);
        **break**;
    observation $\leftarrow \Pi_{compress}(\text{output})$ //Generate concise observation
        summary (optional
    $\mathcal{H} \leftarrow \mathcal{H} \cup \{(t, \text{action}, \text{observation})\}$ //Log step to history
    $M \leftarrow$ UpdateMemory($M$, observation) //Update Memory $M$ with new
        context from observation
    **if** $continue = False$ **then**
        **break** //Stop if agent decides termination

$O \leftarrow \Pi_{final}(\mathcal{H})$ //Produce final answer from the complete history
**return** $(O, \Pi, \mathcal{H})$

---

## I.2 PLAN-EXECUTE

A diagram in Figure 9 illustrates a **Plan-Execute** approach to addressing a complex industrial query, such as "discover the most relevant sensor for Chiller 6 at POKMAIN site for detecting Compressor Overheating failure?". The process begins with the main agent receiving the Query and formulating a detailed Plan. This plan is meticulously broken down into sequential steps. For instance, Step 1 involves a Task to "Identify the sensors available for Chiller 6 at POKMAIN site" and specifies Agent 1 (e.g., an IoT Data Download Agent) to execute this task, with an Expected Output of "A list of sensors available for Chiller 6 at POKMAIN site." Following this, Step 2 takes this output as a Dependency (#S1) to execute the Task: "Determine which of these sensors can detect Compressor Overheating failure," assigning it to a specialized Agent 2 (e.g., a FMSR Agent).

After the Plan is reviewed, it is translated into a dynamic, dependency-aware Workflow represented as a directed graph. This graph outlines the logical flow and potential parallel execution paths (e.g., tasks 3, 4, and 5 running concurrently) based on the sequential nature of the task dependencies. The Context-aware Execution phase then involves a specialized execution engine that manages these tasks, tracking their state, inputs (like the JSON objects and strings containing the intermediate results), and dependencies between agents. For example, the output of the first stage (ID 1) becomes a structured input for subsequent tasks (ID 2), ensuring that information is seamlessly and accurately passed between the specialized agents. The entire process culminates in a Result Summary that provides the final, actionable answer to the initial complex query. This methodology ensures traceability, modularity, and the effective integration of multiple specialized AI agents for industrial problem-solving.

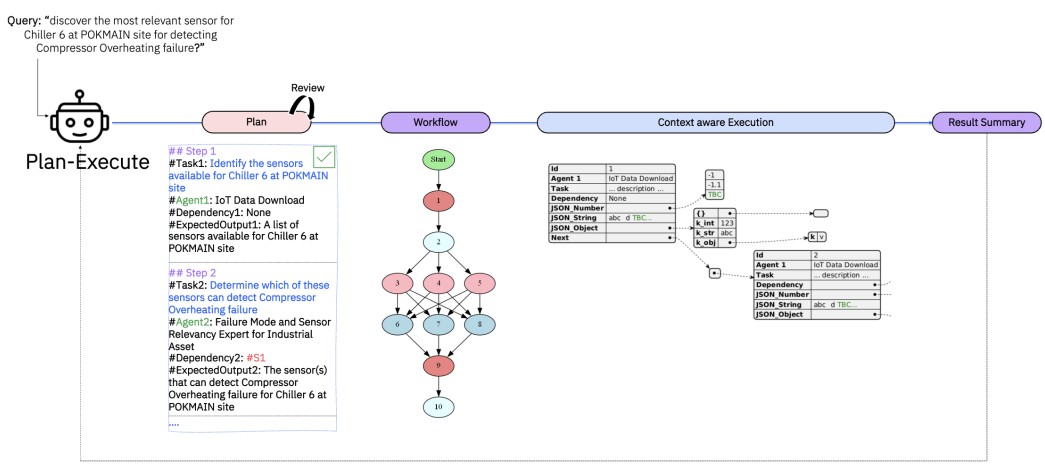

Figure 24: Typical Workflow: Plan-Execute Approach using Review

---

**Algorithm 3:** Plan-Execute with Context-Aware Workflow (Sequencial Workflow)

**Input** : User Query $\mathcal{Q}$, Specialized Agents $\mathcal{A}_{spec}$

**Output:** Final Output $O \in \mathcal{O}$

**Phase 1: Planning and Decomposition**;

Plan $\leftarrow \Pi_{plan}(\mathcal{Q})$ //Generate a list of sequential steps with dependencies

Plan $\leftarrow$ DecomposeToSteps(Plan);

Workflow $\leftarrow$ BuildDAG(Plan) //Translate steps into a Directed Acyclic Graph (DAG)

Context $\leftarrow$ InitializeContext($\mathcal{Q}$);

TaskSequence $\leftarrow$ TopologicalSort(Workflow) //Generate a sequence ensuring dependencies are met

**Phase 2: Context-Aware Execution**;

**foreach** *task* $\in$ *TaskSequence* **do**

    dependencies $\leftarrow$ GetDependencies(task);

    InputContextData $\leftarrow \emptyset$;

    **foreach** *dep_id* $\in$ *dependencies* **do**

        data $\leftarrow$ Context[dep_id] //Retrieve output of dependent task from Context

        InputContextData $\leftarrow$ InputContextData $\cup$ {data}

    FullQuery $\leftarrow$ task.description + FormatContext(InputContextData) //Combine task description with necessary context data

    ;

    agent $\leftarrow$ AssignAgent(task, $\mathcal{A}_{spec}$) //Assign the specialized agent (e.g., Failure Mode Expert)

    **Execute:**;

    RawOutput $\leftarrow$ agent.Execute(FullQuery);

    **Context Management:**;

    StructuredOutput $\leftarrow$ FormatToJSON(RawOutput) //Standardize output for downstream use

    Context $\leftarrow$ UpdateContext(Context, StructuredOutput) //Store result as a labeled output (e.g., #S1)

    **Update Workflow:**;

    Workflow $\leftarrow$ MarkCompleted(Workflow, task) //Mark task as completed in the DAG

**Phase 3: Summarization**;

$O \leftarrow \Pi_{summary}(\mathcal{Q}, \text{Context})$ //Generate final, high-level summary using all collected context

**return** $O$

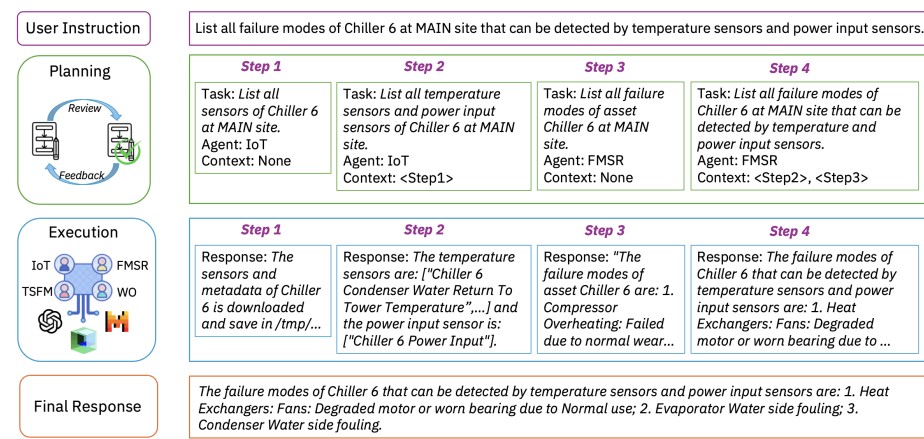

Figure 25: Plan-Execution Workflow Concrete Example.

Algorithm 3 describes a structured, three-phase approach for complex queries, prioritizing explicit planning and efficient data flow management. This approach ensures a systematic resolution of the User Query ($\mathcal{Q}$). The first phase, **Planning and Decomposition**, transforms $\mathcal{Q}$ into an execution structure. A planning policy $\Pi_{plan}$ generates a sequential Plan, which is then compiled into a Directed Acyclic Graph (Workflow) defining task dependencies. The Context is initialized, and a TaskSequence is generated by topologically sorting the Workflow, which guarantees that tasks are processed only after their dependencies have been met. The second phase, **Context-Aware Execution**, manages the task flow based on the TaskSequence. For each task in the sequence, the algorithm retrieves *only* the necessary context. This process involves iterating through all task dependencies to collect the required data from the global Context. This collected data is then formatted and combined with the task.description to form the FullQuery, which is executed by the assigned agent. The resulting RawOutput is immediately standardized via FormatToJSON and stored back into the global Context (e.g., as #S1), making it available for subsequent tasks. The final phase, **Summarization**, occurs upon completion of all tasks. A final summarization policy ($\Pi_{summary}$) synthesizes the definitive Final Output ($O$) from the original $\mathcal{Q}$ and the comprehensive Context.

### I.3 DETAILED WORKFLOW EXAMPLES

Figure 25 presents a concrete example of the Plan-Execute workflow for the user query: "List all failure modes of Chiller 6 at the MAIN site that can be detected by temperature sensors and power-input sensors." The planning stage produces four steps, illustrated in the middle row, each outlining a specific sub-task derived from the original query. The execution stage then follows these steps in sequence, generating intermediate outputs and ultimately producing the final answer. This example highlights how the Plan-Execute approach breaks down a complex request into structured actions and systematically retrieves the required information.

## J   MODEL PERFORMANCE AND PLANNING ANALYSIS

We extended our evaluation to include models that were not part of the original benchmark, specifically Anthropic Claude variants (`claude-3-7-sonnet`, `claude-4-sonnet`) and GCP Gemini (`gemini-2.5-pro`). Model performance was evaluated using planning accuracy metrics (BERTScore, ROUGE, and alignment with ground-truth plans), consistent with our original execution-accuracy leaderboard. We also analyzed planned step statistics to understand model behavior in generating task plans.

### J.1 COMBINED SCORE (BERTSCORE + ROUGE-L)

For each model, we compute a *combined planning score* $S_{\text{combined}}$ that integrates BERTScore ($B$) and ROUGE-L ($R$):

$$S_{\text{combined}} = \frac{B + R}{2},$$

where $B \in [0, 1]$ is the average BERTScore between the model-generated plan and the ground-truth plan, and $R \in [0, 1]$ is the average ROUGE-L F1 score. This provides a balanced measure of both semantic similarity (via BERTScore) and sequence-level overlap (via ROUGE-L).

### J.2 PLANNING ACCURACY

| Model Name | Avg ± Std | Questions |
|---|---|---|
| mistral-medium-2505 | 0.620 ± 0.063 | 141 |
| gemini-2.5-pro | 0.615 ± 0.068 | 141 |
| gpt-oss-120b | 0.606 ± 0.077 | 141 |
| mistral-small-3-1-24b | 0.604 ± 0.062 | 141 |
| claude-4-sonnet | 0.595 ± 0.068 | 141 |
| llama-3-405b-instruct | 0.588 ± 0.074 | 141 |
| claude-3-7-sonnet | 0.571 ± 0.071 | 141 |
| llama-4-maverick-17b | 0.558 ± 0.071 | 141 |
| gpt-5-2025-08-07 | 0.544 ± 0.092 | 141 |
| granite-3-3-8b-instruct | 0.529 ± 0.067 | 141 |
| llama-3-3-70b-instruct | 0.522 ± 0.068 | 141 |

Table 37: Planning accuracy (Avg ± Std) for all evaluated models.

### J.3 PLANNED STEP STATISTICS

| Model Name | Avg Steps ± Std | Min | Max | Zero Steps |
|---|---|---|---|---|
| llama-3-405b-instruct | 3.14 ± 1.84 | 1 | 9 | 0 |
| llama-3-3-70b-instruct | 6.55 ± 1.55 | 3 | 12 | 0 |
| llama-4-maverick-17b | 4.34 ± 1.80 | 1 | 9 | 0 |
| granite-3-3-8b-instruct | 5.56 ± 2.44 | 2 | 30 | 0 |
| gpt-oss-120b | 1.91 ± 1.21 | 1 | 10 | 0 |
| mistral-medium-2505 | 2.38 ± 1.04 | 1 | 5 | 0 |
| mistral-small-3-1-24b | 2.77 ± 1.33 | 1 | 6 | 0 |
| claude-3-7-sonnet | 3.10 ± 1.15 | 1 | 5 | 0 |
| gpt-5-2025-08-07 | 2.33 ± 1.16 | 0 | 5 | 1 |
| gemini-2.5-pro | 1.87 ± 1.01 | 1 | 5 | 0 |
| claude-4-sonnet | 2.45 ± 1.34 | 1 | 5 | 0 |

Table 38: Planned step statistics for all evaluated models.

### J.4 KEY INSIGHTS

- **Top-performing models:** `mistral-medium-2505` achieves the highest planning accuracy (0.620 ± 0.063) and produces concise, low-variance plans (2.38 ± 1.04 steps), combining high performance with efficiency. `gemini-2.5-pro` is also highly competitive (0.615 ± 0.068).

- **Anthropic Claude models:** Both `claude-3-7-sonnet` and `claude-4-sonnet` show solid planning accuracy with moderate plan lengths ( 2.5–3 steps) and low variance, indicating reliable reasoning and execution alignment.

- **Instruction tuning matters:** Medium-sized instruction-tuned models (`mistral-medium`, `mistral-small`) consistently produce efficient plans with low variance, outperforming larger models with longer, more variable plans (`llama-3-3-70b`, `granite-3-3-8b`).

- **Step length vs. performance:** Shorter plans with low variance generally correlate with higher planning accuracy, while overly long plans may introduce redundancy without improving alignment with ground-truth executions.
- **Consistency vs. variability:** High-variance models (`gpt-5-2025-08-07`, `granite-3-3-8b`) occasionally generate very long or empty plans, which may reduce reliability despite moderate average scores.

**Conclusion:** The unified analysis demonstrates that medium-sized, instruction-tuned models offer the best balance of planning accuracy and step efficiency, while the inclusion of Claude and Gemini models extends benchmark coverage and validates performance trends. These results are consistent with our original execution-accuracy leaderboard, confirming the robustness of the benchmark for evaluating reasoning-capable language models.

