# OpenReview forum: "AssetOpsBench: Benchmarking AI Agents for Task Automation in Industrial Asset Operations and Maintenance"
_ICLR.cc/2026/Conference — Submitted to ICLR 2026_

### Official Review · Reviewer_1z7t · 2025-10-20

**Soundness:** 2
**Presentation:** 1
**Contribution:** 2
**Rating:** 2
**Confidence:** 4

**Summary:**

This paper introduces AssetOpsBench, a benchmark for evaluating AI agents on task automation in asset operations and maintenance. The authors argue that existing benchmarks do not adequately address the unique challenges of this domain, such as the diverse data modalities and the complex, collaborative nature of tasks. The paper curates a new multi-source dataset from data center operations , a set of 141 human-authored, intent-driven scenarios , a catalog of domain-specific agents (IoT, FMSR, TSFM, and WO) , and a simulated environment for evaluation. The paper conducts a comparative analysis of two multi-agent architectures, "Agent-As-Tool" and "Plan-Executor", and employs a rigorous three-pronged evaluation methodology combining an LLM-based rubric, reference-based scoring, and manual expert verification. A key finding is that the Agent-As-Tool approach generally outperforms Plan-Execute in task quality, though at a higher computational cost, and the work also presents a systematic procedure for identifying emerging failure modes in these agent systems.

**Strengths:**

- The paper studies an important problem of deploying LLM Agents in asset management.

**Weaknesses:**

- There is an unverified claim in the introduction. On line 94, the paper said “domain complexity in industrial settings necessitates a multiagent approach. ”. However, the authors did not provide a direct comparison between multi-agent and single-agent approaches to empirically prove the necessity of a multi-agent approach for handling domain complexity.

- The paper is not very organized. For example, in section 3, you first talk about AssetOpsBench but then start to discuss AssetOps Agent in the next paragraph. It is unclear why you need to discuss the proposed framework in problem definition. The second paragraph does not help understand the definition of your problem. More importantly, the symbols like \tau, \pi introduced in this paragraph are never used again, which makes them not very meaningful.

- The results section lacks insights. Most results are well-established across most all modern LLM benchmarks. (e.g. proprietary LLMs outperform open-source counterparts. Removing in-context examples would decrease performance) The paper could be strengthened if dive deeper into some deep analysis. For example,

	- Why does Plan-Execute fail? The paper speculates about "over-planning" or "brittle plans" but provides no concrete examples. It doesn't show a failing plan from Plan-Execute and contrast it with a successful "thinking" trace from Agent-As-Tool for the same task
	- Direct comparisons with single-agent system
	- Why adding distractors improve performance? why would it trigger more deliberate reasoning in LLM?

- The paper is not easy to read due to the high density of acronyms and jargon. Additionally, there are too many places where the authors refer the readers to the appendix.

- While the creation process of the benchmark is solid, the paper lacks a critical, quantitative assessment of the data's realism. The paper would be more sound if some user study or human evaluation on the data realism can be shown.

- Typos and wrong citation issues:
	- Line 56 “triggering acti mkons.”
	- Line 87 incomplete sentence?
	- Almost all the citation in \cite should be in \citep
	- Some of the citations are wrong. For example, on line 62 you are citing “agent-based systems”
	- There are some other related work missed by the paper. For example,
TheAgentCompany evaluates LLM agents in addressing long-horizon software-engineering tasks. CRMArena-Pro tackles LLM agents in work environments but focuses on the CRM domain.

**Questions:**

See above suggestions on deeper analysis

---

> ### Author Response · Authors · 2025-11-25
> **Response to Weakness - Part 1**
>
> We thank the reviewer for all the useful and constructive comments. Our review is organized in the order of the weaknesses pointed out in the above review. We have provided a point-by-point response to each weakness and the remediation we implemented.  Please download a revised copy of the paper as we do refer to some sections in the revised paper.
>
> ---
>
> ### W1. There is an unverified claim in the introduction. On line 94, the paper said “domain complexity in industrial settings necessitates a multiagent approach. ”. However, the authors did not provide a direct comparison between multi-agent and single-agent approaches to empirically prove the necessity of a multi-agent approach for handling domain complexity.
>
> A1.
>
> Thank you for highlighting this important point. We first provide a clarification, followed by empirical evidence added in the revised manuscript to support the claim.
>
> **Clarification**
>
> When we made the above statement, we were referring to `Figure 1`, which illustrates the data complexity and application diversity inherent in industrial domains. These factors create strong motivation to build separate agentic systems tailored to application-specific nuances. In such complex ecosystems, it is often preferable to design isolated agents around each application and enable automated workflows across them. This design philosophy is also reflected in the growing popularity of the Model Context Protocol (MCP) ecosystem. Similar architectural ideas are emerging in parallel work such as MCP-Universe[1] and MCPBench[2]. We clarify this rationale explicitly in the revised manuscript.
>
> **Empirical Verification of Single-Agent vs Multi-Agent**:
>
> `Yes, we did it`. The single-agent system serves as a baseline in our work and is straightforward to implement, as shown in `Figure 6`. To build a single agent capable of handling all tasks, we aggregated all tools from the individual agents into a single unified agent (see the full tool list in `Table 5`), along with two additional tools for code generation and execution. The results of these experiments are discussed in our response to the `deep-analysis weakness (W3)`. Empirically, the multi-agent system consistently outperforms the single-agent baseline, supporting our claim that a multi-agent approach is better suited for handling the domain complexity present in industrial settings.
>
> [1] MCP-Universe: Benchmarking Large Language Models with Real-World Model Context Protocol Servers
>
> [2] MCP-Bench: Benchmarking Tool-Using LLM Agents with Complex Real-World Tasks via MCP Servers
>
> ---
>
> ### W6. Typos and wrong citation issues:
>
> A6.
>
> We have corrected all typographical errors and fixed the citation issues noted by the reviewers. Additionally, we incorporated two more related-work entries, focused specifically on benchmarking perspectives (rather than domain- or scenario-specific work), to better position our contributions within the broader evaluation landscape.

---

> ### Author Response · Authors · 2025-11-25
> **Response to Weakness - Part 2**
>
> ---
>
> ### W2. The paper is not very organized. For example, in section 3, you first talk about AssetOpsBench but then start to discuss AssetOps Agent in the next paragraph. It is unclear why you need to discuss the proposed framework in problem definition. The second paragraph does not help understand the definition of your problem. More importantly, the symbols like \tau, \pi introduced in this paragraph are never used again, which makes them not very meaningful.
>
> A2.
>
> Thank you for the suggestion. We have revised the structure of `Section 3` to address this concern. Specifically, we updated the title of `Section 3` (from `Problem Definition` to `PROBLEM AND APPROACH`) to make it explicit that the section covers both the problem and two aspects of our methodology:
> - the proposed framework shown in `Figure 2(a)`, and
> - the taxonomy for intent-aware scenario generation shown in `Figure 2(b)`.
>
> We have removed the mathematical notation to simplify the writing and consolidated the detailed agent design and descriptions into `Appendix A : AGENTIC SYSTEM DEFINITION`. These changes improve the clarity of `Section 3` and ensure that all framework-related details are located in the appropriate section. Additionally, we made similar cosmetic adjustments in `Section 5.3` to better showcase the generality experiments of the proposed framework.

---

> ### Author Response · Authors · 2025-11-25
> **Response to Weakness : Part 3 (Deep Analysis - I)**
>
> W3. The results section lacks insights. Most results are well-established across most all modern LLM benchmarks. (e.g. proprietary LLMs outperform open-source counterparts. Removing in-context examples would decrease performance) The paper could be strengthened if dive deeper into some deep analysis. For example,
>
> A3.
>
> Thank you for the valuable feedback. The experimental section highlights several insights that are not well-established and suggest directions for further investigation. Key results are:
>
> 1. **Agent-As-Tool consistently outperforms Plan-Execute.** To our knowledge, this is the first experimental study demonstrating this effect in greater details (**Figure 4**).
>
> 2. **The best-performing GPT model under Agent-As-Tool performs poorly under Plan-Execute** (**Figure 4**). Open-source models can outperform closed models in this setting, indicating closed models may require introspection strategies.
>
> 3. **Reference-based Scoring**, a new metric introduced here, helps explain poor GPT performance and provides a more grounded assessment of correctness (`Section 5.1` and `Appendix Section E with new pseudocode`).
>
> 4. **Adjusting the agent’s communication style based on failure-mode analysis allows open-source models to outperform GPT** under Agent-As-Tool, demonstrating the value of trajectory-driven adaptation.  (`Section 5.2`)
>
> 5. **Overall performance reaches ~60%**, indicating progress while highlighting room for improvement on complex industrial automation tasks. (`Figure 4`)
>
> Additionally, we provide insights for each experiment, including:
>
> - Why distractors can increase performance in LLaMA models. (`Ablation Study in Section 5.1, Appendix F.8.1 DISTRACTOR AGENTS`)
>
> - Opportunities for hybrid combinations of small (SLM) and large (LLM) language models.
>
> These analyses provide a deeper understanding beyond well-known LLM benchmarks, offering actionable insights for agent design and strategy in industrial settings.
>
> We have slightly adjusted the flow in `Section 5` to make the above insights the center of discussion. These findings emphasize that performance differences are not merely model-driven but are strongly influenced by agent design choices, interaction style, and failure analysis.
>
> ---
>
> Now we provide an answer to your specific Question.
>
> Q: Why does Plan-Execute fail? The paper speculates about "over-planning" or "brittle plans" but provides no concrete examples. It doesn't show a failing plan from Plan-Execute and c...
>
> A: Plan-Execute did not fail, it performed pretty well when compared to some top-performing models; however, there are a few model where Plan-Execute perform better than the Agent-As-Tool (`See Figure 21 in Appendix`), where llama-maverick and mistral outperform gpt-4.1 for plan-execute.
>
> Plan-Execute could incorporate iterative review-revise loops to improve plans. However, our results show that fundamental architectural constraints of the Plan-Execute architecture limit its effectiveness in industrial settings. Specifically, Plan-Execute produces shorter, less detailed plans (`typically 2–3 steps versus 5–6 steps`) and cannot adapt once execution begins (unlike Agents-as-Tools). Many AssetOpsBench tasks are inherently reactive. For example, they require interpreting intermediate results (e.g., detecting a sensor reading of all zeros, the failure cannot be detected based on the current available sensors) and adjusting actions based on real-time observations. This creates a mismatch: Plan-Execute prioritizes efficiency through upfront planning, while industrial asset management demands flexibility and adaptive decision-making. We have a small section in `Section 5.2` and a detailed `Appendix Section F.4`.
>
> These findings align with recent studies (e.g., AOP[6]) showing that reactive, feedback-driven approaches more effectively handle task decomposition than single-agent upfront planning. While Plan-Execute could, in theory, be enhanced with dynamic feedback or plan-revision mechanisms, doing so would break the architecture and reduce its efficiency advantage.
>
> Here is a comparison table summarizing the two agentic systems, helping reviewers interpret the key differences in their performance.
>
>
> | Dimension          | Plan-Execute                 | Agents-as-Tool                |
> |-------------------|------------------------------|-------------------------------|
> | Planning           | Upfront, comprehensive       | Incremental, adaptive, reactive     |
> | Efficiency         | Fewer steps, lower runtime   | More steps, higher runtime    |
> | Flexibility        | Limited (plan is fixed)      | High (adapts to observations)|
> | Performance        | 38–46% task completion       | 59–65% task completion        |
> | Best for           | Well-defined, structured tasks | Complex, reactive, multi-step tasks |
>
>
> [1] TaskBench: Benchmarking Large Language Models for Task Automation
>
> [2] Benchmarking Multi-Agent Architectures, Langchain blog, 2025

---

> ### Author Response · Authors · 2025-11-25
> **Response to Weakness : Part 3 (Deep Analysis - II)**
>
> ---
>
> Q2. Direct comparisons with single-agent system
>
> A2.
>
> We run a default LLM, llama-4-maverick, on all 141 scenarios. As a single-agent baseline, it achieves task completion of 26.95\%, data retrieval accuracy of 34.04\%, and generalized result verification of 28.37\%. Under the Agent-As-Tool setup, the same model achieves roughly a 2-fold improvement (`Figure 4`).} We added a small discussion in Section 5.1 with the title (`Baseline using Single-Agent`). These results also align with the discussion in related work (Paper [2]).
>
> ---
>
> Q3. Why adding distractors improve performance? why would it trigger more deliberate reasoning in LLM?
>
> A3.
>
> We included distractor agents primarily from an ablation perspective, with two objectives: (1) to evaluate the base agent’s ability to perform planning and reasoning under potentially confusing inputs, and (2) to conduct a limited scalability test, extending the setup from 5 to 15 agents. Similar experiments have been reported in recent multi-agent benchmarks [2], which also observe that adding distractor agents does not degrade performance and can even improve outcomes in certain scenarios.
>
> This is an important observation, as it highlights the system prompt’s ability to generate robust plans and the LLM’s capacity to make effective decisions even in the presence of irrelevant or distracting agents. Intuitively, distractors may encourage the model to focus more on identifying the relevant agents and critical steps, indirectly improving task performance in some cases.
>
> In the main paper, we found that we did not include a detailed quantitative performance analysis. Below, we provide a more comprehensive breakdown across multiple models and tasks which we will add to revised manuscript in `Appendix Table 25`:
>
> | Model | Task Completion (No distractors / With distractors / Δ) | Data Accuracy (No / With / Δ) | Result Verification (No / With / Δ) |
> |-------|-------------------------------|-----------------------------|---------------------------------|
> | gpt-4.1-2025-04-14 | 52.5 / 48.5 / -4.0 | 57.6 / 56.6 / -1.0 | 55.6 / 54.5 / -1.1 |
> | granite-3-3-8b-instruct | 40.4 / 40.4 / 0.0 | 44.4 / 44.4 / 0.0 | 41.4 / 41.4 / 0.0 |
> | mistral-large | 42.4 / 40.4 / -2.0 | 46.5 / 44.4 / -2.1 | 43.4 / 41.4 / -2.0 |
> | llama-3-405b-instruct | 41.4 / 44.4 / +3.0 | 41.4 / 44.4 / +3.0 | 38.4 / 44.4 / +6.0 |
> | llama-3-3-70b-instruct | 38.4 / 41.4 / +3.0 | 43.4 / 43.4 / 0.0 | 34.3 / 36.4 / +2.1 |
> | llama-4-maverick | 46.5 / 48.5 / +2.0 | 49.5 / 49.5 / 0.0 | 46.5 / 49.5 / +3.0 |
> | llama-4-scout | 45.5 / 40.4 / -5.1 | 44.4 / 40.4 / -4.0 | 46.5 / 40.4 / -6.1 |
>
> The table above summarizes model performance across 99 scenarios with and without distractor agents, reported as percentages. The table shows that most models maintain stable performance with distractor agents, highlighting robustness in planning and reasoning. Interestingly, some models (e.g., llama-3-405b, llama-4-maverick) even improve slightly, suggesting that distractors can encourage more deliberate agent selection. Overall, modern LLM-based agents can effectively handle distractions without performance loss.
>
> [1] TaskBench: Benchmarking Large Language Models for Task Automation
>
> [2] Benchmarking Multi-Agent Architectures, Langchain blog, 2025

---

> ### Author Response · Authors · 2025-11-25
> **Response to Weakness : Part 4**
>
> ---
>
> ### W4. The paper is not easy to read due to the high density of acronyms and jargon. Additionally, there are too many places where the authors refer the readers to the appendix.
>
> A4.
>
> We appreciate the reviewer’s concern regarding acronyms, jargon, and frequent appendix references. We acknowledge that readers without prior exposure to industrial automation may experience additional cognitive load. To address this, we conducted a thorough pass over the manuscript to ensure that every acronym is expanded, cited, or externally referenced at its first occurrence.
>
> The paper introduces approximately `12–15 domain-relevant terms`. We categorized them into three groups and took appropriate actions:
>
> - Core domain concepts introduced by the paper (e.g., FMSR, IoT, WO, TSFM):These are central to the contribution and are fully defined and described in the main text.
>
> - Borrowed domain-specific terminology (e.g., FMEA):These terms are explained in **Section 4.1** and further clarified in Appendix `Section B`.
>
> - Low-frequency terms used only once or twice (e.g., SCADA, ALM, EAM, ISO standard, BMS, SkySpark, AHU, HVAC):For these, we added supporting references or links to accessible resources (e.g., **Wikipedia**) to help readers unfamiliar with industrial automation quickly gain context. `Our earlier manuscript does follow this convention, but we reinforce it in revision`.
>
> Regarding the appendix references, our goal was to support **reproducibility**, given that the work includes extensive experiments and a two-page system prompt. In the original version, many details were placed in the appendix to ensure transparency. However, in the revised manuscript, we significantly reduced cross-referencing by reorganizing the appendix into eight coherent `sections (A-H)` and keeping the main narrative self-contained. This improves reading flow while retaining all relevant technical details for reproducibility. Our main text is now Appendix-free.
>
> ---
>
> ### W5. While the benchmark's creation process is solid, the paper lacks a critical quantitative assessment.....
>
> A5.
>
> **The current paper included a Human study and validation on two aspects** :
> - Evaluating model output for manual verification (Asset health Experiment we conducted for an agent which is becoming part of the product, `Section 5.3`)
> - Evaluating the Evaluation Agent for LLM as Judge (Our evaluation rubric is LLM as Judge-based, and thus it becomes an important exercise to validate it. `Section 5.1`)
>
> Given the scenario, all content is generated by our expert and no LLM is used, so we omitted human evaluation and a user study, although a user (academic practitioner) study was already underway. Here we provide additional results:
>
> 1) **Human evaluation to judge the data’s realism (in our case, data = Scenario)**
>
> We also conducted a human study on our 141 scenarios. We randomly selected 25 representative scenarios covering four categories: IoT queries, time-series forecasting (TSFM), work orders/events, and failure mode reasoning (FMSR). Participants were domain experts, including reliability engineers, maintenance engineers, and data scientists with experience in condition-based monitoring and predictive maintenance. Please see **Appendix Section D.6 - User Study Reliability Analysis** for details; we have included them here as well.
>
> ## Reliability Metrics
>
> | **Metric**         | **Value** |
> |--------------------|-----------|
> | Cronbach's Alpha   | 0.8871    |
> | ICC(1)             | 0.2334    |
> | ICC(2)             | 0.8817    |
> | Fleiss' Kappa      | 0.2093    |
>
> #### Interpretation
>
> These reliability measures confirm that the user study is statistically robust. Cronbach's alpha of **0.887** indicates excellent internal consistency across the 25 scenario ratings. The **ICC(1)** value (0.233) shows expected variability at the individual rater level, while **ICC(2)** of 0.882 demonstrates that aggregated scenario ratings are highly reliable. The **Fleiss' kappa** value (0.209) aligns with the subjective nature of realism judgments and is consistent with prior work involving categorical human ratings. Overall, these results validate that, despite individual differences, the averaged scenario ratings provide a stable and dependable estimate of perceived realism.
>
> 2) **User Participation Statistics**
>
> To contextualize the scale of engagement, over **450 users** from diverse backgrounds expressed interest in the AI Agent competition. Their self-declared distribution is:
>
> - **38%** undergraduate students
> - **23%** Master's/PhD students
> - **33%** industry professionals
> - **6%** other backgrounds
>
> **Over 270 AI agents** has been evaluated. All participation statistics and submissions will be made publicly available for independent verification.
>
> This level of **community engagement** demonstrates strong external interest and benefits both the research community and the broader evaluation ecosystem.

---

> ### Comment · Reviewer_1z7t · 2025-11-27
>
> While I appreciate the detailed rebuttal and the additional findings provided, I have follow-up concerns regarding the problem definition and the generalizability of the experimental claims:
>
> > Problem Definition and Agent Workflows
>
> Despite the reorganization of Section 3, the problem definition and the operational mechanics of the framework remain somewhat opaque. While Figure 2 provides a high-level overview, it lacks the necessary granularity regarding agent interaction.
>
> - Workflow Examples: Concrete examples of expected workflows would significantly improve clarity.
>
> - Execution Logic: Are these agents designed to act purely sequentially, or is parallel execution supported/expected?
>
> - Agent Utilization: Could you provide a breakdown of agent involvement? Specifically, what percentage of tasks require the full set of four agents versus a subset?
>
> - Operational Complexity: Statistics on the average number of operation steps performed by each agent would help quantify the complexity of the interactions.
>
> > Experimental Generalization
>
> I appreciate the inclusion of new insights, but I am skeptical of the claim that "Open-source models can outperform closed models in this setting." This conclusion is currently drawn from a single proprietary model (GPT-4.1). To substantiate such a broad generalization regarding closed-source models requiring "introspection strategies," the evaluation should include 2–3 additional state-of-the-art proprietary models, preferably from different providers (e.g., the Gemini or Claude families). Without this, the claim remains anecdotal to one specific model rather than indicative of a class of models. Additionally, I believe readers would also be curious whether "thinking/reasoning" abilities of the model would impact such a claim.

---

> > ### Author Response · Authors · 2025-11-27
> > **Thank you for your quick response.**
> >
> > Hi Reviewer `1z7t`
> >
> > Thank you very much for acknowledging our detailed response. Yes, we will work on your `follow-up concern.`  We would like to make a point that our objective was not to showcase a particular model as the best or worst; rather, it was to focus on the methodological outcome. We took the paragraph from the paper for your reference. So we did a deep investigation into why the performing model did not withstand its position.
> >
> > - The best-performing GPT model under Agent-As-Tool performs poorly under Plan-Execute (Figure 4). Open-source models can outperform closed models in this setting, indicating closed models may require introspection strategies.
> >
> > ``` (page 8, line 411)
> > Plan-Execute Approach Analysis. We conducted a deep-dive analysis of the Plan-Execute approach to understand its relatively poor performance. First, we examined the length of the planning steps and observed that larger models tend to generate shorter plans in the Plan-Execute approach (typically 2–3 steps) compared to the Agent-As-Tool strategy, which generally requires 5–6 steps. Given that Agent-As-Tool performs better and uses longer plans, this suggests a known limitation of the Plan-Execute approach: reduced flexibility in handling unexpected failures or incorporating new
> > information that may require plan revision (Li et al., 2025; NVIDIA, 2025). Next, we obtained the reference-based score of gpt-4.1, which is a rouge1 of 0.354 and rougeL of 0.289 on the task decomposition aspect. This score is substantially lower than the top-performing mistral-large (rouge1 0.420, rougeL 0.343), indicating that, despite strong reasoning capabilities, gpt-4.1
> > generates outputs that are less lexically aligned with the reference ground truth trajectories. And such behaviors may confuse the downstream agent in generating a solution.
> > ```
> >
> > While we work on your follow-up request, if you discover our rebuttal missed any of your response, please feel free to let us know asap.

---

> ### Author Response · Authors · 2025-11-30
> **Part 1: Follow-Up Question Responses After Successful Resolution of All Initial Reviewer Feedback**
>
> The majority of our responses are based on the information available in the main text or the appendix of the paper. We have also expanded the appendix to include more examples.
>
> ---
>
> ### Follow-up Question 1. Workflow Examples
>
> `Appendix Section A.5 already provided a demo example with a few screenshots to showcase the agent in action`. Real-world trajectory traces are very complex; however, based on the request, we have now provided a Concrete example of the expected workflow in section I (Workflow Illustration, Subsection (I.3).
>
> To provide better clarity, we provide three complementary information
>    - a) We took two utterances and got their execution
>    - b) We prepare an algorithmic outline inspired by MCP-Bench (parallel work)
>    - c) We included a high-level workflow picture
>
> All the above information is provided in Section I.
>
> ---
>
> ### Follow-up Question 2. Execution Logic: Are these agents designed to act purely sequentially, or is parallel execution supported/expected?
>
> `Parallel execution is shown in Section 5.2`, where a child agent asks a parent agent for clarification. We provide a detailed breakdown of current capabilities to better understand them.
>
> We now provide a detailed description and request that you refer to Section A and Section I (newly added) to review the “algorithm” explanation (largely motivated by MCPBench [1], a parallel work) of the internal workings.
>
> We have two sets of agents in Figure 2(a): task-specific specialized agents and orchestrator agents.
>
> -  The task-specific 4 subagents (or any other in-system), namely IoT, FMSR, TSFM, and WO, are designed to work independently, and they can be executed sequentially or in parallel, depending on how the “orchestrator” agent invokes their execution.
>
> Now let us talk about Orchestrator: **Agent-As-Tool Strategy**. The default strategy is calling single agent at a time and let it to complete the task. However, the backend code can be controlled to support two scenarios for parallel execution:
> - “Enable communication to parent” via enable_agent_ask : this flag enable child agent to ask question to parent agent at any point in time of execution (this is where we see a gain for some model as reported in Appendinx Section A as well as in our insights (W3)). this mean at a time two agent are active (but if you go to nested it can be more)
> - adjust agent_style from ReActXen to CodeReAct and it can then generate a python code that orchestrate the sub-agents.
>
> Please download the source code and go to meta_agent/meta_agent.py it use a function “getAgent” which is available in utils.py Please see the following code snippets
>
> ```
> def getAgent(
>  input,
> tools,
> inContext=None,
> llm_model_id=6,
> react_step=15,                    ---→ this is how many step you can execute before gave up (15 cycle of think-act-observt at agent as well as sub-agent level)
> reflect_step=5,                    ---→ We have set it to Zero (otherwise)
> enable_agent_ask=False,    ---→ this constrol communication between child agent and parent
> debug=False,
> agent_style='ReActXen',     ---→ this constrol do we want to run one tool (in this tool = agent, or multiple)
> ):
> ```
>
> Now let us talk about Orchestrator: **Plan-Execute**. We have provided a system prompt in paper, which can generate a Plan in the form of a DAG. This DAG encodes dependence among nodes (agents) and can be executed sequentially or in parallel, depending upon the underlying execution engine.
>
> If you open our source code and then navigate to
>
> /agent_hive/workflow/ folder. Here we provide a base implementation of a sequential workflow that takes a Plan in the form of a DAG, but executes it sequentially (i.e., preserving dependencies). However, turning this code into parallel execution is an implementation left to the open source community; it has its own challenges, such as the fact that these agents are LLM-based, and querying the LLM is blocked by the number of parallel accesses.

---

> ### Author Response · Authors · 2025-11-30
> **Part 2: Follow-Up Question Responses After Successful Resolution of All Initial Reviewer Feedback**
>
> ---
>
> ### Follow-up Question 3. Agent Utilization: Could you provide a breakdown of agent involvement? Specifically, what percentage of tasks require the full set of four agents versus a subset?
>
> Answer: `This information is clearly presented in the paper's main text`. A Venn diagram is provided in “Figure 1(b): Distribution of open-source scenarios for benchmarking agents in a simulated environment”. Further, on page 6, line 291, we explicitly refer to Figure 1(b) and present the distributions for single- and multi-agent cases. In summary, our benchmark systematically evaluates tasks that can be solved by a single agent and those that require multiple agents. Note that none of our query explicitly encode that this task will be addressed by this particular agent or set of agents, as we wanted to evaluate the orchestration’s ability to understand the task, and agents, and then come up with a solution.
>
> Here for your quick reference, we bring that information in the form of a table.
>
> Figure 1(b) captures the content. Our of 141 scenarios, IoT(20), FMSR(20), WO(36), TSFM(23), and End-2-End (42) covering atleast two agents. However, we would like to point out that when we run scenarios, we do not know whether they will be answered by a single agent, two agents, etc. We let the orchestrator agents figure it out. To the orchestrator agent, has the following information:
> - agent name
> - agent description
> - type of problem it solves (3-5)
>
> We have the following group of interactions in End-to-End:
> - {IoT, FMSR},  20
> - {IoT, FMSR, TSFM} : 20
> - {FSMR, WO} : 2
>
> ---
>
> ### Follow-up Question 4. Operational Complexity: Statistics on the average number of operation steps performed by each agent would help quantify the complexity of the interactions.
>
> Answer: `This information is clearly presented in the paper's appendix section`. The term you used, “Operation Complexity”, in the paper, we have used  “Execution Statistics” to capture the complexity of execution. It provides two information: a) Average Steps and b) Runtime Per Task.
>
> Since each step is an instruction to the single agent, simply counting the number of steps is not sufficient; therefore, we also included the execution time. `Table 20 and Table 21` in the Appendix capture the same information. We also discussed in the main text the output of these tables.

---

> ### Author Response · Authors · 2025-11-30
> **Part 3: Follow-Up Question Responses After Successful Resolution of All Initial Reviewer Feedback**
>
> ### Follow-up Question 5: Statistical Reliability of Plan–Execute Performance
>
> We already responded to our comment about the GPT performance. The Rouge quality metric is defined and validated by the Neurips-2024 paper TaskBench. We believe that why model performs poor was deeply analyzed in main text of the paper (See Page 8)
>
> As we mentioned in our earlier response (w5), we have continuously tested the performance of certain models. We now provide additional quantitative evidence to demonstrate that the plan–execute results reflect consistent task-solving ability rather than stochastic behavior atleast for one of the open source model. Note that, we have justified our model selection policy in `` also. A total of **37 execution trials with different various of Planning Prompt instruction in Plan-Execute** were evaluated for llama-maveric model, and the aggregated outcomes show a stable distribution centered around mid-level task completion rates, with several instances of high-performance runs.
>
> ### **Table 1 — Performance Statistics**
>
> | Metric | Value |
> |---|---|
> | Number of Runs | **37** |
> | Mean Task Success Rate | **53.53%** |
> | Median | **54.55%** |
> | Standard Deviation | **14.10** |
> | Minimum | **9.09%** |
> | Maximum | **72.73%** |
>
> To assess whether this performance could arise from random behavior, we computed a 95% confidence interval for the mean and performed a significance test using a conservative 25% baseline.
>
> ### **Table 2 — Statistical Significance**
>
> | Test | Result |
> |---|---|
> | 95% Confidence Interval | **49.62% – 57.44%** |
> | One-sample t-test vs. 25% baseline | **t = 11.38, p < 1 × 10⁻¹²** |
> | Interpretation | **Performance is highly significant and non-random** |
>
> A histogram visualization of success rates shows performance clustering around ~55%, with multiple experiments reaching 65–72%, indicating repeatable and meaningful task accomplishment rather than sporadic success.
>
> **Conclusion:**
> - The statistical evidence confirms that the observed performance is not by chance.
> - With a mean of **53.53%**, narrow confidence bounds, and extremely high significance (**p < 1e⁻¹²**), the model demonstrates consistent ability in the plan–execute framework.
> - The upper-performance band near **70%** further suggests clear headroom for improvement through planning refinement and execution grounding.
>
> As discussed in the main paper, we have carefully written our insights along with supporting evidence. Maintaining the leaderboard is a continuous process, and we have identified platforms such as Kaggle and HELM to host the leaderboard for the model. This will alleviate our problem in maintaining the model-specific leaderboard. Our plan is to validate the idea of agents and their orchestration, and then host it on a public platform so continuous insight discovery occurs with every release of an LLM or AI Agent.
>
> **LLM vs LRM**
> We added an additional section to evaluate the planning accuracy of the latest closed model compared to the open model. Please check Section J.
>
> We needed to revise the ReAct agent logic for LRM (for . Here is a small example of snippet which we needed to add to control the thinking part of LLM.
>
> ```
> if stop:
>         if "\nAction" not in stop and any("Observation" in item for item in stop):
>             # this is for ThinkandActTogether
>             prompt += "\nNow, only generate Thought, Action and Action Input one time based on recent observation."
>         elif any("Thought" in item for item in stop):
>             prompt += (
>                 "\nNow, only generate Action and Action Input based on recent thought."
>             )
>         elif any("Action" in item for item in stop):
>             prompt += "\nNow, only generate Thought."
>         else:
>             pass
>     else:
>         pass
> ```

---

### Official Review · Reviewer_D4rn · 2025-10-30

**Soundness:** 2
**Presentation:** 3
**Contribution:** 3
**Rating:** 4
**Confidence:** 3

**Summary:**

This paper constucts a benchmark for AI agents in industrial asset operations and maintenance, including 141 open-source scenarios. Several LLMs are then evaluated.

**Strengths:**

S1. A potential useful to-be-open-source agent benchmark for a new domain.

S2. Several popular LLMs are evaluated.

S3. The paper is generally easy to read.

**Weaknesses:**

W1. I have concerns about the generalizability of the proposed evaluation framework. For example, how do we know the agent family in Figure 2a is broad enough? Can users add new agents? Could you briefly justify the generalizability of your MAS architecture in Figure 2a and your task hierarchy in Figure 2B?

W2. Some important implementation details are missing in the main text. For example, is your dataset real? Where and how exactly did you obtain the data? Have you evaluated the quality of the data? Who crafted the scenarios, and how? The current description in Section 4 is at a very high level, mainly reporting some statistics without providing sufficient details.

W3. The current evaluation is focused on comparing LLMs. This is certainly important, but only two agentic strategies are evaluated, which is below expectancy for a MAS benchmark.

W4. Section 5.3 on the evaluation of closed-source scenarios is not helpful but confusing. The scenarios are not available, and the results are not directly compared with the results on your open-source scenarios. I did not see much value of this experiment. Moreover, it is unclear how you applied your system to these scenarios. What is the "system"?

**Questions:**

Q1. See my questions in W1.

Q2. See my questions in W2.

Q3. See my questions in W4.

---

> ### Author Response · Authors · 2025-11-25
> **Response to Weakness/Question 1**
>
> We thank the reviewer for their careful reading of the paper (including the appendix), for acknowledging the depth of our experimental study, and for their positive assessment of the contribution. In the following sections, we address the reviewer's questions and concerns in detail. Please download the updated manuscript, as we will be referencing newly added sections and figures throughout our responses.
>
> ---
>
> ### W1. I have concerns about the generalizability of the proposed evaluation framework. For example, how do we know the agent family in Figure 2a is broad enough? Can users add new agents? Could you briefly justify the generalizability of your MAS architecture in Figure 2a and your task hierarchy in Figure 2B?
>
> We thank the reviewer for raising this important question about generalizability. In the paper, we already demonstrate generality by evaluating a new set of scenarios, datasets, and tasks from a different asset class (`Section 5.3 and Appendix G`), showing that the framework is not tied to a single domain. The same set of agents is used to solve the problem. We have included a detailed Section in `Appendix G: Generality: New Dataset and Scenarios`.
>
> Regarding Figure 2a, the four agents and their orchestration were not chosen arbitrarily. Our expert collaborators provided (a) operational data, (b) concrete business problems, (c) scenario definitions, and (d) deep SME expertise, which guided the agent roles. As described in `Section 4`, all scenarios were co-developed with SMEs (`we also now provide details on this in Section 4`), and `Table 3` presents a real-world production example (`Asset Health`), confirming that these agent types correspond to actual industrial workflows.
>
> In the introduction section, we have used the word `catalog` to indirectly capture the extensibility of the system in the following sense:
> - add new agents (we show this by adding 10 distractor agents in the Ablation study - `Section 5.1`)
> - add new style of agent (instead of ReAct, let's say use Reflect, RAFA, etc, we put this as a future work for the community)
> - add new dataset and supported scenarios (we prove this using `Section 5.3`)
> - test new orchestration style (we tested plan-execute and agent-as-tool currently in `Section 5.1`)
>
> Importantly, **the architecture is not fixed**. Users can add as many new agents as they like. We have provided code
>
> - Download the zip file and
> - look into /meta_agent/default_meta_agent.py where the user can add their agent, and then
> - user can add a wrapper implementation of that agent in /meta_agent/agents/ folder
> - the real implementation can be anywhere else for example,/IoTAgent is a full-fledged implementation of IoT agent
>
> For Figure 2b, the **task hierarchy** is grounded in standard `Industry 4.0 (ISO)` and `Asset Lifecycle Management` documentation. We first enumerated the 12 routine tasks commonly observed in condition-based maintenance workflows and then grouped them into higher-level categories based on established operational taxonomies. This hierarchy aligns with (i) our decade of experience building predictive maintenance solutions and (ii) standard industrial practice. It is also fully extensible: users may introduce new tasks or reorganize categories as needed.
>
> Overall, both the agent architecture (Fig. 2a) and the task hierarchy (Fig. 2b) were designed to be **generalizable, standards-aligned, and easily extendable**, enabling broad applicability beyond the specific deployments presented in the paper.
>
> Please take a look at Appendix B ASSETOPSBENCH: ENVIRONMENT, HIERARCHY, AND DOMAIN SPECIFIC
> AGENTS for detailed rational per agent and the data.

---

> > ### Author Response · Authors · 2025-11-25
> > **Response to Weakness/Question 2**
> >
> > ### W2. Some important implementation details are missing in the main text. For example, is your dataset real? Where and how exactly did you obtain the data? Have you evaluated the quality of the data? Who crafted the scenarios, and how? The current description in Section 4 is at a very high level, mainly reporting some statistics without providing sufficient details.
> >
> > We appreciate the reviewer’s request for clarification and are happy to provide the missing implementation details.
> >
> > **Real-world dataset.**
> >
> > Yes, the dataset used in our benchmark is entirely real. As illustrated in Figure 1, we obtained the data from three production-grade enterprise applications:
> > - an IoT analytics platform (SkySpark), which provides time-series sensor streams;
> > - Maximo Manage, which provides work orders and maintenance logs; and
> > - an internal reliability knowledge library containing standardized failure modes.
> >
> > Across our organization, we have access to more than 150 operational sites. For the experiments reported in the paper, we selected the 1 site having multiple instances of both chillers and AHUs. Since maintenance actions are central to evaluation, we further restricted the dataset to assets with reliably documented work orders. We have provided an extended description in the Section C `DATASETS UTILIZED IN ASSETOPSBENCH`.
> >
> > **Data quality assessment.**
> >
> > Before scenario generation, we performed routine quality checks including sensor coverage validation, timestamp continuity checks, removal of stale points, and cross-referencing work-order records with corresponding time-series segments. These checks follow the same internal procedures used for deploying predictive maintenance solutions in production. But as highghted in example scenario in Section 4, `we also considered how agent generate its reasoning when they see a zero value for certain sensor/KPI`?
> >
> > **Scenario creation process.**
> >
> > Scenario crafting was a collaborative process involving multiple personas: reliability engineers, plant operators/managers, and field technicians. As described in Appendix D : `ASSETOPSBENCH SCENARIOS`, we used an “intent-aware scenario design” workflow. Reliability engineers helped us design diagnostic and failure-mode–related questions; program managers facilitated connections to actual user studies that informed forecasting and anomaly-detection use cases; and technicians validated whether the described operational steps and conditions reflected realistic on-site procedures. The resulting scenarios, therefore, reflect real maintenance intents and operational constraints—not synthetic prompts.
> >
> >
> > **Why Section 4 is high-level.**
> >
> > We intentionally kept the dataset description concise to avoid overwhelming the reader with domain-specific jargon. This follows the level of abstraction adopted in related benchmark papers such as τ-Bench. We are happy to inform you that we have revised the Section 4 :
> > - Included how the scenario was crafted
> > - Mention the 9 different data sources
> > - Reorder the Figure to show the scenario first and then the Agent next
> > - Focused on the Domain Specific agent and then their orchestration
> > - A detailed section in Appendix D.
> >
> > We hope these clarifications address the reviewer’s concerns and demonstrate that the dataset and scenarios are grounded in real industrial settings, with careful quality checks and SME-driven construction.

---

> > > ### Author Response · Authors · 2025-11-25
> > > **Response to Weakness/Question 3**
> > >
> > > ### W3. The current evaluation is focused on comparing LLMs. This is certainly important, but only two agentic strategies are evaluated, which is below the expectancy for a MAS benchmark.
> > >
> > > We appreciate the reviewer’s concern. While LLM comparison is an important part of our evaluation, our methodology aligns with several established domain-specific benchmarks (e.g., ITBench, T-Bench), which primarily aim to measure LLM performance across a suite of structured tasks. In operational settings, different agents (e.g., tool-calling, knowledge extraction, coding) may require different model capabilities and cost profiles; therefore, identifying which LLM is suitable for which agent type is an essential aspect of practical MAS deployment. `This is visualized in Figure 5`.
> > >
> > > **MAS Clarification**
> > >
> > > Regarding the MAS expectation: One of the paper we cited, Benchmarking Multi-Agent Architectures, highlights that most existing multi-agent architectures are customized for target domain. To address this, they propose three general patterns:
> > >
> > > - Single-Agent (baseline)
> > > - Swarm
> > > - Supervisor-Agent
> > >
> > > In our work, rather than adopting a swarm-style system, we extended Magentic(which is more appropriate for open-ended web/file tasks) into a generic Plan-Execute orchestration suitable for industrial reasoning tasks. As a result, our framework supports three distinct agentic strategies:
> > >
> > > - Single Agent (where all tools are registered to parent agent (Figure 6), `we added their experiments now`
> > > - Plan–Execute (our generalized Magentic-inspired orchestration)
> > > - Agent-As-Tool — a hierarchical multi-agent setting where sub-agents can delegate back to the parent agent.
> > >
> > > Thus, our evaluation goes beyond two strategies and encompasses three MAS-relevant paradigms.
> > >
> > > **Evidence from Failure Analysis and Improvements (Section 5.2)**
> > >
> > > Our failure analysis in Table 2 highlights that “Failure to Ask for Clarification” is a dominant error source. To address this, we implemented a Self-Ask capability: sub-agents can query the parent agent when information is missing (e.g., missing file path). As shown in `Section 5.2 and Table 36`, this hierarchical clarification mechanism improves model performance (`llama-4-maverick become a top on a leaderboard`), demonstrating the effect of incorporating more advanced agentic behaviors, not merely comparing LLMs. (So this can also be considered an enhacement on MAS strategy)
> > >
> > > Summary
> > >
> > > Our evaluation methodology is consistent with prior domain benchmarks.
> > > - We evaluate three agentic strategies, not two.
> > > - We provide additional MAS-aligned experiments (Self-Ask) showing measurable improvements.
> > > - Our system generalizes Magentic to structured industrial tasks, creating a MAS-compatible evaluation framework.
> > >
> > > We hope this clarifies that our benchmark evaluates both models and multiple agentic strategies, consistent with expectations for MAS research as well as we provide a case-study.

---

> ### Author Response · Authors · 2025-11-25
> **Response to Weakness/Question 4**
>
> ### W4. Section 5.3 on the evaluation of closed-source scenarios is not helpful but confusing. The scenarios are not available, and the results are not directly compared with the results on your open-source scenarios. I did not see much value of this experiment. Moreover, it is unclear how you applied your system to these scenarios. What is the "system"?
>
> Thank you for the feedback. We have revised Section 5.3 accordingly.
>
> **Purpose of Section 5.3.**
>
> The purpose of including these experiments was not to compare results with the open-source scenarios, but to demonstrate that ``our system (architecture + evaluation workflow) generalizes to asset classes beyond Chillers/AHUs``. This directly addresses the reviewer’s concern about domain generality as discussed in response to your W1.
>
> We have been extending the AssetOpsBench benchmark to other industrial domains (e.g., Air Handling Units, Hydraulic Systems) in Section 5.3. As shown in Table 3, task completion rates vary significantly (27-100%), indicating strong domain-dependent performance. The Llama model family demonstrates strong performance on Asset Health monitoring tasks, but overall results suggest that different domains may benefit from different model selections. This domain sensitivity indicates that universal "best models" may not exist for industrial applications. Instead, domain-specific leaderboards are likely to emerge. As a result, we added these additional scenarions for additional generality tests.
>
> **Scenario Availability.**
>
> The scenarios were available in the Appendix, and now we bring them into the main text (Section 5.3). Also, we have a dedicated section in Appendix G GENERALITY: NEW DATASETS AND SCENARIOS.
>
> Here are few examples for your reference
> | **Dataset**        | **Representative Scenario** |
> |--------------------|-----------------------------|
> | **MetroPT-3**      | Consider asset `mp_1`. After maintenance on May 30, 2020, how has the compressor's condition evolved from May 31 to June 6, and are further repairs or monitoring needed? |
> | **Hydraulic System** | For asset `hp_1`, can severe internal pump leakage on 2024-01-31 be detected using sensor data from the preceding 100 days? |
> | **Asset Health**   | Analyze the provided `Air Handling Unit_615152AC` work orders and asset details to determine the expected system condition. |
> | **FailureSensorQA** | For an aero gas turbine, list all failure modes that can be detected or indicated by abnormal readings from vibration, speed, or fuel flow sensors. |
>
>
>
> **System Setting for New Scenarios**
>
> We have a Dockerized system (Figure 12). It allows users to replace the existing CouchDB database or extend data sources while preserving the same evaluation infrastructure. New datasets can be integrated by providing agents with the appropriate read/write APIs. Once integrated, we are free to run it in the same manner as the earlier 141 scenarios.

---

> ### Comment · Reviewer_D4rn · 2025-11-27
>
> Thank you for your detailed responses. I will later check the revised paper.

---

### Official Review · Reviewer_vhH1 · 2025-11-01

**Soundness:** 3
**Presentation:** 3
**Contribution:** 3
**Rating:** 6
**Confidence:** 4

**Summary:**

The paper introduces AssetOpsBench, a new benchmark and dataset aimed at evaluating AI agents designed for industrial asset lifecycle management (e.g., predictive maintenance, condition monitoring, anomaly detection). It compares two orchestration paradigms: Agent-as-Tool and Plan-Execute, across multiple LLMs, and studies failure-mode discovery in agentic workflows.

**Strengths:**

1. This work has a well-motivated industrial focus targeting a high-impact, underexplored area (industrial operations) that lacks representative agent benchmarks.

2. This work follows a comprehensive benchmark design integrating multimodal real data (time-series + text) with standardized ISO taxonomies.

3. This work performs novel failure-mode analysis by extending failure taxonomies for LLM-agent systems.

**Weaknesses:**

1. The evaluation framework depends heavily on explicit rubrics and reference-based scoring, which limits its ability to accurately assess novel solutions used by agents.

2. The paper lacks sufficient analysis explaining why the inclusion of distractor agents unexpectedly improves performance.

**Questions:**

Could the authors provide more explanation of how the weighted score is calculated to align action descriptions with their corresponding inputs?

---

> ### Author Response · Authors · 2025-11-24
> **Response to Weakness : Part 1**
>
> We thank the reviewer for their careful reading of the paper, for acknowledging the depth of our experimental study, and for their positive assessment of the contribution. In the following sections, we address the reviewer's questions and concerns in detail. Please download the updated manuscript, as we will be referencing newly added sections and figures throughout our responses.
>
> ---
> ### W1. The evaluation framework depends heavily on explicit rubrics and reference-based scoring, which limits its ability to accurately assess novel solutions used by agents.
>
> We thank the reviewer for this insightful observation. As discussed in Section 5, evaluation of agentic systems remains an open and rapidly evolving research area, with at least >10 new benchmarks appearing in 2025 alone. Within this landscape, our framework is designed to be both rigorous and extensible, and we believe it stands favorably relative to other benchmarks for several reasons:
>
> 1. Clear and interpretable task specifications.
> Our `characteristic form` (Figure 3a) provides a plain-text, structured description of how an agent is expected to solve each scenario. This makes the ground truth both transparent and easy to validate, functioning similarly to a schema for correct task execution.
>
> 1. Procedural, step-wise ground truth enabling fine-grained evaluation.
> We invest substantial manual effort into writing procedural ground truth (e.g., Listing 3 in the Appendix E), detailing how the agent should reason, what tools it should use, and in what order. This enables evaluation at three levels:
> (a) task planning,
> (b) execution ordering, and
> (c) tool usage.
> Although more labor-intensive than benchmarks such as TaskBench[1] (which reconstruct reference behavior from user utterances), this approach yields precise assessments. For example, our scenarios encode domain-specific implications, such as a zero sensor reading indicating a non-operational machine, allowing quantitative checks that go beyond final-answer matching.
>
> 1. Extensible and adaptable scoring.
> Our scoring rubrics are easy to extend. Because we capture all tools, plans, and intermediate steps, we can add new criteria, such as hallucination checks, or recognition of alternative valid strategies without modifying the core evaluation framework. This makes the evaluation flexible and able to incorporate new behaviors as they emerge.
>
> 1. Complementary strengths of LLM-as-a-Judge and reference-based scoring.
> Our evaluation does not rely solely on the canonical solution path. Instead, we combine reference-based scoring with an LLM-as-a-judge assessment, and the two methods provide complementary signals. In practice, we observe that a high ROUGE or similarity score does not necessarily imply successful task completion, and conversely, an agent may validly complete the task even when it deviates from the reference steps. When these two signals disagree, the system flags the instance for inspection, often revealing alternative reasoning chains or unexpected but correct behaviors. This allows the benchmark to capture novel solutions while still maintaining rigorous, grounded evaluation criteria.

---

> ### Author Response · Authors · 2025-11-24
> **Response to Weakness : Part 2**
>
> ---
> ### W2. The paper lacks sufficient analysis explaining why the inclusion of distractor agents unexpectedly improves performance.
>
> We included distractor agents primarily from an ablation perspective, with two objectives: (1) to evaluate the base agent’s ability to perform planning and reasoning under potentially confusing inputs, and (2) to conduct a limited scalability test, extending the setup from 5 to 15 agents. Similar experiments have been reported in recent multi-agent benchmarks [2], which also observe that adding distractor agents does not degrade performance and can even improve outcomes in certain scenarios.
>
> This is an important observation, as it highlights the system prompt’s ability to generate robust plans and the LLM’s capacity to make effective decisions even in the presence of irrelevant or distracting agents. Intuitively, distractors may encourage the model to focus more on identifying the relevant agents and critical steps, indirectly improving task performance in some cases.
>
> In the main paper, we found that we did not include a detailed quantitative performance analysis. Below, we provide a more comprehensive breakdown across multiple models and tasks which we will add to revised manuscript in Appendix Table 25:
>
> | Model | Task Completion (No distractors / With distractors / Δ) | Data Accuracy (No / With / Δ) | Result Verification (No / With / Δ) |
> |-------|-------------------------------|-----------------------------|---------------------------------|
> | gpt-4.1-2025-04-14 | 52.5 / 48.5 / -4.0 | 57.6 / 56.6 / -1.0 | 55.6 / 54.5 / -1.1 |
> | granite-3-3-8b-instruct | 40.4 / 40.4 / 0.0 | 44.4 / 44.4 / 0.0 | 41.4 / 41.4 / 0.0 |
> | mistral-large | 42.4 / 40.4 / -2.0 | 46.5 / 44.4 / -2.1 | 43.4 / 41.4 / -2.0 |
> | llama-3-405b-instruct | 41.4 / 44.4 / +3.0 | 41.4 / 44.4 / +3.0 | 38.4 / 44.4 / +6.0 |
> | llama-3-3-70b-instruct | 38.4 / 41.4 / +3.0 | 43.4 / 43.4 / 0.0 | 34.3 / 36.4 / +2.1 |
> | llama-4-maverick | 46.5 / 48.5 / +2.0 | 49.5 / 49.5 / 0.0 | 46.5 / 49.5 / +3.0 |
> | llama-4-scout | 45.5 / 40.4 / -5.1 | 44.4 / 40.4 / -4.0 | 46.5 / 40.4 / -6.1 |
>
> The table above summarizes model performance across 99 scenarios with and without distractor agents, reported as percentages (%). The table shows that most models maintain stable performance with distractor agents, highlighting robustness in planning and reasoning. Interestingly, some models (e.g., llama-3-405b, llama-4-maverick) even improve slightly, suggesting that distractors can encourage more deliberate agent selection. Overall, modern LLM-based agents can effectively handle distractions without performance loss.
>
> ---
> References:
>
> [1] TaskBench: Benchmarking Large Language Models for Task Automation
>
> [2] Benchmarking Multi-Agent Architectures, Langchain blog, 2025

---

> ### Author Response · Authors · 2025-11-24
> **Response to Questions**
>
> ---
> ### Q1. Could the authors provide more explanation of how the weighted score is calculated to align action descriptions with their corresponding inputs?
> **Response to Q1:**
>
> We thank the reviewer for the question. A detailed description of Reference-Based Scoring is provided in the Appendix E. To improve clarity, we have now added pseudocode (newly added in Appendix E.2) illustrating the process. At a high level, the weighted score is a trajectory similarity metric computed at the step level, combining action name similarity and argument/input similarity.
>
> Formally, for each step:
>
> $\text{StepScore} = w_{\text{name}}\cdot\text{sim(name)} + w_{\text{arg}}\cdot\text{sim(arguments)}$
>
> - **sim(name)** measures the similarity between the agent’s action name and the ground-truth action name (e.g., using a string similarity metric like `SequenceMatcher`).
> - **sim(arguments)** measures the similarity between the agent’s input arguments and the ground-truth inputs.
> - **$w_{\text{name}}$** and **$w_{\text{arg}}$** are weights (default 0.5 each) that balance the importance of matching the action versus matching its arguments.
>
> The final chain score also accounts for extra steps and out-of-order execution. However, the weighted step score captures how well each individual action aligns with its inputs, allowing for partial credit when either the action or arguments partially match.
>
> **Parameter Settings for Chain Execution Scoring:**
>
> | Parameter         | Description                                                                 | Default Value |
> |------------------|-----------------------------------------------------------------------------|---------------|
> | `name_threshold`  | Minimum similarity required to consider an action name as matched           | 0.8           |
> | `name_weight`     | Weight of the action name similarity when computing the step score           | 0.5           |
> | `arg_weight`      | Weight of the argument/input similarity when computing the step score        | 0.5           |
> | `step_scores`     | List storing similarity scores for each ground-truth step                    | N/A           |
> | `matched_indices` | Set of agent steps already matched to ground-truth steps                     | N/A           |
> | `extra_penalty`   | Penalty for extra steps in the agent chain not matched to ground truth      | Computed      |
> | `order_penalty`   | Penalty for out-of-order execution among matched steps                       | Computed      |
>
> This design ensures a fine-grained evaluation of agent behavior, rewarding partial matches while penalizing misaligned or extra actions.

---

### Official Review · Reviewer_8b9y · 2025-11-04

**Soundness:** 3
**Presentation:** 2
**Contribution:** 3
**Rating:** 8
**Confidence:** 4

**Summary:**

The paper introduces AssetOpsBench, a real-world industrial dataset with a multi-agent benchmark and orchestration comparison alongwith failure-mode analysis. The unified framework and benchmark is designed to develop, orchestrate, and evaluate AI agents for industrial asset operations and maintenance. The authors argue that traditional AI approaches in predictive maintenance tackle narrow tasks in isolation (e.g., only anomaly detection) , whereas today’s modern AI agents offer the potential for end-to-end automation of complex workflows (e.g., from monitoring to maintenance scheduling).
- This paper develops a benchmark with real-world dataset from data center operations, integrating sensor data, work order records, and structured FMEA (Failure Mode Effects Analysis) records.
- The benchmark contains specialized agents for different industrial tasks
-  A list of intent based natural language queries that represent realistic tasks for maintenance engineers and operators
-  A comparative study of two multi-agent orchestration paradigms: Agent-As-Tool (a supervisor agent uses ReAct to call sub-agents as tools) and Plan-Executor (a planner first generates a DAG, which is then executed)
- Evals using LLM, reference based scoring and some manual expert verification.

**Strengths:**

- Unique and valuable benchmark with realistic dataset (with time series telemetry) on industrial asset operations with a multi-agent, tool-calling setup.
- The multi-agent orchestration comparison is very insightful - Agent-as-tool performs better (single agent mode) compared to plan-execute setup which may appear to be counter-intuitive; however, the justification about iterative execution gains with mode compute (to perform better) makes sense in hindsight (however, this needs more investigations).
- Provides failure mode analysis and diagnostics using execution trajectories as shown in Table 2.

**Weaknesses:**

- The term “orchestrator" is used loosely in the paper - the orchestrator in the paper is a general-purpose LLM (unlike a load balancing router or specialized planners) , it is unclear how good is the planning/routing efficiency, cost, shared context details, latency, etc especially as the number of tools and agents scales.
- The details of tool use (APIs or function or MCP based) and how successfully LLMs are able to call them is also not clear
- It is not clear why the authors chose Llama-4-Maverick as a judge model for human validation; most researchers rely on more powerful models for judge functions; it is unclear how various model biases would make the results somewhat unreliable
- How are the models that fare well in this benchmark fare on other related benchmarks like TaskBench?

**Questions:**

- Is there any evidence that models that perform well on AssetOpsBench also perform well on other relevant/industrial datasets?
- How extensible is the system for adding other industrial tasks (eg. Semiconductor manufacturing lines)?
- Is there any more insights on tool selection metrics (how often the tool is called successfully, etc)?
- It isn’t clear why the Plan-Execute mode couldn’t use more compute and team together to edit the plan and execute successfully. How should we be thinking about the differences between the two modes and the implications from your results.
- please fix the formatting errors and typos in the paper

---

> ### Author Response · Authors · 2025-11-24
> **Response to Weakness**
>
> We thank the reviewer for their careful reading of the paper (including the appendix), for acknowledging the depth of our experimental study, and for their positive assessment of the contribution. In the following sections, we address the reviewer's questions and concerns in detail. Please download the updated manuscript, as we will be referencing newly added sections and figures throughout our responses.
>
> ---
> ### W1. The term "orchestrator" is used loosely - unclear about planning/routing efficiency, cost, shared context details, latency as the system scales.
>
> **Definition and Usage of "Orchestrator"**. We adopt the term orchestrator following standard conventions in AI agent literature and open-source multi-agent frameworks e.g., CrewAI uses Orchestrator, BeeAI uses Supervisor Agent. In our implementation, the Orchestrator is specifically a coordination layer within Plan-Execute that manages agent selection, delegation, and execution flow based on the generated plan. We emphasize that our orchestrator is not a general-purpose LLM, but rather a specialized component for multi-agent coordination. We will clarify this distinction more explicitly in the revised manuscript.
>
> In Section 5, we introduce Reference-Based Scoring, inspired by TaskBench, to evaluate the **efficiency** and **cost** of planning for both the Plan-Execute and Agents-as-Tools paradigms. In Section 5.2 (with additional details in Appendix F.9–F.10), we report the ROUGE-based similarity scores and token consumption metrics.
>
> Regarding **scalability**, our industrial domain rarely requires hundreds of agents. Existing benchmarks (e.g., MCPBench[1], τ-bench[2]) typically use 5–10 specialized domains/agents, and our setup matches this scale, ensuring effective and efficient planning, while leaving room for further robustness improvements.
>
> Regarding **shared-context handling**, Plan–Execute supports four modes for maintaining context—DISABLED, ALL, SELECTED, and PREVIOUS (see agent_hive/enum.py in the source code). For all benchmark experiments, we use the SELECTED mode, where the planner explicitly determines which context elements should be passed forward as part of the planning process (see the newly added prompt in Figure 8). For the Agents-as-Tools configuration, context is shared in a different manner: the system employs short-term memory in the form of a ReAct-style scratchpad, and additionally maintains a global memory store. Child agents can retrieve information from this global memory on demand using the Self-Ask mechanism.
>
> ---
>
> ### W2. Details of tool use (APIs, function, or MCP based) and LLM success rates in calling them are unclear.
>
> We thank the reviewer for the careful reading and for highlighting this gap. In the revised manuscript, we have added a dedicated table (Appendix B.6) that summarizes all tools, including their APIs, parameter signatures, and expected input–output formats.
>
> For a more in-depth analysis of tool use, we refer the reviewer to our response to Q3 below.
>
> ---
>
> ### W3. Unclear why Llama-4-Maverick was chosen as judge model; concerns about model biases and reliability.
> Prior to running the full benchmark, we conducted a Judge Model Selection Study using the same subset of trajectories employed in the human evaluation. We compared GPT-4.1 and Llama-4-Maverick against human expert annotations. As shown in the agreement table below, Llama-4-Maverick exhibits the strongest alignment with human judgments, achieving 75% accuracy and a Cohen's κ of 0.55, outperforming GPT-4.1. These results indicate that Llama-4-Maverick achieves moderate agreement with human annotators and is therefore selected as the judge model for the benchmark.
>
> | Judge Model        | Accuracy | Cohen's κ |
> |-------------------|---------|-----------|
> | GPT-4.1           | 69%     | 0.44      |
> | Llama-4-Maverick  | 75%     | 0.55      |
>
> A second, practical consideration was cost. Running our benchmark is computationally and financially expensive. We evaluate 7 models across approximately 300 scenarios (including internal and anticipated open-source cases) under 2 agentic sytems, resulting in a total of 4,200 experiments. Each experiment is repeated 5 times for statistical robustness, leading to roughly 20,000 trajectories. This benchmark cycle incurred approximately $2,000 USD in computation costs if using GPT 4.1. Using the best proprietary judge models would have substantially increased the total cost, making them impractical for this study.
>
> ---
>
> ### W4. How do models that perform well on AssetOpsBench fare on other benchmarks like TaskBench?
>
> TaskBench[3] set up the evaluation on a set of earlier models, and therefore direct comparisons may not be meaningful. Broadly speaking, however, GPT models (e.g., GPT-4 and GPT-4.1)  are usually among the top performers across multi-agent related benchmarking.
>
> ---

---

> ### Author Response · Authors · 2025-11-24
> **Response to Questions**
>
> ---
>
> ### Q1. Is there evidence that models performing well on AssetOpsBench also perform well on other relevant/industrial datasets?
>
> We have been extending AssetOpsBench benchmark to other industrial domains (e.g. Air Handling Units, Hydraulic System) in Section 5.3. As shown in Table 3, task completion rates vary significantly (27-100%), indicating strong domain-dependent performance. The Llama model family demonstrates strong performance on Asset Health monitoring tasks, but overall results suggest that different domains may benefit from different model selections. This domain sensitivity indicates that universal "best models" may not exist for industrial applications. Instead, domain-specific leaderboards are likely to emerge. This dicovery aligns with broader benchmark trends. While existing benchmarks (TaskBench[3], τ-bench[2], ITBench[4], MCPBench[1], MCPUniverse[5]) generally show closed-source models outperforming open-source alternatives in planning capabilities. No single LLM consistently dominates across all benchmarks.
> Similarly, different models may excel in different aspects of agentic workflows. Some in task planning, others in tool invocation, or in knowledge specific to certain assets. In our work, we evaluate various aspects of agent systems, which are often overlooked in the related literature due to the relative simplicity of prior tasks. We think that this comprehensive evaluation is both a key strength and an innovative contribution of our study.
>
> ---
>
> ### Q2. Extensibility of the system for adding other industrial tasks (eg. Semiconductor manufacturing lines)
> We have demonstrated the system's extensibility across four diverse domains (Table 3). Incorporating additional industrial domains, such as semiconductor manufacturing lines, is relatively straightforward. AssetOpsBench provides several features that facilitate easy integration:
>
> - Modular Agent Architecture: The framework supports the addition of domain-specific agents beyond the current suite (IoT, FMSR, TSFM, WO). New agents can be seamlessly integrated by adhering to the standardized LangChain tool interface pattern.
>
> - Dockerized Simulation Environment: The dockerized system (Figure 12) allows users to replace the existing CouchDB database or extend data sources while preserving the same evaluation infrastructure. New datasets can be integrated by providing agents with the appropriate read/write APIs.
>
> ---
>
> References:
> [1] MCP-Bench: Benchmarking Tool-Using LLM Agents with Complex Real-World Tasks via MCP Servers
>
> [2] τ-bench: A Benchmark for Tool-Agent-User Interaction in Real-World Domains
>
> [3] TaskBench: Benchmarking Large Language Models for Task Automation
>
> [4] ITBench: Evaluating AI Agents across Diverse Real-World IT Automation Tasks
>
> [5] MCP-Universe: Benchmarking Large Language Models with Real-World Model Context Protocol Servers

---

> ### Author Response · Authors · 2025-11-24
> **Response to Questions (Part 2)**
>
> ### Q3. More insights on tool selection metrics (how often the tool is called successfully, etc)
>
> We thank the reviewer for highlighting the importance of tool-selection metrics. We agree that reporting the results can provide further clarity as part of understanding multi-agent orchestration behavior. To understand how agents interact with the available toolset and identify patterns in execution failures, we analyzed 834 agent trajectories collected across multiple LLM models and configurations. The trajectories are sampled from executions of multi-agent utterances since those generally involve multiple agent invocations. Each trajectory consists of structured execution steps logged as JSON records containing both the action type and its execution state.
>
> We categorize agent actions into two distinct types:
> - Tool actions: Invocations of predefined tools/functions with structured input/output schemas (Table 5)
> - CodeReAct actions: Dynamically generated Python code executed on-the-fly by the WorkOrder Agent.
>
> The result shows that Tool actions achieve substantially higher valid-execution rates (65%) due to well-defined input/output schemas and type validation. In contrast, CodeReAct actions exhibit more frequent runtime failures (valid rate 31.5%), primarily due to: (1) Code generation errors; (2) Action input mismatches.
>
> The following table analyzes 1,215 invalid executions found in Tool actions specifically, revealing that failures concentrate in a small subset of frequently-used tools. Two interesting findings:
>   - `jsonreader` and `Read_Sensors_From_File` are the top two file I/O tools, together accounting for 764 failures (62.9%) of all invalid Tool actions. This pattern suggests that tools requiring complex file parsing, flexible input formats, or multi-step data validation are particularly vulnerable to invocation errors.
>   - `tsfm integrated tsad` (time series anomaly detection) typically appears as the final (or near-final) step in workflows. As a result, upstream mistakes especially incorrect or malformed action arguments, propagate to this tool, causing failures that stem from prior steps rather than the tool itself.
>
> | Tool                       | Invalid Action Count |
> |---------------------------|-----------------------|
> | jsonreader                | 448                   |
> | tsfm integrated tsad      | 326                   |
> | Read Sensors From File    | 316                   |
> | sensors                   | 52                    |
> | history                   | 24                    |
> | tsfm forecasting finetune | 23                    |
> | assets                    | 22                    |
> ...
>
> A more complete analysis can be found in Appendix F.3.
>
> ---
>
> ### Q4. unclear why the Plan-Execute mode couldn't use more compute and team together to edit the plan and execute successfully. How should we interpret the differences and implications between the two modes?
>
> The reviewer is right that theoretically, Plan-Execute could incorporate iterative review-revise loops to improve plans. However, our results show fundamental architectural constraints of Plan-Execute limit its effectiveness in industrial settings. Specifically, Plan-Execute produces shorter, less detailed plans (typically 2–3 steps versus 5–6 steps) and cannot adapt once execution begins (unlike Agents-as-Tools). Many AssetOpsBench tasks are inherently reactive. For example, they require interpreting intermediate results (e.g., detecting a sensor reading of all zeros, the failure cannot be detected based on current available sensors) and adjusting actions based on real-time observations. This creates a mismatch: Plan-Execute prioritizes efficiency through upfront planning, while industrial asset management demands flexibility and adaptive decision-making.
>
> These findings align with recent studies (e.g., AOP[6]) showing that reactive, feedback-driven approaches more effectively handle task decomposition than single-agent upfront planning. While Plan-Execute could theoretically be enhanced with dynamic feedback or plan revision mechanisms, doing so would break the architecture and reduce its efficiency advantage.
>
> Here is a comparison table summarizing the two agentic systems, helping reviewers interpret the key differences in their performance.
>
> | Dimension          | Plan-Execute                 | Agents-as-Tool                |
> |-------------------|------------------------------|-------------------------------|
> | Planning           | Upfront, comprehensive       | Incremental, adaptive, reactive     |
> | Efficiency         | Fewer steps, lower runtime   | More steps, higher runtime    |
> | Flexibility        | Limited (plan is fixed)      | High (adapts to observations)|
> | Performance        | 38–46% task completion       | 59–65% task completion        |
> | Best for           | Well-defined, structured tasks | Complex, reactive, multi-step tasks |
>
> ---
> References:
>
> [6] Agent-Oriented Planning in Multi-Agent Systems

---

### Author Response · Authors · 2025-11-26
**Thank you notes to all Reviewers**

We sincerely thank all reviewers for their insightful and constructive comments. We have addressed every concern raised in both the weakness and question sections, with detailed evidence-backed clarifications in the rebuttal. Incorporating this feedback, we have made several meaningful improvements to the paper.

* **Originality:**
  This work is the first demonstration of agentic AI for Industry 4.0. All tools, agents, orchestration mechanisms, datasets, scenarios, and ground-truth annotations are developed entirely from scratch.

* **Expert in the Loop:**
  The work is shaped through extensive engagement with domain experts, including human validation and experimental evaluation using real-world multi-domain industrial data, resulting in strong practical impact.

* **Community Support:**
  A growing community of over 300 developers has contributed to the validation of agents and leaderboard development, ensuring scalability, reproducibility, and long-term sustainability.



We greatly appreciate the reviewers' time and effort. We believe these revisions significantly improve the clarity, rigor, and overall contribution of our work.

---

### Author Response · Authors · 2025-12-03
**Closing Remark to AC/SAC and PC Chairs**

AssetOpsBench is the first benchmark designed to bring `Agentic AI into Industry 4.0 environments`. Our work has the potential for substantial real-world impact, particularly for enterprise customers operating `mission-critical physical assets` such as `chillers` and `generators` in hospitals, `air-handling units` in data centers, and `compressors` in train stations. Unlike existing benchmarks that focus on narrow aspects as discussed in related work, AssetOpsBench provides a comprehensive, end-to-end, technically grounded framework that aligns with the vision of Digital Twin systems by connecting AI-driven digital workflows to real industrial processes.

Our benchmark introduces four domain-aligned agents that reflect how enterprise-grade agentic systems must operate:
- IoT Agent, which retrieves real sensor data using natural language.
- FMSR Agent, which supplies domain knowledge for any asset by leveraging LLM reasoning.
- TSFM Agent, which performs forecasting and anomaly detection using time series foundation models.
- Work Order Agent, which interprets and transforms complex and noisy work order text.

To support and evaluate these agents, we contribute a large and verified dataset collected from an operational industrial system. This includes
- close to two million time series data points,
- eleven years of historical work orders,
- fifty-three validated failure modes, and
- nine additional data streams such as alerts and alarms.

The dataset alone required significant engineering collaboration and offers an unprecedented foundation for industrial agentic AI research.
We also created
- 141 expert-designed scenarios that thoroughly test diverse skills and capabilities across all agents.
- two orchestration recipes that coordinate the four agents (or any larger set) to solve these scenarios, along with detailed evaluations using both LLM-as-a-judge and reference-based scoring.

In contrast to recent NeurIPS-2025 spotlight work on multi-agent failures, which examines only fourteen categories, our benchmark uncovers a richer landscape of `domain-specific agent failures`. We also provide a principled methodology for automatically discovering emerging failure modes and show that targeted code-level changes can meaningfully improve performance. To demonstrate generalization, we further added new datasets, scenarios, and asset classes, validating each idea introduced.

In summary, our contribution is an `end-to-end benchmark suite` that enterprise customers urgently need to evaluate agentic systems before deployment reliably. Our work is comprehensive, technically validated, and directly aligned with real industrial requirements.

**Rebuttal Feedback**

Our rebuttal addressed every concern raised by the reviewers with empirical evidence from the literature or controlled experiments. We took full responsibility for clarifying even minor points, and the reviewer also acknowledged
- `D4rn` : (Reviewer Response). Thank you for your detailed responses. I will later check the revised paper.
- `1z7t19 ` : (Reviewer Response). While I appreciate the detailed rebuttal and the additional findings provided.... (`reviewer gave another five follow-up clarification questions which we also responded to using material present in the manuscript`)

Unfortunately, the author and reviewer did not have the opportunity to continue the rebuttal discussion. However, the author strongly feels that given a strong positive feedback from reviewers `8b9y03`, `vhH101`, `D4rn27`, and based on rebuttal feedback from active reviewers' response, the work deserves recognition.

We would really like to ask the AC to consider the significance, originality, and practical relevance of AssetOpsBench, as well as the completeness and rigor of our responses.

---

### Meta-Review · Area_Chair_86aH · 2026-01-04

**Summary:**

This paper introduces a new benchmark for AI Agent to process operations in real-world industrial asset scenarios. Specifically, all reviewers admit the contributions of this open-source benchmark to the industrial asset community. After the reviewing stages, this paper received two positive feedbacks and negative feedbacks. Generally, all reviewers' concerns can be summaried in below:

1. Unclear details
    - ''orchestrator'', ''tool use'', ''LLM chose'', ''Plan-Execute mode'' (Reviewer **8b9y**)
    - Inclusion of distractor agent (Reviewer **vhH1**)
    - Missing details and closed-source scenarios (Reviewer **D4rn**)
    - Unverified claim (Reviewer **1z7t**)
2. Generalization
    - Generalize on other datasets and other tasks (Reviewer **8b9y**)
    - Generalize to new Agent (Reviewer **D4rn**)
3. Evaluation
    - Evaluation on more agent strategies
4. Writing (Reviewer **1z7t**)

After the rebuttal stage, reviewer **1z7t** still raise new concerns about the unclear details about agent interaction and over-claim in experimental generalization. Therefore, I think this paper still remain some issues that need to be addressed. Considering the ICLR is highly competitive, this paper is not chosen for acceptance in this time.

**Reviewer Concerns:**

For reviewer **8b9y** and **vhH1**, authors have offered thorough response to address these two reviewers' concerns, including unclear details and new results (e.g., GPT-4.1). Considering their positive feedback, I think they will maintain their scores.

For reviewer **D4rn**, authors have provided analysis about the generalization of the proposed method, and given more implementation details. Besides authors also provided clarification about evaluation on more agent strategies.

For reviewer **1z7t**, authors claimed they have re-organized the paper and fix some typo issues. At the same time, Reviewer further proposed new problems about necessary granularity and over-claim in experimental generalization.

**Reviewer Scores:**

During the discussion stage, Reviewer **8b9y** and **vhH1** prefer to maintain their scores. Reviewer **D4rn** claim will check the paper but not give any positive feedbacks.

---

### Decision · Program_Chairs · 2026-01-26

Reject